# Parallel Layer Normalization for Universal Approximation

## Abstract

This paper studies the approximation capabilities of neural networks that combine layer normalization (LN) with linear layers. We prove that networks consisting of two linear layers with parallel layer normalizations (PLNs) inserted between them (referred to as PLN-Nets) achieve universal approximation, whereas architectures that use only standard LN exhibit strictly limited expressive power. We further analyze approximation rates of shallow and deep PLN-Nets under the $L^\infty$ norm as well as in Sobolev norms. Our analysis extends beyond LN to RMSNorm, and from standard MLPs to position-wise feed-forward networks, the core building blocks used in RNNs and Transformers. Finally, we provide empirical experiments to explore other possible potentials of PLN-Nets.

## 1. Introduction

Universal approximation theorems (UATs) constitute a cornerstone of the theoretical understanding of deep learning. They provide formal justification for the expressive power of deep neural networks (DNNs) and partially explain their widespread empirical success. Over the past decade, normalization techniques (Ba et al., 2016; Ioffe & Szegedy, 2015) have become ubiquitous components in modern DNN architectures, originally introduced to stabilize and accelerate training. Such techniques are now standard in convolutional neural networks (CNNs) (He et al., 2016) and Transformers (Vaswani et al., 2017a). However, to the best of our knowledge, existing UAT analyses do not explicitly account for the presence of normalization layers. This omission creates a gap between classical approximation theory and the architectures used in practice.

To bridge this gap, we study the fundamental approximation properties of networks that incorporate normalization

[1]Anonymous Institution, Anonymous City, Anonymous Region, Anonymous Country. Correspondence to: Anonymous Author <anon.email@domain.com>.

Preliminary work. Under review by the International Conference on Machine Learning (ICML). Do not distribute.

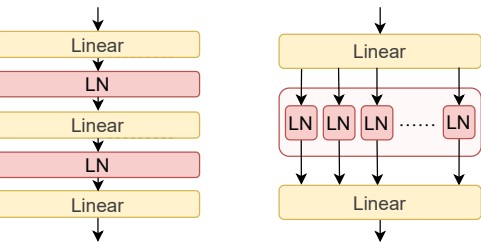

*(a)* Sequential LNs      *(b)* Parallel LNs.

*Figure 1.* (a) shows the substructures in LN-Nets, and (b) shows the substructures in PLN-Nets.

together with linear transformations. In particular, we focus on Layer Normalization (LN) (Ba et al., 2016), whose intrinsic nonlinearity was recently highlighted in (Ni et al., 2024). Ni et al. introduced a network formed by the layer-wise composition of linear transformations and LN, referred to as an LN-Net (see Figure 1a), and established its universal classification capacity when the depth is sufficiently large. In contrast, our perspective shifts from deep classification networks to wide networks for function approximation.

We show that shallow LN-Nets possess limited representation capacity (Theorem 4.1). However, when multiple LNs are applied in parallel between two linear layers (as illustrated in Figure 1b), the resulting architecture exhibits significantly stronger expressive power. We formalize this structure as Parallel Layer Normalization (PLN) (Definition 3.2) and refer to the corresponding models as PLN-Nets. We prove that shallow PLN-Nets can universally approximate $L$-Lipschitz continuous univariate functions under the $L^\infty$ norm (Theorem 4.2) when the width is large enough, and we further extend this result to multivariate functions in Sobolev spaces (Theorem 5.1). Our analysis also covers deep PLN-Nets (Theorems 5.2 and 5.3).

In addition, we present preliminary results on derivative approximation (Theorem 5.5) under generalized normalization schemes. Moreover, we discuss how our theoretical framework can be extended from multilayer perceptrons (MLPs) to position-wise feed-forward networks in recurrent neural networks (RNNs) and Transformers (Vaswani et al., 2017b) (Theorem 5.4), and from LN to RMSNorm (Zhang & Sennrich, 2019) (Lemma 4.4). Finally, we conduct experiments to empirically explore the training performance of PLN-Nets.

## 2. Related Work

The universal approximation property of neural networks has been extensively studied over the past three decades. Early results by Cybenko established that single-hidden-layer networks with infinite width and sigmoidal activations are universal approximators. This result was later generalized to networks with arbitrary bounded and nonconstant activation functions (Hornik, 1991). Building on these density arguments, Barron further characterized approximation rates for superpositions of sigmoidal functions. Subsequent works investigated how architectural constraints affect approximation power, including arbitrary depth (Gripenberg, 2003), bounded depth and width (Maiorov & Pinkus, 1999), and minimal width requirements for universal approximation (Park et al., 2020; Shin et al., 2025). Beyond classical UAT results, the expressive power of neural networks has also been analyzed through alternative complexity measures, such as the number of linear regions (Montufar et al., 2014) and VC dimension (Bartlett et al., 2019). These approximation analyses have further been extended from multilayer perceptrons (MLPs) to structured architectures, including CNNs (Oono & Suzuki, 2023), RNNs (Li et al., 2022), and Transformers (Yun et al., 2020).

A common feature of many UAT proofs is their reliance on carefully constructed weight configurations that exploit specific properties of the activation function. For instance, ReLU networks benefit from their piecewise linear structure, which allows approximation arguments to be localized to selected regions of the input space while enforcing zero output elsewhere. Such constructions play a crucial role in approximation rate analyses (Yarotsky, 2017; Yun et al., 2020). In contrast, smooth activations such as tanh produce nonzero outputs almost everywhere, making it more difficult to control approximation errors outside target regions (De Ryck et al., 2021). Because normalization layers may introduce additional nonlinear transformations (Ni et al., 2024), that may interfere with these structural properties most existing theoretical studies on representation and approximation in deep neural networks (Oono & Suzuki, 2023; Li et al., 2022; Yun et al., 2020) omit normalization for analytical tractability.

Normalization itself has been widely studied from an optimization perspective. Existing theoretical works primarily attribute its effectiveness to scale-invariance properties that stabilize training dynamics (Arora et al., 2019; Ba et al., 2016; Huang et al., 2023) and to improvements in the conditioning of the optimization problem (Cai et al., 2019; Daneshmand et al., 2020; Ghorbani et al., 2019; Karakida et al., 2019; Lyu et al., 2022; Santurkar et al., 2018). In contrast, its influence on representation and approximation capacity remains far less understood. A recent study by Ni et al. revealed the intrinsic nonlinearity of Layer Normalization and showed that LN-Nets with only three neurons per layer and $O(m)$ LN layers can shatter any set of $m$ labeled samples. Nevertheless, these results focus on classification capacity and depth scaling, leaving open the question of function approximation under width constraints.

## 3. Preliminary and Notations

We use a lowercase letter $x \in \mathbb{R}$ to denote a scalar, a boldface lowercase letter $\mathbf{x} \in \mathbb{R}^n$ to denote a vector, and a boldface uppercase letter $\boldsymbol{X} \in \mathbb{R}^{d \times n}$ to denote a matrix. Here, $\mathbb{R}$ denotes the set of real numbers, and $d, n$ are positive integers. For indexing, we use $x_i$ or $[\mathbf{x}]_i$ to denote the $i$-th entry of a vector, while $\mathbf{x}_i$ denotes the $i$-th vector in a sequence of vectors.

**Linear Layers.** Let $\mathbf{x} \in \mathbb{R}^d$ be an input vector of dimension $d$. We say $\varphi$ denotes a linear layer mapping $\mathbb{R}^d$ to $\mathbb{R}^m$, if

$$\varphi(\mathbf{x}) = \boldsymbol{W}\mathbf{x} + \boldsymbol{b}, \qquad (1)$$

for all $\mathbf{x} \in \mathbb{R}^d$, where $\boldsymbol{W} \in \mathbb{R}^{m \times d}$ and $\boldsymbol{b} \in \mathbb{R}^m$ are learnable parameters.

**Neural Networks.** We consider feed-forward neural networks that implement mappings from an input space $\mathbb{R}^{d_x}$ to an output space $\mathbb{R}^{d_y}$. Such a network is defined as a composition of linear layers and intermediate operators.

**Definition 3.1** ($\phi$-networks). A $\phi$-network (or $\phi$-Net) of depth $L$ is a function $f : \mathbb{R}^{d_x} \to \mathbb{R}^{d_y}$ of the form

$$f = \varphi_{L+1} \circ \phi_L \circ \varphi_L \circ \cdots \circ \phi_1 \circ \varphi_1, \qquad (2)$$

where $\varphi_1 : \mathbb{R}^{d_x} \to \mathbb{R}^{d_1}$, $\varphi_\ell : \mathbb{R}^{d_{\ell-1}} \to \mathbb{R}^{d_\ell}$ for $\ell = 2, \ldots, L$, and $\varphi_{L+1} : \mathbb{R}^{d_L} \to \mathbb{R}^{d_y}$ are linear layers. $\phi_\ell : \mathbb{R}^{d_\ell} \to \mathbb{R}^{d_\ell}$ for $\ell = 1, \ldots, L$ denote the same class of operators $\phi$, e.g., ReLU.

*Remark* 3.1. Each $\phi_\ell$ may represent any operator mapping $\mathbb{R}^{d_\ell}$ to $\mathbb{R}^{d_\ell}$, including normalization layers, softmax layers, attention mechanisms, and other linear and nonlinear transformations. When $\phi_\ell$ is an activation function acting on $\mathbb{R}$, it is applied element-wise to vectors in $\mathbb{R}^{d_\ell}$. Unless otherwise specified, all $\phi_\ell$ in a $\phi$-Net are assumed to belong to the same class of operators.

**Depth and Width.** For the $\phi$-Net defined in Eqn.2, we define the depth as $L$ and the width as $\max\{d_1, \ldots, d_L\}$, counting only hidden layers. For simplicity, we denote by $\mathcal{F}(\phi; L, N)$ the class of all $\phi$-networks with depth $L$ and width $N$. A function $f \in \mathcal{F}(\phi; L, N)$ is called a *shallow* $\phi$-network if $L = 1$, and a *deep* $\phi$-network if $L > 1$.

**Layer Normalization.** Layer Normalization (LN) is a fundamental component in modern deep neural networks,

originally introduced to stabilize the training process. Given a single hidden representation $\mathbf{h} = [h_1, h_2, \cdots, h_d]^\top \in \mathbb{R}^d$ with $d$ neurons, LN standardizes $\mathbf{h}$ across its coordinates as[1]

$$\hat{h}_j = [\mathrm{LN}(\mathbf{h})]_j = \frac{h_j - \mu}{\sigma}, \quad j = 1, 2, \cdots, d, \quad (3)$$

where $\mu = \frac{1}{d}\sum_{j=1}^d h_j$ and $\sigma^2 = \frac{1}{d}\sum_{j=1}^d (h_j - \mu)^2$ denote the mean and variance of the sample, respectively.

*Remark* 3.2. To ensure that Eqn.3 is well-defined on $\mathbb{R}^d$, one may either (i) add a small constant $\delta > 0$ to the denominator, or (ii) explicitly define $\mathrm{LN}(\mathbf{h})$ when $\sigma = 0$ (for example, $\mathrm{LN}(\mathbf{h}) = \mathbf{0}$). Throughout this paper, we adopt the second convention as default.

Building on LN, we introduce Parallel Layer Normalization (PLN). PLN partitions the hidden neurons into multiple groups and applies LN independently within each group. The formal definition is given below.

**Definition 3.2** (Parallel Layer Normalization). Given a single input sample $\mathbf{h} \in \mathbb{R}^d$, Parallel Layer Normalization (PLN) operates by partitioning $\mathbf{h}$ into $m$ groups $\{\mathbf{h}_i\}_{i=1}^m$ and applying LN to each group separately. Assume

$$[\mathbf{h}_1^\top, \mathbf{h}_2^\top, \cdots, \mathbf{h}_m^\top]^\top = \mathbf{h}, \quad (4)$$

the output of PLN is then defined as

$$\hat{\mathbf{h}} = \mathrm{PLN}(\mathbf{h}) = \left[\mathrm{LN}(\mathbf{h}_1)^\top, \cdots, \mathrm{LN}(\mathbf{h}_m)^\top\right]^\top. \quad (5)$$

For simplicity, we may assume each group has the same size $n_s$, which we call the *norm size*.

*Remark* 3.3. Under the simplification in Definition 3.2, PLN has a structure similar to LN-G (Ni et al., 2024), which can be viewed as a generalized form of Group Normalization (GN) (Wu & He, 2018) derived from LN (Ba et al., 2016). LN-G and GN typically emphasize flexible group counts with fixed channel widths, whereas PLN emphasizes flexible widths with a fixed norm size.

Take image models as instance, LN treats all channels of an image as a single normalization group, while GN partitions channels within each image into several groups. In contrast, PLN groups together $n_s$ neurons that originate from different channels but the same position. Analogous to how activation functions treat each neuron as a basic unit, PLN treats each group of size $n_s$ as a basic unit. As the network width increases, the number of such units grows proportionally.

With the above notation and definitions, we next conduct our analyses and present the results on the representation capacity of LN-Nets and PLN-Nets.

[1]LN typically includes additional learnable scale and shift parameters (Ioffe & Szegedy, 2015). We omit them here for simplicity, since they constitute an affine transformation.

## 4. Approximations of Univariate Functions

In this section, we study the approximation capabilities of networks composed of linear layers and LN layers. We show that shallow LN-Nets possess intrinsically limited representation power (Theorem 4.1). To achieve stronger approximation ability, we introduce the parallel LN (PLN) structure and establish its universal approximation property (Theorem 4.2). We also extend our analysis to RMSNorm (Zhang & Sennrich, 2019) in Section 4.3.

We focus primarily on relationships between different function classes and analyze basic univariate function approximation in this section. More complex settings will be discussed in Section 5.

### 4.1. The Representation Capacity of Shallow LN-Nets

We first show that shallow LN-Nets can represent certain shallow $\phi$-Nets of width 1, where $\phi$ is a specific element-wise activation function. This serves as a building block for the universal approximation property of shallow PLN-Nets. Nevertheless, shallow LN-Nets still exhibit fundamentally limited representation capacity.

**Shallow LN-Nets** A shallow LN-Net has the form

$$f = \mathrm{Linear} \circ \mathrm{LN} \circ \mathrm{Linear} \in \mathcal{F}(\mathrm{LN}(n_s); 1, n_s), \quad (6)$$

where $n_s$ denotes both the width of the LN-Net and the norm size of LN. We use $\mathrm{LN}(n_s)$ or $\mathrm{PLN}(n_s)$ to denote an LN or PLN layer with norm size $n_s$. Unlike LN-Nets, the width of a PLN-Net can be a multiple of the norm size.

Before turning to PLN-Nets, we first show that shallow LN-Nets can simulate simple activation functions, which forms the foundation of our universal approximation results for PLN-Nets. To begin, we show that a shallow LN-Net with width $n_s \geq 2$ can represent[2] a shallow sign-Net of width 1, as stated in Lemma 4.1.

**Lemma 4.1.** *Given any $\hat{f} \in \mathcal{F}(\mathrm{sign}; 1, 1)$ with $\hat{f} : \mathbb{R} \to \mathbb{R}$, there exists $f \in \mathcal{F}(\mathrm{LN}(n_s); 1, n_s)$ with $n_s \geq 2$ such that $f = \hat{f}$. In short[3],*

$$\mathcal{F}(\mathrm{sign}; 1, 1) \subseteq \mathcal{F}(\mathrm{LN}(n_s); 1, n_s), \quad (7)$$

*where $\mathrm{sign}(x)$ is the element-wise sign function taking values 1 if $x > 0$, 0 if $x = 0$, and $-1$ if $x < 0$.*

The proof is given in Appendix A.1. For the special case $n_s = 2$, we provide an intuitive illustration in Figure 2a.

[2]In this paper, we use "represent" to mean exact realization (zero approximation error), and "approximate" when small error is allowed.

[3]Throughout this paper, when comparing two function classes $\mathcal{F}_1$ and $\mathcal{F}_2$ as sets, we assume by default that they share the same input and output spaces.

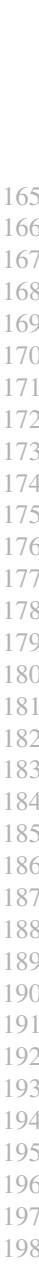

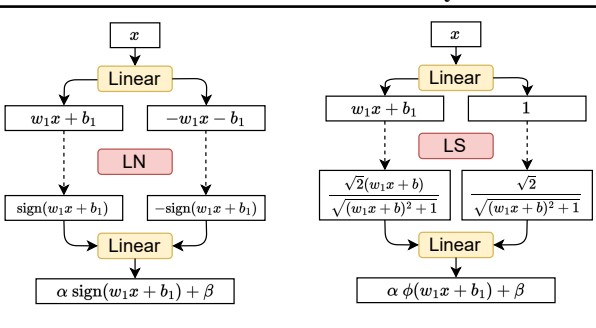

*(a)* An intuitive proof of Lemma 4.1 of the case $n_s = 2$.

*(b)* An intuitive proof of Lemma 4.2 of the case $n_s = 3$.

*Figure 2.* We can choose suitable weights in the linear layers in an LN-Net or LS-Net, then obatin the form of the target functions. (a) shows the weight construction in an LN-Net to obtain a sign-Net, while (b) the weight construction in an LS-Net to obtain a $\phi$-Net.

The requirement $n_s \geq 2$ arises from the two constraints imposed by LN on hidden representations, namely zero mean and unit variance. Increasing the width further enables smoother effective activation functions, such as $\phi(x) = x/\sqrt{x^2 + 1}$. We next show that a shallow LN-Net with width $n_s \geq 3$ can represent a shallow $\phi$-Net of width 1.

**Lemma 4.2.** *Given any $\hat{f} \in \mathcal{F}(\phi; 1, 1)$ with $\hat{f} : \mathbb{R} \to \mathbb{R}$, there exists $f \in \mathcal{F}(\mathrm{LN}(n_s); 1, n_s)$ with $n_s \geq 3$ such that $f = \hat{f}$. In short,*

$$\mathcal{F}(\phi; 1, 1) \subseteq \mathcal{F}(\mathrm{LN}(n_s); 1, n_s), \quad (8)$$

*where $\phi(x) = x/\sqrt{x^2 + 1}$.*

The proof is provided in Appendix A.4. If we use Lemma 4.4 in advance, we can simplify the proof—the argument can be reduced to proving

$$\mathcal{F}(\phi; 1, 1) \subseteq \mathcal{F}(\mathrm{LS}(n_s - 1); 1, n_s - 1). \quad (9)$$

For the special case $n_s = 3$, we also provide an intuitive illustration in Figure 2b.

Despite these expressivity results, the overall representation capacity of shallow LN-Nets remains limited. In particular, any univariate function represented by a shallow LN-Net has at most one stationary point. Otherwise, the function reduces to a sign-Net, which has at most one discontinuity and two constant regions. A formal proof is given in Appendix A.5.

**Theorem 4.1** (Weak representation capacity of shallow LN-Nets)**.** *There exists a continuous function $\hat{f}$ on $\Omega \subset \mathbb{R}$, such that for any shallow LN-Net $f \in \mathcal{F}(\mathrm{LN}(n_s); 1, n_s)$ with $n_s \geq 2$,*

$$|f - \hat{f}|_{L^\infty(\Omega)} \geq 1. \quad (10)$$

The counterexample used in the proof is the smooth elementary function $\hat{f}(x) = \cos(\pi x)$, indicating that the limitation stems from the normalization structure rather than function complexity. Please see Appendix A.5 for the detailed proof.

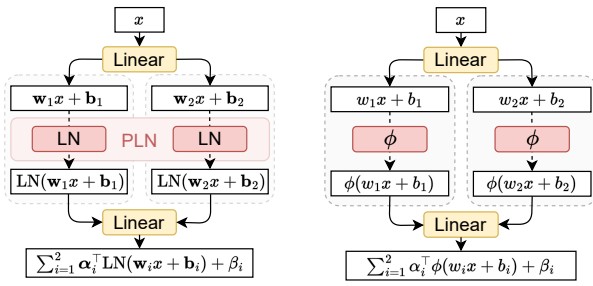

*(a)* An intuitive observation of LN-Nets in a PLN-Net.

*(b)* An intuitive observation of an element-wise $\phi$-Net.

*Figure 3.* (a) intuitively shows that such PLN-Net (the whole network) is equivalent to the sum of two LN-Nets (half of the network). Similarly, (b) shows that such $\phi$-Net width 2 is equivalent to the sum of two $\phi$-Nets of width 1.

Theorem 4.1 shows that shallow LN-Nets are not universal approximators, even if their width tends to infinity. To significantly enhance representation power, one must increase the number of LN operations—either by increasing depth or by adopting the PLN structure. We mainly study PLN-Nets in this paper.

### 4.2. Universal Approximation by Shallow PLN-Nets

Since PLN consists of multiple LN operations applied in parallel and connected by linear layers, a PLN-Net can be viewed as a sum of several LN-Nets, as formalized in Lemma 4.3.

**Lemma 4.3.** *Given $N$ shallow LN-Nets $\hat{f}_i \in \mathcal{F}(\mathrm{LN}(n_s); 1, n_s)$ with $\hat{f}_i : \mathbb{R} \to \mathbb{R}$ for $1 \leq i \leq N$, there exists a shallow PLN-Net $f \in \mathcal{F}(\mathrm{PLN}(n_s); 1, n_s N)$ such that*

$$f(x) = \sum_{i=1}^{N} \hat{f}_i(x), \quad \forall x \in \mathbb{R}. \quad (11)$$

*Conversely, any such PLN-Net can be decomposed into a sum of $N$ shallow LN-Nets.*

We provide an illustration in Figure 3a, and please refer to the formal proof in Appendix A.2.

Similarly, a shallow $\phi$-Net of width $N$ can be written as $\sum_{i=1}^{N} v_i \phi(w_i x + b_i) + c$, which is a sum of $N$ width-1 $\phi$-Nets. This together with Lemma 4.3 leads to the following corollary.

**Corollary 4.1.** *Suppose there exists $g \in \mathcal{F}(\mathrm{LN}(n_s); 1, n_s)$ that can represent any $\hat{g} \in \mathcal{F}(\phi; 1, 1)$. Then for any $\hat{f} \in \mathcal{F}(\phi; 1, N)$ with $\hat{f} : \mathbb{R} \to \mathbb{R}$, there exists a shallow PLN-Net $f \in \mathcal{F}(\mathrm{PLN}(n_s); 1, n_s N)$ such that $f = \hat{f}$. In short,*

$$\mathcal{F}(\phi; 1, N) \subseteq \mathcal{F}(\mathrm{PLN}(n_s); 1, n_s N). \quad (12)$$

Please refer to the proof and its extension to $\mathbb{R}^d \to \mathbb{R}^m$ (Corollary A.4) in Appendix A.6.

We now state the universal approximation theorem for shallow PLN-Nets.

**Theorem 4.2** (Universal approximation of univariate Lipschitz functions by shallow PLN-Nets)**.** *Given any $L$-Lipschitz continuous function $\hat{f} \in C([0,1])$, there exists a shallow PLN-Net $f \in \mathcal{F}(\text{PLN}(n_s); 1, n_s N)$ with $n_s \geq 2$ and $N = \lfloor L/(2\epsilon) \rfloor + 1$ such that*

$$|f - \hat{f}|_{L^\infty} < \epsilon. \tag{13}$$

*Proof Sketch.* The proof consists of three steps. Step 1: Construct $\bar{f} \in \mathcal{F}(\text{sign}; 1, N)$ such that $|\bar{f} - \hat{f}|_{L^\infty} < \epsilon$. Step 2: Use Lemma 4.1 and Corollary 4.1 to represent $\bar{f}$ exactly by a PLN-Net $f$. Step 3: Combine the two bounds to obtain $|f - \hat{f}|_{L^\infty} < \epsilon$.

Please refer to the complete proof in Appendix A.7.

*Remark* 4.1. There are two common conventions for defining LN when $\sigma = 0$. The proof above adopts the default convention in Remark 3.2. We also provide a proof under the alternative convention (adding a small constant $\delta$ to the denominator) in Appendix A.8. That argument further extends to S-shaped activation functions such as $\tanh$ or $\phi(x) = x/\sqrt{x^2 + 1}$.

### 4.3. Extension to RMSNorm

RMSNorm (Zhang & Sennrich, 2019) can be viewed as a variant of Layer Normalization that removes the centering operation. For simplicity, we use LS (Layer Scaling) to denote RMSNorm in this paper.

**Layer Scaling.** Given a hidden representation $\mathbf{h} = [h_1, \ldots, h_d]^\top \in \mathbb{R}^d$, Layer Scaling (LS) normalizes $\mathbf{h}$ as

$$\hat{h}_j = [\text{LS}(\mathbf{h})]_j = \frac{h_j}{\sigma}, \quad j = 1, 2, \cdots, d, \tag{14}$$

where $\sigma^2 = \frac{1}{d} \sum_{j=1}^{d} h_j^2$ is the second moment of the sample.

Analogously to LN-Nets and PLN-Nets, we can define LS-Nets and PLS-Nets. We show that PLN-Nets and PLS-Nets have essentially equivalent representation capacity.

**Lemma 4.4.** *Shallow LN-Nets with width $n_s$ and shallow LS-Nets with width $n_s - 1$ can represent each other:*

$$\mathcal{F}(\text{LN}(n_s); 1, n_s) = \mathcal{F}(\text{LS}(n_s - 1); 1, n_s - 1), \tag{15}$$

*for $n_s \geq 2$. Furthermore,*

$$\mathcal{F}(\text{PLN}(n_s); 1, n_s N) = \mathcal{F}(\text{PLS}(n_s - 1); 1, (n_s - 1)N). \tag{16}$$

Please refer to Appendix A.3 for the proof. Lemma 4.4 shows that PLN-Nets and PLS-Nets have nearly identical

representation capacity, especially for large widths. This equivalence also simplifies several arguments by allowing us to ignore the centering operation in LN. For instance, Lemma 4.4 together with Figure 2b provides an intuitive proof of Lemma 4.2 in the case $n_s = 3$.

**Summary** In this section, we established the limited representation capacity of shallow LN-Nets and the universal approximation property of shallow PLN-Nets for univariate functions. We also present the equivalent representation capacity between PLN-Nes and PLS-Nets. We extend these results to more general and higher-dimensional settings in the next section.

## 5. Approximations by Deep Neural Networks

We now extend the previous analysis to more general approximation settings, including: (i) multivariate inputs, (ii) target functions in Sobolev spaces $\mathcal{W}^{s,\infty}(\Omega)$, (iii) deep architectures (depth $\geq 2$), (iv) position-wise feed-forward networks (FFNs) in recurrent and Transformer-based models, and (v) approximation of derivatives.

We begin with multivariate approximation in Sobolev spaces $\mathcal{W}^{s,\infty}(\Omega)$[4], which provides a standard framework for measuring smoothness via bounded higher-order derivatives up to order $s$.

As in the previous section, we use $\phi$-networks as an intermediate class to transfer approximation results to PLN-Nets. We extend Corollary 4.1 to deep architectures and vector-valued functions $\hat{f} : \mathbb{R}^d \to \mathbb{R}^m$.

**Lemma 5.1.** *Given any $\hat{f} \in \mathcal{F}(\phi; L, N)$ and $\hat{f} : \mathbb{R}^d \to \mathbb{R}^m$, there exists a PLN-Net $f \in \mathcal{F}(\text{PLN}(n_s); L, n_s N)$, such that $f = \hat{f}$. In other words, we have*

$$\mathcal{F}(\phi; L, N) \subseteq \mathcal{F}(\text{PLN}(n_s); L, n_s N), \tag{17}$$

*where $\phi(x) = x/\sqrt{x^2 + 1}$ is an activation function.*

Lemma 5.1 follows by stacking the shallow constructions layer by layer. A deep $\phi$-Net can be expressed as a composition of shallow $\phi$-Nets, each of which can be represented by a shallow PLN-Net. The detailed proof is given in Appendix B.1.

Based on Lemma 5.1, we can transfer classical approximation results for $\phi$-networks to PLN-Nets. In particular, combining our structural equivalence with existing Sobolev approximation theory yields the following rate.

**Theorem 5.1.** *Let $\hat{f} \in \mathcal{W}^{s,\infty}(B^d)$ and $\|\hat{f}\|_{\mathcal{W}^{s,\infty}} \leq 1$, where $B^d = \{\mathbf{x} \in \mathbb{R}^d : \|\mathbf{x}\| \leq 1\}$, there exists a shallow*

---

[4]The classical Sobolev space is $W^{s,p}(\Omega)$; we restrict to $p = \infty$ for simplicity. Please see Appendix B.2 for the detailed definition.

PLN-Net $f \in \mathcal{F}(\text{PLN}(n_s); 1, n_s N)$, such that

$$\|f - \hat{f}\|_{L^\infty(B^d)} \leq CN^{-s/d}, \tag{18}$$

for some $C$ independent of $N$.

Theorem 5.1 builds upon Lemma 5.1 and Theorem 6.8 in (Pinkus, 1999). This theorem establishes the universal approximation capability of shallow PLN-Nets for functions in $\mathcal{W}^{s,\infty}$ over bounded domains. Please refer to Appendix B.2 for the detailed proof.

In the following, we broaden this analysis in several directions. Section 5.1 investigates approximation properties of deep PLN-Nets. Extensions to position-wise feed-forward networks in sequence models (RNNs and Transformers) are discussed in Section 5.2. Finally, Section 5.3 provides a preliminary study of approximation guarantees for derivatives.

### 5.1. Approximations by Deep PLN-Nets

ReLU is one of the most widely used activation functions in modern neural networks. If PLN-Nets are capable of representing ReLU-Nets, then many existing approximation results established for ReLU architectures can be transferred to PLN-Nets.

**Theorem 5.2** (Approximation of ReLU Networks by PLN-Nets). *Let $\hat{f} \in \mathcal{F}(\text{ReLU}; L, N)$, $f : [0, 1]^d \to \mathbb{R}^n$ and let $\epsilon > 0$. Then there exists a PLN-Net $f \in \mathcal{F}(\text{PLN}(n_s); 2L, 3n_s N)$ such that*

$$\|f - \hat{f}\|_{L^\infty([0,1]^d)} < \epsilon. \tag{19}$$

Theorem 5.2 follows from Theorem 1 in (Zhang et al., 2024). The detailed proof is provided in Appendix B.3.

While Theorem 5.2 allows us to derive many approximation results for PLN-Nets via ReLU-Nets as an intermediate representation, ReLU is inherently piecewise linear and therefore does not naturally capture higher-order smoothness, which is important in applications such as physics-informed learning (Raissi et al., 2019) and operator approximation. To address this limitation, we instead build on approximation results for smooth activations such as tanh (De Ryck et al., 2021), and extend them to PLN-Nets, as stated in Theorem 5.3.

**Theorem 5.3** (Approximation of Sobolev Functions by PLN-Nets). *Let $d, s \in \mathbb{N}$, $\delta > 0$, and $\hat{f} \in \mathcal{W}^{s,\infty}([0,1]^d)$. Then there exists a constant $C_{d,s,f}$ such that for every $N \in \mathbb{N}$ with $N > 3d/2$, there exists a two-hidden-layer PLN-Net $f$ with norm size $n_s \geq 3$, whose layer widths are at most $n_s\left(3\left\lceil\frac{s}{2}\right\rceil|P_{s-1,d+1}| + d(N-1)\right)$ and $3n_s\left\lceil\frac{d+2}{2}\right\rceil|P_{d+1,d+1}|N^d$, respectively, such that*

$$\|f - \hat{f}\|_{L^\infty([0,1]^d)} \leq (1 + \delta)\frac{C_{d,s,f}}{N^s}. \tag{20}$$

One possible choice of the constant is

$$C_{d,s,f} = \max_{0 \leq \ell \leq s} \frac{1}{(s-\ell)!} \left(\frac{3d}{2}\right)^{s-\ell} \|\hat{f}\|_{\mathcal{W}^{s,\infty}([0,1]^d)}. \tag{21}$$

Here

$$|P_{n,d}| = \binom{n + d - 1}{n}. \tag{22}$$

*Remark* 5.1. The proof follows the framework of (De Ryck et al., 2021); see Appendix B.4 for details. Note that Theorem 5.3 does not directly yield approximation guarantees for higher-order derivatives. This limitation stems from structural differences between PLN and the tanh activations considered in (De Ryck et al., 2021). We revisit this issue in Section 5.3.

By Theorems 5.2 and 5.3, we conclude that PLN-Nets possess strong universal approximation capabilities across both piecewise-linear and smooth function classes. We now extend to position-wise feed-froward networks (FFNs).

### 5.2. Approximations of Position-Wise FFNs

The results above are established for standard MLP architectures. We now extend the representation results to sequence models such as Transformers (Vaswani et al., 2017b) and RNNs, focusing on their position-wise feed-forward sublayers. The key structural difference lies in parameter sharing across sequence positions rather than in the nonlinear transformation itself.

**Sequence Linear Layers.** Let $X \in \mathbb{R}^{s \times d}$ denote a sequence input with length $s$ and token dimension $d$. A sequence linear layer mapping token dimension $d \to m$ is defined as

$$\varphi(X) = XW + \mathbf{1}_s b^\top, \tag{23}$$

for all $X \in \mathbb{R}^{s \times d}$, where $W \in \mathbb{R}^{d \times m}$ and $b \in \mathbb{R}^m$ are the learnable parameters.

**Theorem 5.4** (Represent Position-Wise FFNs by PLN-Nets). *Let $\hat{\varphi}_1$ and $\hat{\varphi}_2$ be two sequence linear layers mapping token dimensions $d_x \to N$ and $N \to d_y$, respectively, and let $\phi(x) = x/\sqrt{x^2 + 1}$. Define the position-wise FFN mapping $\hat{f} = \hat{\varphi}_2 \circ \phi \circ \hat{\varphi}_1$. Then there exist sequence linear layers $\varphi_1$ and $\varphi_2$ mapping token dimensions $d_x \to n_s N$ and $n_s N \to d_y$, respectively, such that*

$$f = \varphi_2 \circ \text{PLN}(n_s) \circ \varphi_1 \tag{24}$$

*exactly represents $\hat{f}$ for all sequence inputs.*

Theorem 5.4 is a parameter-sharing extension of the MLP representation result (Corollary A.4) and applies to the position-wise feed-forward modules in Transformers and RNNs. A detailed proof is provided in Appendix B.5.

*Remark* 5.2. In sequence models, Layer Normalization operates at the token level, while PLN groups $n_s$ neurons within each token. This grouping ensures that each PLN block corresponds to one activation $\phi$ in the equivalent feed-forward representation.

*Remark* 5.3. The same argument applies to $1 \times 1$ convolution layers, which preserve spatial resolution similarly to how sequence linear layers preserve sequence length. However, extending the result to larger convolution kernels is nontrivial and remains an open problem for future work.

### 5.3. Approximation of Derivatives

In this section, we present a preliminary study on derivative approximation. To facilitate the analysis, we first generalize our framework to a broader class of normalization schemes; the necessity of this generalization will be discussed later.

**Definition 5.1** (($p, q$)-normalization). Given a hidden representation $\mathbf{h} = [h_1, h_2, \cdots, h_d]^\top \in \mathbb{R}^d$ of a single sample with $d$ neurons, let $p \geq q \geq 1$. The ($p, q$)-normalization transforms $\mathbf{h}$ into $\hat{\mathbf{h}}$ across neurons as

$$\hat{h}_i = [\phi_{p,q}^{\mathrm{norm}}(\mathbf{h})]_i = h_i^{\frac{p}{q}} / |\overline{h^p}|^{\frac{1}{q}}, \qquad (25)$$

for $i = 1, 2, \ldots, d$, where $\overline{h^p} = \frac{1}{d} \sum_{i=1}^{d} |h_i|^p$ denotes the sample-wise mean of $|h_i|^p$.

The ($p, q$)-normalization applies a power transformation before normalization and enforces the constraint $\|\hat{\mathbf{h}}\|_q^q = d$. Similar to Lemma 4.2, we can derive the corresponding ($p, q$)-activation function $\phi_{p,q}$:

$$\phi_{p,q}(x) = \frac{x^{p/q}}{(|x|^p + 1)^{1/q}}. \qquad (26)$$

By definition, $\phi_{p,q}(+\infty) = 1$. We further require $\phi_{p,q}$ to be an odd function; therefore, throughout the paper we assume $p, q \in \mathbb{N}$ and that $p/q$ is an odd integer. As a special case, $\phi_{2,2}(x) = x/\sqrt{x^2 + 1}$ corresponds to the activation used in our previous universal approximation results. For general $\phi_{p,q}$, we extend the approximation theory to Sobolev norms.

**Theorem 5.5** (Approximation in Sobolev spaces by $\phi_{p,q}$ networks). *Let $p \geq q \geq 1$ be even integers such that $r := p/q$ is an odd integer. Let $d, s \in \mathbb{N}$, $k \in \mathbb{N}_0$ with $r > \frac{k(s+d+k)}{s-k}$, $\delta > 0$, and let $\hat{f} \in \mathcal{W}^{s,\infty}([0,1]^d)$. Then there exist constants $C_1$ and $C > 0$ such that for every $N \in \mathbb{N}$ with $N > N_0(d)$, there exists a two-hidden-layer $\phi_{p,q}$ neural network $f$ satisfying*

$$\|f - \hat{f}\|_{L^\infty([0,1]^d)} \leq (1 + \delta)\frac{C_1}{N^s}, \qquad (27)$$

*and for $1 \leq k < s$,*

$$\|f - \hat{f}\|_{W^{k,\infty}([0,1]^d)} \leq C \frac{1 + \delta}{\delta^{k/r} N^{s-k-(s+d+k)k/r}}, \qquad (28)$$

*where the constants $C$ and $C_1$ are independent of $\delta$ and $N$.*

*Remark* 5.4. We provide additional details for Theorem 5.5. The threshold $N_0(d)$ is defined as $N_0 = 3d/2$ if $\hat{f} \in C^s([0,1]^d)$, and $N_0 = 5d$ otherwise.
The network widths are bounded by

$$\frac{p+1}{2}\left(p\left\lceil\frac{s-1-r}{p}\right\rceil+r+1\right)|P_{s-1,d+1}|+d(N-1) \quad (29)$$

for the first hidden layer, and by

$$\frac{p+1}{2}\left(p\left\lceil\frac{d+1-r}{p}\right\rceil + r + 1\right)|P_{d+1,d+1}| \qquad (30)$$

for the second hidden layer, where $|P_{s,d}|$ is defined in Eqn.22. Explicit expressions for $C$ and $C_1$ are given in Appendix C. Also see the proof of Theorem 5.5 there.

*Remark* 5.5. The convergence rate differs from the classical $N^{s-k}$ rate. We provide some intuition: (i) Compared with (De Ryck et al., 2021), the key difference lies in the decay behavior of derivatives. While derivatives of $\tanh$ decay exponentially, those of $\phi_{p,q}$ exhibit only polynomial decay. As a result, logarithmic factors in the original analysis are replaced here by polynomial terms, leading to the factor $N^{(s+d+k)k/r}$. (ii) The present bound may not be tight. The dominant error term originates from Lemma C.9. Refining this step could potentially improve the convergence rate, which we leave for future work.

*Remark* 5.6. When $p = q$, the normalization reduces to standard $L^p$ normalization. However, in this case the right-hand side of Eqn.28 does not necessarily converge to zero for $k \geq 1$. This explains why derivative approximation was not included in Theorem 5.3.

**Summary**  In this section, we extend our theory to deeper PLN-Nets and broaden the representation results from MLPs to position-wise FFNs in RNNs and Transformers. We also provide a preliminary theoretical analysis of derivative approximation for deep PLN-Nets with parallel ($p, q$)-normalizations.

## 6. Experiments Observation

We have proved that PLN-Nets possess strong representation capacity, yet they are not widely used in current neural network architectures. Therefore, we include a set of preliminary experiments to explore the practical potential of PLN-Nets.

We first compare PLN-Nets with four types of $\phi$-Nets. The activation functions $\phi$ considered here include ReLU : $x \to \max(0, x)$, SiLU : $x \to x/(1 + e^{-x})$, Tanh : $x \to (e^x - e^{-x})/(e^x + e^{-x})$, and SAT : $x \to \sin(\arctan x) = x/\sqrt{x^2 + 1}$.

*Table 1.* We record the best relative $\mathcal{L}_2$ error of MLP with different activation layers and width under different experiment settings on solving Helmholtz equation. The best result is marked in bold.

| Width | PLN(4) | SAT | Tanh | ReLU |
|-------|--------|-----|------|------|
| 128 | **0.000035** | 0.000104 | 0.000250 | 0.993593 |
| 256 | **0.000036** | 0.000052 | 0.000188 | 0.998485 |

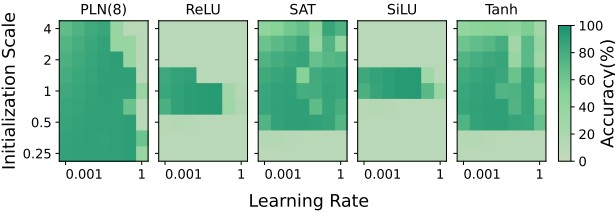

*Figure 4.* We record the average validation accuracy (higher if the green color is darker) of different activation layers under different initializations and learning rates. For each subfigure, the learning rate increase from 0.001 (the left) to 1 (the right). Besides, we adopt He initialization (He et al., 2015) as the baseline (the middle row), and scale the weights by factors from $1/4$ (the bottom) to 4 (the top) relative to this baseline.

**Differential Equation Solving with PINNs.** We first investigate the performance of PLN in MLPs. We employ a physics-informed neural network (PINN) (Raissi et al., 2019) to solve the Helmholtz equation, which requires accurate function values as well as smoothness. All experiments are conducted using MLPs without normalization layers. For each activation layer, we perform independent runs with five different random seeds among different depths, loss weights, and optimizers. We use the relative $\mathcal{L}_2$ error (Raissi et al., 2019) to evaluate the overall fitting accuracy of the model. In Table 1, we report the best result among all configurations for the MLP width 128 and 256. We observe that PLN(4) achieves the best performance among all configurations, which preliminarily indicates the potential of PLN in PINNs. The detailed experimental settings are provided in Appendix D.1.

**CIFAR-10 Classification with VGGs** We next move to a CNN architecture to investigate the performance of PLN. We conduct image classification experiments on the CIFAR-10 dataset (Krizhevsky et al., 2009) using VGG-16 (Simonyan & Zisserman, 2014) with different activation layers. We sweep different learning rates, and initialization methods to investigate the training stability. For each hyperparameter configuration, we perform three independent runs with different random seeds and report the mean validation accuracy in Figure 4. We observe that PLN is relatively less sensitive to initialization compared with other activation layers in VGG-16, which suggests its potential under rough initialization conditions. More details are provided in Appendix D.2.

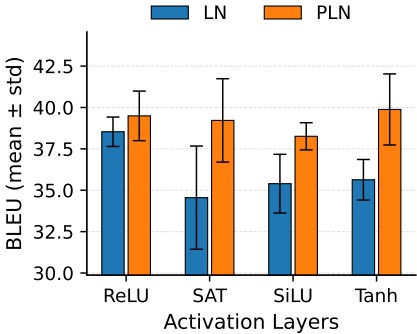

*Figure 5.* BLEU scores (the higher is the better) for different combinations of normalization layers and activation functions in the Multi30k task.

**Machine Translation with Transformers** Motivated by Theorem 4.1 and Theorem 4.2, we explore replacing the LN in Transformers with PLN to examine potential performance improvements. We conduct machine translation experiments on the Multi30k dataset (Elliott et al., 2016) using Transformers (Vaswani et al., 2017a) with different activation layers. We compare PLN with LN as normalization methods and report the mean and standard deviation of BLEU scores over three random seeds in Figure 5. We observe that replacing LN with PLN indeed improved translation performance in this setting. More details can be found in Appendix D.3.

## 7. Conclusion

**Conclusions.** We studied the approximation properties of neural networks that combine linear layers with Layer Normalization (LN) and Parallel Layer Normalization (PLN). We showed that sufficiently wide PLN-Nets possess universal approximation capability, whereas LN-only architectures have substantially more limited expressivity. We also established approximation rates under the $L^\infty$ norm. Our representation results further extend from LN to RMSNorm and from MLPs to position-wise FFNs in Transformers and RNNs. We additionally provided a preliminary analysis of derivative approximation. Limited empirical results offer supporting evidence for the practical potential of PLN-Nets.

**Limitations and Future Work.** Our analysis focuses on architectures composed of linear layers and LN/PLN-type normalizations. Their interaction with standard nonlinear activations in conventional networks remains open. Extending the theory to convolutional layers with larger kernels also requires further work. The derivative approximation results are still preliminary and may be sharpened in future studies. Finally, our limited experiments only reveal the potential of PLN-Nets but have not widely verified its performances.

## Impact Statement

This paper presents work whose goal is to advance the field of Machine Learning. There are many potential social consequences of our work, none which feel must be specifically highlighted here.

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

## A. Mathematical Proofs in Section 4

### A.1. Proof of Lemma 4.1

**Lemma 4.1**  Given any $\hat{f} \in \mathcal{F}(\text{sign}; 1, 1)$ with $\hat{f} : \mathbb{R} \to \mathbb{R}$, there exists $f \in \mathcal{F}(\text{LN}(n_s); 1, n_s)$ with $n_s \geq 2$ such that $f = \hat{f}$. In short,

$$\mathcal{F}(\text{sign}; 1, 1) \subseteq \mathcal{F}(\text{LN}(n_s); 1, n_s), \tag{31}$$

where $\text{sign}(x)$ is the element-wise sign function taking values 1 if $x > 0$, 0 if $x = 0$, and $-1$ if $x < 0$.

*Proof.* For any $\hat{f} \in \mathcal{F}(\text{sign}; 1, 1)$ and $\hat{f} : \mathbb{R} \to \mathbb{R}$, we have $\hat{f}(x) = w_2 \, \text{sign}(w_1 x + b_1) + b_2$.

For $f \in \mathcal{F}(\text{LN}(n_s); 1, n_s)$ with $n_s \geq 2$, we construct $f(x) = \boldsymbol{v}_2^\top \text{LN}(\boldsymbol{v}_1 x + \boldsymbol{c}_1) + c_2$, where $\boldsymbol{v}_1, \boldsymbol{c}_1, \boldsymbol{v}_2 \in \mathbb{R}^{n_s}$ and $c_2 \in \mathbb{R}$.

Let

$$\boldsymbol{v}_1 = \begin{bmatrix} w_1 \\ -w_1 \\ \boldsymbol{0}_{n_s-2} \end{bmatrix}, \boldsymbol{c}_1 = \begin{bmatrix} b_1 \\ -b_1 \\ \boldsymbol{0}_{n_s-2} \end{bmatrix}, \boldsymbol{v}_2 = \begin{bmatrix} \sqrt{2/n_s}\,w_2 \\ 0 \\ \boldsymbol{0}_{n_s-2} \end{bmatrix}, c_2 = b_2. \tag{32}$$

We can identify that the first element of $\text{LN}(\boldsymbol{v}_1 x + \boldsymbol{c}_1)$ is

$$\frac{w_1 x + b_1}{\sqrt{2(w_1 x + b_1)^2/n_s}} = \sqrt{\frac{n_s}{2}} \frac{w_1 x + b_1}{|w_1 x + b_1|} = \sqrt{\frac{n_s}{2}} \, \text{sign}(w_1 x + b_1), \tag{33}$$

if $w_1 x + b_1 \neq 0$.

For the case $w_1 x + b_1 = 0$, we have $\sigma^2 = 2(w_1 x + b_1)^2/n_s = 0$, then $\text{LN}(\boldsymbol{v}_1 x + \boldsymbol{c}_1) = \boldsymbol{0}$ by its definition.

Therefore, the first element of $\text{LN}(\boldsymbol{v}_1 x + \boldsymbol{c}_1)$ is equal to $\sqrt{\frac{n_s}{2}} \, \text{sign}(w_1 x + b_1)$, thus

$$\begin{aligned} f(x) &= \boldsymbol{v}_2^\top \text{LN}(\boldsymbol{v}_1 x + \boldsymbol{c}_1) + c_2 \\ &= \sqrt{2/n_s}\,w_2 \cdot \sqrt{\frac{n_s}{2}} \, \text{sign}(w_1 x + b_1) + c_2 \\ &= w_2 \, \text{sign}(w_1 x + b_1) + b_2 = \hat{f}(x). \end{aligned} \tag{34}$$

This completes the proof. □

We can extend Lemma 4.1 to the version $\hat{f} : \mathbb{R}^d \to \mathbb{R}^m$.

**Corollary A.1.**  *Given any $\hat{f} \in \mathcal{F}(\text{sign}; 1, 1)$ with $\hat{f} : \mathbb{R}^d \to \mathbb{R}^m$, there exists $f \in \mathcal{F}(\text{LN}(n_s); 1, n_s)$ with $n_s \geq 2$, such that $f = \hat{f}$. In short,*

$$\mathcal{F}(\text{sign}; 1, 1) \subseteq \mathcal{F}(\text{LN}(n_s); 1, n_s), \tag{35}$$

*where $\text{sign}(x)$ is the element-wise sign function, which takes 1 if $x > 0$, 0 if $x = 0$, and $-1$ if $x < 0$.*

*Proof.* For any $\hat{f} \in \mathcal{F}(\text{sign}; 1, 1)$ and $\hat{f} : \mathbb{R}^d \to \mathbb{R}^m$, we have $\hat{f}(x) = \boldsymbol{w}_2 \, \text{sign}(\boldsymbol{w}_1^\top \mathbf{x} + b_1) + \boldsymbol{b}_2$, where

$$\boldsymbol{w}_1 \in \mathbb{R}^d, b_1 \in \mathbb{R}, \boldsymbol{w}_2, \boldsymbol{b}_2 \in \mathbb{R}^m. \tag{36}$$

We can just adjust the construction as

$$\boldsymbol{V}_1 = \begin{bmatrix} \boldsymbol{w}_1^\top \\ -\boldsymbol{w}_1^\top \\ \boldsymbol{O}_{(n_s-2)\times d} \end{bmatrix} \in \mathbb{R}^{n_s \times d}, \boldsymbol{c}_1 = \begin{bmatrix} b_1 \\ -b_1 \\ \boldsymbol{0}_{n_s-2} \end{bmatrix} \in \mathbb{R}_2^n, \boldsymbol{V}_2 = \begin{bmatrix} \sqrt{2/n_s}\,\boldsymbol{w}_2^\top \\ \boldsymbol{O}_{1\times d} \\ \boldsymbol{O}_{(n_s-2)\times m} \end{bmatrix} \in \mathbb{R}^{n_s \times m}, \boldsymbol{c}_2 = \boldsymbol{b}_2 \in \mathbb{R}^m, \tag{37}$$

and $f(\mathbf{x}) = \boldsymbol{V}_2^\top \text{LN}(\boldsymbol{V}_1 \mathbf{x} + \boldsymbol{c}_1) + \boldsymbol{c}_2$.

The following proof is almost the same as that in the proof of Lemma 4.1. □

### A.2. Proof of Lemma 4.3

**Lemma 4.3** Given $N$ shallow LN-Nets $\hat{f}_i \in \mathcal{F}(\mathrm{LN}(n_s); 1, n_s)$ with $\hat{f}_i : \mathbb{R} \to \mathbb{R}$ for $1 \leq i \leq N$, there exists a shallow PLN-Net $f \in \mathcal{F}(\mathrm{PLN}(n_s); 1, n_s N)$ such that

$$f(x) = \sum_{i=1}^{N} \hat{f}_i(x), \quad \forall x \in \mathbb{R}. \tag{38}$$

Conversely, any such PLN-Net can be decomposed into a sum of $N$ shallow LN-Nets.

*Proof.* Given $N$ shallow LN-Nets $\hat{f}_i \in \mathcal{F}(\mathrm{LN}(n_s); 1, n_s)$ for $1 \leq i \leq N$, set $\hat{f}_i(x) = \boldsymbol{w}_{i2}^{\top}\mathrm{LN}(\boldsymbol{w}_{i1}x + \boldsymbol{b}_{i1}) + b_{i2}$.

For $f \in \mathcal{F}(\mathrm{PLN}(n_s); 1, n_s N)$, set $f(x) = \boldsymbol{w}_2^{\top}\mathrm{PLN}(\boldsymbol{w}_1 x + \boldsymbol{b}_1) + b_2$, where $\boldsymbol{w}_1, \boldsymbol{w}_2, \boldsymbol{b}_1 \in \mathbb{R}^{n_s N}$.

Let

$$\boldsymbol{w}_1 = \begin{bmatrix} \boldsymbol{w}_{11} \\ \boldsymbol{w}_{21} \\ \vdots \\ \boldsymbol{w}_{N1} \end{bmatrix}, \boldsymbol{b}_1 = \begin{bmatrix} \boldsymbol{b}_{11} \\ \boldsymbol{b}_{21} \\ \vdots \\ \boldsymbol{b}_{N1} \end{bmatrix}, \boldsymbol{w}_2 = \begin{bmatrix} \boldsymbol{w}_{12} \\ \boldsymbol{w}_{22} \\ \vdots \\ \boldsymbol{w}_{N2} \end{bmatrix}, b_2 = \sum_{i=1}^{N} b_{i2}. \tag{39}$$

By Definition 3.2, under the setting that $\boldsymbol{P}$ is identity and fixing each size of the groups as the same $n_s$, we have the input of PLN is

$$\mathbf{h} = \boldsymbol{w}_1 x = \begin{bmatrix} \boldsymbol{w}_{11}x + \boldsymbol{b}_{11} \\ \boldsymbol{w}_{21}x + \boldsymbol{b}_{21} \\ \vdots \\ \boldsymbol{w}_{N1}x + \boldsymbol{b}_{N1} \end{bmatrix}. \tag{40}$$

Since each $\boldsymbol{w}_{i1}x + \boldsymbol{b}_{i1} \in \mathbb{R}^{n_s}$, we have $\mathbf{h}_i = \boldsymbol{w}_{i1}x + \boldsymbol{b}_{i1} \in \mathbb{R}^{n_s}$. Therefore, we obtain

$$\mathrm{PLN}(\mathbf{h}) = \begin{bmatrix} \mathrm{LN}(\mathbf{h}_1) \\ \mathrm{LN}(\mathbf{h}_2) \\ \vdots \\ \mathrm{LN}(\mathbf{h}_N) \end{bmatrix} = \begin{bmatrix} \mathrm{LN}(\boldsymbol{w}_{11}x + \boldsymbol{b}_{11}) \\ \mathrm{LN}(\boldsymbol{w}_{21}x + \boldsymbol{b}_{21}) \\ \vdots \\ \mathrm{LN}(\boldsymbol{w}_{N1}x + \boldsymbol{b}_{N1}) \end{bmatrix}. \tag{41}$$

Furthermore, we obtain that

$$\begin{aligned} f(x) &= \boldsymbol{w}_2^{\top}\mathrm{PLN}(\boldsymbol{w}_1 x + \boldsymbol{b}_1) + b_2 \\ &= \sum_{i=1}^{N} \boldsymbol{w}_{i2}^{\top}\mathrm{LN}(\boldsymbol{w}_{i1}x + \boldsymbol{b}_{i1}) + \sum_{i=1}^{N} b_{i2} \\ &= \sum_{i=1}^{N} \hat{f}_i(x). \end{aligned} \tag{42}$$

Conversely, given $f \in \mathcal{F}(\mathrm{PLN}(n_s); 1, n_s N)$, set $f(x) = \boldsymbol{w}_2^{\top}\mathrm{PLN}(\boldsymbol{w}_1 x + \boldsymbol{b}_1) + b_2$, where $\boldsymbol{w}_1, \boldsymbol{w}_2, \boldsymbol{b}_1 \in \mathbb{R}^{n_s N}$, we separate the parameters by Eqn.39. Furthermore, we can construct $N$ shallow LN-Nets $\hat{f}_i \in \mathcal{F}(\mathrm{LN}(n_s); 1, n_s)$ for $1 \leq i \leq N$, whose sum is equal to $f$.

This completes the proof. $\square$

We can also extend to the high-dimension version.

**Corollary A.2.** *Given any $N$ shallow LN-Nets $\hat{f}_i \in \mathcal{F}(\mathrm{LN}(n_s); 1, n_s)$ and $\hat{f}_i : \mathbb{R}^d \to \mathbb{R}^m$, for $1 \leq i \leq N$, there exists a shallow PLN-Net $f \in \mathcal{F}(\mathrm{PLN}(n_s); 1, n_s N)$, such that*

$$f(x) = \sum_{i=1}^{N} \hat{f}_i(x), \forall x \in \mathbb{R}. \tag{43}$$

*It holds vice versa, namely $N$ shallow LN-Nets $\hat{f}_i \in \mathcal{F}(\mathrm{LN}(n_s); 1, n_s)$ for $1 \le i \le N$ can represent a shallow PLN-Net $f \in \mathcal{F}(\mathrm{PLN}(n_s); 1, n_s N)$ by addition.*

*Proof.* Given $N$ shallow LN-Nets $\hat{f}_i \in \mathcal{F}(\mathrm{LN}(n_s); 1, n_s)$ for $1 \le i \le N$, set $\hat{f}_i(x) = \boldsymbol{W}_{i2}\mathrm{LN}(\boldsymbol{W}_{i1}\mathbf{x} + \boldsymbol{b}_{i1}) + \boldsymbol{b}_{i2}$, where

$$\boldsymbol{W}_{i1} \in \mathbb{R}^{n_s \times d}, \boldsymbol{b}_{i1} \in \mathbb{R}^{n_s}, \boldsymbol{W}_{i2} \in \mathbb{R}^{m \times n_s}, \boldsymbol{b}_{i2} \in \mathbb{R}^m. \tag{44}$$

For $f \in \mathcal{F}(\mathrm{PLN}(n_s); 1, n_s N)$, set $f(x) = \boldsymbol{W}_2^\top \mathrm{PLN}(\boldsymbol{W}_1 x + \boldsymbol{b}_1) + \boldsymbol{b}_2$, where

$$\boldsymbol{W}_1 \in \mathbb{R}^{n_s N \times d}, \boldsymbol{b}_1 \in \mathbb{R}^{n_s N}, \boldsymbol{W}_2 \in \mathbb{R}^{m \times n_s N}, \boldsymbol{b}_2 \in \mathbb{R}^m. \tag{45}$$

This time, we construct that

$$\boldsymbol{W}_1 = \begin{bmatrix} \boldsymbol{W}_{11} \\ \boldsymbol{W}_{21} \\ \vdots \\ \boldsymbol{W}_{N1} \end{bmatrix}, \boldsymbol{b}_1 = \begin{bmatrix} \boldsymbol{b}_{11} \\ \boldsymbol{b}_{21} \\ \vdots \\ \boldsymbol{b}_{N1} \end{bmatrix}, \boldsymbol{W}_2 = \begin{bmatrix} \boldsymbol{W}_{12} & \boldsymbol{W}_{22} & \cdots & \boldsymbol{W}_{N2} \end{bmatrix}, \boldsymbol{b}_2 = \sum_{i=1}^{N} \boldsymbol{b}_{i2}. \tag{46}$$

The following proof is almost the same as that in the proof of Lemma 4.3. $\qquad\square$

### A.3. Proof of Lemma 4.4

Lemma 4.4 can simplify most the proofs, so we prove it at the early time.

**Lemma 4.4** Shallow LN-Nets with width $n_s$ and shallow LS-Nets with width $n_s - 1$ can represent each other, namely

$$\mathcal{F}(\mathrm{LN}(n_s); 1, n_s) = \mathcal{F}(\mathrm{LS}(n_s - 1); 1, n_s - 1), \tag{47}$$

for $n_s \ge 2$. Furthermore,

$$\mathcal{F}(\mathrm{PLN}(n_s); 1, n_s N) = \mathcal{F}(\mathrm{PLS}(n_s - 1); 1, (n_s - 1)N). \tag{48}$$

To begin with, we require Lemma A.2 first, which is proved based on Lemma 6 in (Ni et al., 2024).

**Lemma A.1** (Lemma 6 in (Ni et al., 2024))**.** *There is some orthogonal matrix $\boldsymbol{Q} \in \mathbb{R}^{d \times d}$, such that $\boldsymbol{z} = \boldsymbol{Q}\begin{bmatrix} \mathbf{x} \\ 0 \end{bmatrix} \in \{\boldsymbol{z} \in \mathbb{R}^d : z^{(1)} + \cdots + z^{(d)} = 0\}$ (namely $\boldsymbol{z}$ is centralized), for $\mathbf{x} \in \mathbb{R}^{d-1}$,*

**Lemma A.2.** *Denote $\mathrm{LN}(\cdot)$ as the LN operation in $\mathbb{R}^d (d \ge 3)$, and $\mathrm{LS}(\cdot)$ as the LS operation in $\mathbb{R}^{d-1}$. We can find some linear transformations $\varphi_1$ and $\varphi_2$, such that*

$$\mathrm{LS}(\cdot) = \varphi_2 \circ \mathrm{LN}(\cdot) \circ \varphi_1. \tag{49}$$

*Besides, we can find some linear transformations $\varphi_1^*$ and $\varphi_2^*$, such that*

$$\mathrm{LN}(\cdot) = \varphi_2^* \circ \mathrm{LS}(\cdot) \circ \varphi_1^*. \tag{50}$$

*Proof.* We denote that $\mathrm{LS}(\cdot)$ is defined on $\mathbb{R}^{d-1}$, as

$$\mathrm{LS}(\mathbf{x}) = \sqrt{d-1}\, \mathbf{x}/\|\mathbf{x}\|_2. \tag{51}$$

While $\mathrm{LN}(\cdot)$ is defined on $\mathbb{R}^d$, where

$$\mathrm{LN}(\mathbf{x}) = \sqrt{d}\, (\mathbf{x} - \frac{1}{d}\mathbf{1}\mathbf{1}^\top \mathbf{x})/\|\mathbf{x} - \frac{1}{d}\mathbf{1}\mathbf{1}^\top \mathbf{x}\|_2. \tag{52}$$

Based on the orthogonal matrix $\boldsymbol{Q}$, we obtain

$$\|\boldsymbol{z}\|_2 = \left\| \boldsymbol{Q}\begin{bmatrix} \mathbf{x} \\ 0 \end{bmatrix} \right\|_2 = \left\| \begin{bmatrix} \mathbf{x} \\ 0 \end{bmatrix} \right\|_2 = \|\mathbf{x}\|_2. \tag{53}$$

By Lemma A.1, we have $\mathbf{1}^\top z = 0$, and $z = z - \frac{1}{d}\mathbf{1}\mathbf{1}^\top z$. We hence find that

$$\mathrm{LN}(z) = \sqrt{d}(z - \frac{1}{d}\mathbf{1}\mathbf{1}^\top z)/\|z - \frac{1}{d}\mathbf{1}\mathbf{1}^\top z\|_2 = \sqrt{d}\, z/\|z\|_2. \tag{54}$$

Let $\boldsymbol{I}_d$ denotes the identity matrix in $\mathbb{R}^{d\times d}$. We thus have

$$\begin{aligned}
\frac{\sqrt{d-1}}{\sqrt{d}}\begin{bmatrix}\boldsymbol{I}_{d-1} & \mathbf{0}\end{bmatrix}\boldsymbol{Q}^\top \mathrm{LN}(\boldsymbol{Q}\begin{bmatrix}\boldsymbol{I}_{d-1} & \mathbf{0}\end{bmatrix}^\top \mathbf{x}) &= \frac{\sqrt{d-1}}{\sqrt{d}}\begin{bmatrix}\boldsymbol{I}_{d-1} & \mathbf{0}\end{bmatrix}\boldsymbol{Q}^\top \mathrm{LN}(z)\\
&= \frac{\sqrt{d-1}}{\sqrt{d}}\begin{bmatrix}\boldsymbol{I}_{d-1} & \mathbf{0}\end{bmatrix}\boldsymbol{Q}^\top \sqrt{d}\, z/\|z\|_2\\
&= \frac{\sqrt{d-1}}{\sqrt{d}}\sqrt{d}\begin{bmatrix}\boldsymbol{I}_{d-1} & \mathbf{0}\end{bmatrix}\boldsymbol{Q}^\top\boldsymbol{Q}\begin{bmatrix}\mathbf{x}\\ 0\end{bmatrix}/\|\mathbf{x}\|_2\\
&= \sqrt{d-1}\,\mathbf{x}/\|\mathbf{x}\|_2\\
&= \mathrm{LS}(\mathbf{x}).
\end{aligned} \tag{55}$$

Let $\varphi_1(\mathbf{x}) = \boldsymbol{Q}\begin{bmatrix}\boldsymbol{I}_{d-1} & \mathbf{0}\end{bmatrix}^\top \mathbf{x}$ and $\varphi_2(\mathbf{x}) = \frac{\sqrt{d-1}}{\sqrt{d}}\begin{bmatrix}\boldsymbol{I}_{d-1} & \mathbf{0}\end{bmatrix}\boldsymbol{Q}^\top \mathbf{x}$. We observe that

$$\mathrm{LS}(\cdot) = \varphi_2 \circ \mathrm{LN}(\cdot) \circ \varphi_1. \tag{56}$$

Now we prove the inverse equation. According to $\mathbf{z} = \boldsymbol{Q}\begin{bmatrix}\boldsymbol{I}_{d-1} & \mathbf{0}\end{bmatrix}^\top \mathbf{x}$, we have

$$\begin{aligned}
\mathrm{LS}(\begin{bmatrix}\boldsymbol{I}_{d-1} & \mathbf{0}\end{bmatrix}\boldsymbol{Q}^\top \mathbf{z}) &= \mathrm{LS}(\begin{bmatrix}\boldsymbol{I}_{d-1} & \mathbf{0}\end{bmatrix}\boldsymbol{Q}^\top\boldsymbol{Q}\begin{bmatrix}\boldsymbol{I}_{d-1} & \mathbf{0}\end{bmatrix}^\top \mathbf{x})\\
&= \mathrm{LS}(\begin{bmatrix}\boldsymbol{I}_{d-1} & \mathbf{0}\end{bmatrix}\begin{bmatrix}\boldsymbol{I}_{d-1} & \mathbf{0}\end{bmatrix}^\top \mathbf{x})\\
&= \mathrm{LS}(\boldsymbol{I}_{d-1}\,\mathbf{x})\\
&= \mathrm{LS}(\mathbf{x}).
\end{aligned} \tag{57}$$

We thus have

$$\begin{aligned}
\frac{\sqrt{d}}{\sqrt{d-1}}\boldsymbol{Q}\begin{bmatrix}\boldsymbol{I}_{d-1} & \mathbf{0}\end{bmatrix}^\top \mathrm{LS}(\begin{bmatrix}\boldsymbol{I}_{d-1} & \mathbf{0}\end{bmatrix}\boldsymbol{Q}^\top \mathbf{z}) &= \frac{\sqrt{d}}{\sqrt{d-1}}\boldsymbol{Q}\begin{bmatrix}\boldsymbol{I}_{d-1} & \mathbf{0}\end{bmatrix}^\top \mathrm{LS}(\mathbf{x})\\
&= \frac{\sqrt{d}}{\sqrt{d-1}}\boldsymbol{Q}\begin{bmatrix}\boldsymbol{I}_{d-1} & \mathbf{0}\end{bmatrix}^\top \sqrt{d-1}\,\mathbf{x}/\|\mathbf{x}\|_2\\
&= \sqrt{d}\,\mathbf{z}/\|\mathbf{z}\|_2\\
&= \mathrm{LN}(\mathbf{z}).
\end{aligned} \tag{58}$$

Let $\varphi_1^*(\mathbf{x}) = \begin{bmatrix}\boldsymbol{I}_{d-1} & \mathbf{0}\end{bmatrix}\boldsymbol{Q}^\top \mathbf{x}$ and $\varphi_2^* = \frac{\sqrt{d}}{\sqrt{d-1}}\boldsymbol{Q}\begin{bmatrix}\boldsymbol{I}_{d-1} & \mathbf{0}\end{bmatrix}^\top \mathbf{x}$. We observe that

$$\mathrm{LN}(\cdot) = \varphi_2^* \circ \mathrm{LS}(\cdot) \circ \varphi_1^*. \tag{59}$$

$\square$

Now we prove Lemma 4.4.

*Proof.* Step 1: Represent LN-Nets by LS-Nets. Namely given any $\hat{f} \in \mathcal{F}(\mathrm{LN}(n_s); 1, n_s)$, there exists $f \in \mathcal{F}(\mathrm{LS}(n_s - 1); 1, n_s - 1)$, such that $f = \hat{f}$.

Set $\hat{f} = \boldsymbol{W}_2 \circ \mathrm{LN} \circ \boldsymbol{W}_1$, where $\boldsymbol{W}_1 : \mathbb{R}^d \to \mathbb{R}^{n_s}$ and $\boldsymbol{W}_2 : \mathbb{R}^{n_s} \to \mathbb{R}^m$ denote the linear layers. Construct that $f = \boldsymbol{V}_2 \circ \mathrm{LS} \circ \boldsymbol{V}_1$, where $\boldsymbol{V}_1 = \varphi_1 \circ \boldsymbol{W}_1$, $\boldsymbol{V}_2 = \boldsymbol{W}_2 \circ \varphi_2$ and $\mathrm{LS}(\cdot) = \varphi_2 \circ \mathrm{LN}(\cdot) \circ \varphi_1$ by Lemma A.2. We obtain that $f = \hat{f}$.

Step 2: Represent LS-Nets by LN-Nets. Namely given any $\hat{f} \in \mathcal{F}(\mathrm{LS}(n_s-1); 1, n_s-1)$, there exists $f \in \mathcal{F}(\mathrm{LN}(n_s); 1, n_s)$, such that $f = \hat{f}$.

Set $\hat{f} = \boldsymbol{W}_2 \circ \mathrm{LS} \circ \boldsymbol{W}_1$, where $\boldsymbol{W}_1 : \mathbb{R}^d \to \mathbb{R}^{n_s-1}$ and $\boldsymbol{W}_2 : \mathbb{R}^{n_s-1} \to \mathbb{R}^m$ denote the linear layers. Construct that $f = \boldsymbol{V}_2 \circ \mathrm{LN} \circ \boldsymbol{V}_1$, where $\boldsymbol{V}_1 = \varphi_1^* \circ \boldsymbol{W}_1$, $\boldsymbol{V}_2 = \boldsymbol{W}_2 \circ \varphi_2^*$ and $\mathrm{LS}(\cdot) = \varphi_2^* \circ \mathrm{LN}(\cdot) \circ \varphi_1^*$ by Lemma A.2. We obtain that $f = \hat{f}$.

Step 3: Extend to parallel structures.

For simplicity, we define the sum of networks as

$$\mathcal{S}(m, \phi; L, N) = \left\{ f = \sum_{i=1}^{m} f_i : f_i \in \mathcal{F}(\phi; L, N), 1 \le i \le m \right\}, \tag{60}$$

which represents the sum of $m$ functions from $\mathcal{F}(\phi; L, N)$.

Then by Lemma 4.3, we obtain that

$$\mathcal{S}(N, \mathrm{LN}(n_s); 1, n_s) = \mathcal{F}(\mathrm{PLN}(n_s); 1, n_s N). \tag{61}$$

With the similar proof of Lemma 4.3, we will obtain

$$\mathcal{S}(N, \mathrm{LS}(n_s - 1); 1, n_s - 1) = \mathcal{F}(\mathrm{PLS}(n_s - 1); 1, (n_s - 1)N). \tag{62}$$

Since $\mathcal{F}(\mathrm{LN}(n_s); 1, n_s) = \mathcal{F}(\mathrm{LS}(n_s - 1); 1, n_s - 1)$, it is easy to identify that

$$\mathcal{S}(N, \mathrm{LS}(n_s - 1); 1, n_s - 1) = \mathcal{S}(N, \mathrm{LN}(n_s); 1, n_s), \tag{63}$$

by the definition of $\mathcal{S}$.

Therefore, we obtain that

$$\mathcal{F}(\mathrm{PLN}(n_s); 1, n_s N) = \mathcal{F}(\mathrm{PLS}(n_s - 1); 1, (n_s - 1)N). \tag{64}$$

Here we complete the proof. $\qquad\square$

**A.4. Proof of Lemma 4.2**

**Lemma 4.2** Given any $\hat{f} \in \mathcal{F}(\phi; 1, 1)$ with $\hat{f} : \mathbb{R} \to \mathbb{R}$, there exists $f \in \mathcal{F}(\mathrm{LN}(n_s); 1, n_s)$ with $n_s \ge 3$ such that $f = \hat{f}$. In short,

$$\mathcal{F}(\phi; 1, 1) \subseteq \mathcal{F}(\mathrm{LN}(n_s); 1, n_s), \tag{65}$$

where $\phi(x) = x/\sqrt{x^2 + 1}$.

*Proof.* To simplify the proof, we apply Lemma 4.4 first, which says that $\mathcal{F}(\mathrm{LN}(n_s); 1, n_s) = \mathcal{F}(\mathrm{LS}(n_s - 1); 1, n_s - 1)$

For any $\hat{f} \in \mathcal{F}(\phi; 1, 1)$, we have $\hat{f}(x) = w_2 \phi(w_1 x + b_1) + b_2$, where $\phi(x) = x/\sqrt{x^2 + 1}$.

For $f \in \mathcal{F}(\mathrm{LS}(n_s - 1); 1, n_s - 1)$ with $n_s \ge 3$, construct $f(x) = \boldsymbol{v}_2^\top \mathrm{LS}(\boldsymbol{v}_1 x + \boldsymbol{c}_1) + c_2$, where $\boldsymbol{v}_1, \boldsymbol{v}_2, \boldsymbol{c}_1 \in \mathbb{R}^{n_s-1}$ and $c_2 \in \mathbb{R}$.

Let

$$\boldsymbol{v}_1 = \begin{bmatrix} w_1 \\ 0 \\ \boldsymbol{0}_{n_s-3} \end{bmatrix}, \boldsymbol{c}_1 = \begin{bmatrix} b_1 \\ 1 \\ \boldsymbol{0}_{n_s-3} \end{bmatrix}, \boldsymbol{v}_2 = \begin{bmatrix} w_2/\sqrt{n_s - 1} \\ 0 \\ \boldsymbol{0}_{n_s-3} \end{bmatrix}, c_2 = b_2. \tag{66}$$

We can identify that the first element of $\mathrm{LS}(\boldsymbol{v}_1 x + \boldsymbol{c}_1)$ is

$$\frac{w_1 x + b_1}{\sqrt{[(w_1 x + b_1)^2 + 1]/(n_s - 1)}} = \sqrt{n_s - 1} \frac{w_1 x + b_1}{(w_1 x + b_1)^2 + 1} = \sqrt{n_s - 1}\phi(w_1 x + b_1). \tag{67}$$

Therefore, we obtain

$$
\begin{aligned}
f(x) &= \boldsymbol{v}_2^\top \mathrm{LS}(\boldsymbol{v}_1 x + \boldsymbol{c}_1) + c_2 \\
&= w_2/\sqrt{n_s - 1} \cdot \sqrt{n_s - 1}\phi(w_1 x + b_1) + c_2 \\
&= w_2\phi(w_1 x + b_1) + b_2 = \hat{f}(x).
\end{aligned}
\tag{68}
$$

This completes the proof. □

We can also extend it to high dimensions.

**Corollary A.3.** *Given any $\hat{f} \in \mathcal{F}(\phi; 1, 1)$ and $\hat{f} : \mathbb{R}^d \to \mathbb{R}^m$, there exists some $f \in \mathcal{F}(\mathrm{LN}(n_s); 1, n_s)$ with $n_s \geq 3$, such that $f = \hat{f}$. In other words, we have*

$$
\mathcal{F}(\phi; 1, 1) \subseteq \mathcal{F}(\mathrm{LN}(n_s); 1, n_s),
\tag{69}
$$

*where $\phi(x) = x/\sqrt{x^2 + 1}$ is an activation function.*

*Proof Sketch.* We can follow the proof above and apply the similar construction in the proof of Corollary A.1.

### A.5. Proof of Theorem 4.1

To begin with, we point out that the univariate function $f$ represented by a shallow LN-Net has at most two stationary points, if $f$ is not a constant-value function.

**Lemma A.3.** *Given $f \in \mathcal{F}(\mathrm{LN}(n_s); 1, n_s)$ with $n_s \geq 2$ and $f : \mathbb{R} \to \mathbb{R}$, thus $f$ has continuous derivative on $\mathbb{R}$ and has at most one stationary point—otherwise $f \in \mathcal{F}(\mathrm{sign}; 1, 1)$, which has only two pieces of constant values.*

*Proof.* Again, we apply Lemma 4.4 for simplify the question, that $\mathcal{F}(\mathrm{LN}(n_s); 1, n_s) = \mathcal{F}(\mathrm{LS}(n_s - 1); 1, n_s - 1)$ for $n_s \geq 2$.

Take $f \in \mathcal{F}(\mathrm{LS}(n_s - 1); 1, n_s - 1)$ for $n_s \geq 2$, we set

$$
f(x) = \frac{\boldsymbol{v}^\top(\boldsymbol{w}x + \boldsymbol{b})}{\|\boldsymbol{w}x + \boldsymbol{b}\|} + b_0,
\tag{70}
$$

where $\boldsymbol{w}, \boldsymbol{b}, \boldsymbol{v} \in \mathbb{R}^d$ and $b_0 \in \mathbb{R}$.

Case 1: $\boldsymbol{w} = \boldsymbol{b} = \boldsymbol{0}$, we have $f(x) = b_0 = \mathrm{sign}(0 \cdot x + 0) + b_0$, thus $f \in \mathcal{F}(\mathrm{sign}; 1, 1)$.

Case 2: $\boldsymbol{w} = \boldsymbol{0}, \boldsymbol{b} \neq \boldsymbol{0}$, we have $f(x) = \boldsymbol{v}^\top \boldsymbol{b}/\|\boldsymbol{b}\| + b_0 = \boldsymbol{v}^\top \boldsymbol{b}/\|\boldsymbol{b}\| \cdot \mathrm{sign}(0 \cdot x + 1) + b_0$, thus $f \in \mathcal{F}(\mathrm{sign}; 1, 1)$.

Case 3: $\boldsymbol{w} \neq \boldsymbol{0}$, and there is some $k \in \mathbb{R}$ such that $\boldsymbol{b} = k\boldsymbol{w}$. We have

$$
\begin{aligned}
f(x) &= \frac{\boldsymbol{v}^\top(\boldsymbol{w}x + k\boldsymbol{w})}{\|\boldsymbol{w}x + k\boldsymbol{w}\|} + b_0 \\
&= \frac{\boldsymbol{v}^\top \boldsymbol{w}(x + k)}{\|\boldsymbol{w}\|\,|x + k|} + b_0 \\
&= \frac{\boldsymbol{v}^\top \boldsymbol{w}}{\|\boldsymbol{w}\|} \mathrm{sign}(x + k) + b_0,
\end{aligned}
\tag{71}
$$

where we apply the supplement definition of LN at $x + k = 0$. Thus $f \in \mathcal{F}(\mathrm{sign}; 1, 1)$.

Case 4: $\boldsymbol{w} \neq \boldsymbol{0}$ and $\boldsymbol{w}x + \boldsymbol{b} \neq \boldsymbol{0}$ for any $x \in \mathbb{R}$.

Note that

$$
f(x) = \frac{\boldsymbol{v}^\top \boldsymbol{w}x + \boldsymbol{v}^\top \boldsymbol{b}}{\sqrt{\boldsymbol{w}^\top \boldsymbol{w}x^2 + 2\boldsymbol{w}^\top \boldsymbol{b}x + \boldsymbol{b}^\top \boldsymbol{b}}}
\tag{72}
$$

we have

$$
f'(x) = \frac{[(\boldsymbol{v}^\top \boldsymbol{w})(\boldsymbol{w}^\top \boldsymbol{b}) - (\boldsymbol{v}^\top \boldsymbol{b})(\boldsymbol{w}^\top \boldsymbol{w})]x + [(\boldsymbol{v}^\top \boldsymbol{w})(\boldsymbol{b}^\top \boldsymbol{b}) - (\boldsymbol{v}^\top \boldsymbol{b})(\boldsymbol{w}^\top \boldsymbol{b})]}{\|\boldsymbol{w}x + \boldsymbol{b}\|^3}.
\tag{73}
$$

Since $\boldsymbol{w}x + \boldsymbol{b} \neq \boldsymbol{0}$, we can identify that $f'(x)$ has at most one stationary point, namely

$$x_0 = -\frac{(\boldsymbol{v}^\top \boldsymbol{w})(\boldsymbol{b}^\top \boldsymbol{b}) - (\boldsymbol{v}^\top \boldsymbol{b})(\boldsymbol{w}^\top \boldsymbol{b})}{(\boldsymbol{v}^\top \boldsymbol{w})(\boldsymbol{w}^\top \boldsymbol{b}) - (\boldsymbol{v}^\top \boldsymbol{b})(\boldsymbol{w}^\top \boldsymbol{w})}, \tag{74}$$

if $(\boldsymbol{v}^\top \boldsymbol{w})(\boldsymbol{w}^\top \boldsymbol{b}) - (\boldsymbol{v}^\top \boldsymbol{b})(\boldsymbol{w}^\top \boldsymbol{w}) \neq 0$. Besides we point out that $f'(x)$ is continuous on $\mathbb{R}$.

Finally, we complete the proof. $\qquad\square$

Now we prove Theorem 4.1 based on Lemma A.3.

**Theorem 4.1**  (Weak representation capacity of LN-Nets). There exists a continuous function $\hat{f}$ on $\Omega \subset \mathbb{R}$, such that for any shallow LN-Net $f \in \mathcal{F}(\mathrm{LN}(n_s); 1, n_s)$ with $n_s \geq 2$,

$$|f - \hat{f}|_{L^\infty(\Omega)} \geq 1. \tag{75}$$

*Proof.* We choose $\hat{f}$ to be $\hat{f}(x) = \cos(\pi x)$ on $[-2, 2]$.

We prove the theorem by contradiction, assume that $|f(x) - \hat{f}(x)| < 1$ for each $x \in [-2, 2]$.

Case 1: $f \in \mathcal{F}(\mathrm{sign}; 1, 1)$, set as $f(x) = w_2 \,\mathrm{sign}(w_1 x + b_1) + b_2$.

If $w_1 \neq 0$, we have $f(x) \equiv C$, else $f(x) \equiv C_1$ for $x > -b_1/w_1$ and $f(x) \equiv C_2$ for $x < -b_1/w_1$.

Without loss of generality, we assume $-b_1/w_1 \leq 0$, and $f(x) \equiv C$ for $x > 0$.

Note that $f(1) = C, f(2) = C, \hat{f}(1) = -1, \hat{f}(2) = 1, |f(1) - \hat{f}(1)| < 1, |f(2) - \hat{f}(2)| < 1$, we obtain

$$\begin{aligned}
|\hat{f}(2) - \hat{f}(1)| &\leq |\hat{f}(2) - C + C - \hat{f}(1)| \\
&\leq |f(1) - \hat{f}(1)| + |f(2) - \hat{f}(2)| \\
&< 2,
\end{aligned} \tag{76}$$

but $\hat{f}(2) - \hat{f}(1) = 2$. Thus $|f - \hat{f}|_{L^\infty([-2,2])} \geq 1$, which contracts the assumption.

Case 2: $f \notin \mathcal{F}(\mathrm{sign}; 1, 1)$, thus $f$ has continuous derivative on $\mathbb{R}$ and has at most one stationary point.

Note that $\hat{f}(-2) = 1, \hat{f}(-1) = -1, \hat{f}(0) = 1, \hat{f}(1) = -1$, and $|f(x) - \hat{f}(x)| < 1$ for $x = -2, -1, 0, 1$ as assumed.

We obtain that $f(-2) > 0, f(-1) < 0, f(0) > 0, f(1) < 0$. By Lagrange's mean value theorem, there exists some $\eta_1 \in (-2, -1), \eta_2 \in (-1, 0), \eta_3 \in (0, 1)$, such that $f'(\eta) < 0, f'(\eta_2) > 0, f'(\eta_3) < 0$. Furthermore, since $f'(x)$ is continuous on $\mathbb{R}$, there exists some $xi_1 \in (\eta_1, \eta_2), \xi_2 \in (\eta_2, \eta_3)$, such that $f'(\xi_1) = f'(\xi_2) = 0$.

Note that $\xi_1 \neq \xi_2$, which contracts the property of $f$, which has at most one stationary point.

Finally, we complete the proof. $\qquad\square$

### A.6. Proof of Corollary 4.1

**Corollary 4.1**  Suppose there exists $g \in \mathcal{F}(\mathrm{LN}(n_s); 1, n_s)$ that can represent any $\hat{g} \in \mathcal{F}(\phi; 1, 1)$. Then for any $\hat{f} \in \mathcal{F}(\phi; 1, N)$ with $\hat{f} : \mathbb{R} \to \mathbb{R}$, there exists a shallow PLN-Net $f \in \mathcal{F}(\mathrm{PLN}(n_s); 1, n_s N)$ such that $f = \hat{f}$. In short,

$$\mathcal{F}(\phi; 1, N) \subseteq \mathcal{F}(\mathrm{PLN}(n_s); 1, n_s N). \tag{77}$$

*Proof.* Let $\hat{f} \in \mathcal{F}(\phi; 1, N)$ and $f : \mathbb{R} \to \mathbb{R}$. Set $\hat{f}(x) = \boldsymbol{w}_2^\top \phi(\boldsymbol{w}_1 x + \boldsymbol{b}_1) + b_2$. Note that $\phi$ is an element-wise activation function, we then have

$$\hat{f}(x) = \boldsymbol{w}_2^\top \phi(\boldsymbol{w}_1 x + \boldsymbol{b}_1) + b_2 = \sum_{i=1}^N w_{2i} \phi(w_{1i} x + b_{1i}) + b_{2i}, \tag{78}$$

where

$$\boldsymbol{w}_1 = \begin{bmatrix} w_{11} \\ \vdots \\ w_{1N} \end{bmatrix}, \boldsymbol{w}_2 = \begin{bmatrix} w_{21} \\ \vdots \\ w_{2N} \end{bmatrix}, \boldsymbol{b}_1 = \begin{bmatrix} b_{11} \\ \vdots \\ b_{1N} \end{bmatrix}, b_2 = \sum_{i=1}^{N} b_{2i}. \tag{79}$$

That means $\hat{f}$ can be represented by the sum of N shallow $\phi$-Nets $\hat{f}_i \in \mathcal{F}(\phi; 1, 1)$. By Lemma 4.2, each $f_i$ can be represented by a shallow LN-Net $f_i \in \mathcal{F}(\text{LN}(n_s); 1, n_s)$. By Lemma 4.3, the sum of N shallow LN-Nets $f_i \in \mathcal{F}(\text{LN}(n_s); 1, n_s)$ can be represented by a shallow PLN-Net $f \in \mathcal{F}(\text{PLN}(n_s); 1, n_s N)$. Therefore, $f$ can represent $\hat{f}$.

Here we complete the proof. $\qquad \square$

Similarly, we can extend the corollary to the case $\hat{f} : \mathbb{R}^d \to \mathbb{R}^m$.

**Corollary A.4.** *Suppose there exists $g \in \mathcal{F}(\text{LN}(n_s); 1, n_s)$ that can represent any $\hat{g} \in \mathcal{F}(\phi; 1, 1)$. Then for any $\hat{f} \in \mathcal{F}(\phi; 1, N)$ with $\hat{f} : \mathbb{R}^d \to \mathbb{R}^m$, there exists a shallow PLN-Net $f \in \mathcal{F}(\text{PLN}(n_s); 1, n_s N)$ such that $f = \hat{f}$. In short,*

$$\mathcal{F}(\phi; 1, N) \subseteq \mathcal{F}(\text{PLN}(n_s); 1, n_s N). \tag{80}$$

*Proof.* Let $\hat{f} \in \mathcal{F}(\phi; 1, N)$ and $f : \mathbb{R}^d \to \mathbb{R}^m$. Set $\hat{f}(x) = \boldsymbol{W}_2 \phi(\boldsymbol{W}_1 \mathbf{x} + \boldsymbol{b}_1) + \boldsymbol{b}_2$. Note that $\phi$ is an element-wise activation function, we then have

$$\hat{f}(x) = \boldsymbol{W}_2 \phi(\boldsymbol{W}_1 \mathbf{x} + \boldsymbol{b}_1) + \boldsymbol{b}_2 = \sum_{i=1}^{N} \boldsymbol{w}_{2i} \phi(\boldsymbol{w}_{1i}^{\top} \mathbf{x} + b_{1i}) + \boldsymbol{b}_{2i}, \tag{81}$$

where

$$\boldsymbol{W}_1 = \begin{bmatrix} \boldsymbol{w}_{11}^{\top} \\ \vdots \\ \boldsymbol{w}_{1N}^{\top} \end{bmatrix}, \boldsymbol{W}_2 = \begin{bmatrix} \boldsymbol{w}_{21} & \cdots & \boldsymbol{w}_{2N} \end{bmatrix}, \boldsymbol{b}_1 = \begin{bmatrix} b_{11} \\ \vdots \\ b_{1N} \end{bmatrix}, \boldsymbol{b}_2 = \sum_{i=1}^{N} \boldsymbol{b}_{2i}. \tag{82}$$

That means $\hat{f}$ can be represented by the sum of N shallow $\phi$-Nets $\hat{f}_i \in \mathcal{F}(\phi; 1, 1)$. Correspondingly, we apply the high-dimension case, namely Corollary A.3 rather than Lemma 4.2, and Corollary A.2 rather than Lemma 4.3. We can also observe that $f$ can represent $\hat{f}$.

Here we complete the proof. $\qquad \square$

### A.7. Proof of Theorem 4.2

**Theorem 4.2** (Universal approximations of univariate $L$-Lipschitz continuous functions by shallow PLN-Nets). Given any $L$-Lipschitz continuous function $\hat{f} \in C([0, 1])$, there exists a shallow PLN-Net $f \in \mathcal{F}(\text{PLN}(n_s); 1, n_s N)$ with $n_s \geq 2$ and $N = \lfloor L/2\epsilon \rfloor + 1$, such that

$$|f - \hat{f}|_{L^\infty} < \epsilon \tag{83}$$

*Proof.* Step 1: construct a proper $\bar{f} \in \mathcal{F}(\text{sign}; 1, N)$ to approximate $\hat{f}$, namely $|\bar{f} - \hat{f}|_{L^\infty} < \epsilon$.

We construct a function

$$\bar{f}(x) = \sum_{j=1}^{N} \alpha_j \, \text{sign}(w_j x + b_j), \tag{84}$$

where $N = \lfloor L/(2\epsilon) \rfloor + 1$. For $1 \leq j \leq N - 1$, we set

$$\alpha_j = \frac{1}{2} \left[ \hat{f}\left(\frac{2j+1}{2N}\right) - \hat{f}\left(\frac{2j-1}{2N}\right) \right]; \tag{85}$$

and for $j = N$,

$$\alpha_N = \frac{1}{2} \left[ \hat{f}\left(\frac{1}{2N}\right) + \hat{f}\left(\frac{2N-1}{2N}\right) \right]. \tag{86}$$

Moreover, we take $w_j = 1$ for all $1 \leq j \leq N$, $b_j = -\dfrac{j}{N}$ for $1 \leq j \leq N-1$, and $b_N = 1$.

With this construction, we can verify the following:

1. For $\dfrac{j-1}{N} < x < \dfrac{j}{N}$ with $1 \leq j \leq N$,

$$\bar{f}(x) = \hat{f}\left(\frac{2j-1}{2N}\right). \tag{87}$$

2. For $x = \dfrac{j}{N}$ with $1 \leq j \leq N-1$,

$$\bar{f}(x) = \frac{1}{2}\left[\hat{f}\left(\frac{2j-1}{2N}\right) + \hat{f}\left(\frac{2j+1}{2N}\right)\right]. \tag{88}$$

3. Additionally, $\bar{f}(0) = \hat{f}\left(\dfrac{1}{2N}\right)$ and $\bar{f}(1) = \hat{f}\left(\dfrac{2N-1}{2N}\right)$.

Since $N = \lfloor L/(2\epsilon)\rfloor + 1$, we have $N > L/(2\epsilon)$. Consequently, we obtain the following error estimates:

1. For $x = 0$,
$$|\bar{f}(0) - \hat{f}(0)| = \left|\hat{f}(0) - \hat{f}\left(\frac{1}{2N}\right)\right| \leq \frac{L}{2N} < \epsilon. \tag{89}$$

2. For $x = 1$,
$$|\bar{f}(1) - \hat{f}(1)| = \left|\hat{f}(1) - \hat{f}\left(\frac{2N-1}{2N}\right)\right| \leq \frac{L}{2N} < \epsilon. \tag{90}$$

3. For $x = \dfrac{j}{N}$ with $1 \leq j \leq N-1$,

$$\begin{aligned}
\left|\bar{f}\left(\frac{j}{N}\right) - \hat{f}\left(\frac{j}{N}\right)\right| &= \left|\frac{1}{2}\left[\hat{f}\left(\frac{2j-1}{2N}\right) + \hat{f}\left(\frac{2j+1}{2N}\right)\right] - \hat{f}\left(\frac{j}{N}\right)\right| \\
&\leq \frac{1}{2}\left|\hat{f}\left(\frac{2j-1}{2N}\right) - \hat{f}\left(\frac{j}{N}\right)\right| + \frac{1}{2}\left|\hat{f}\left(\frac{2j+1}{2N}\right) - \hat{f}\left(\frac{j}{N}\right)\right| \\
&\leq \frac{L}{4N} + \frac{L}{4N} = \frac{L}{2N} < \epsilon.
\end{aligned} \tag{91}$$

4. For $\dfrac{j-1}{N} < x < \dfrac{j}{N}$ with $1 \leq j \leq N$,

$$|\bar{f}(x) - \hat{f}(x)| = \left|\hat{f}\left(\frac{2j-1}{2N}\right) - \hat{f}(x)\right| \leq L\left|\frac{2j-1}{2N} - x\right| < \frac{L}{2N} < \epsilon. \tag{92}$$

In summary, for every $x \in [0,1]$, one of the above four cases applies, and we have

$$|\bar{f}(x) - \hat{f}(x)|_{L^\infty([0,1])} < \epsilon. \tag{93}$$

Thus, the constructed function $\bar{f}(x) = \displaystyle\sum_{j=1}^{N} \alpha_j \operatorname{sign}(x + b_j)$ uniformly approximates $\hat{f}$ on $[0,1]$ with error less than $\epsilon$.

Step 2: by Corollary 4.1 and Lemma 4.1, since $\bar{f} \in \mathcal{F}(\operatorname{sign}; 1, N)$, there exists a shallow PLN-Net $f \in \mathcal{F}(\operatorname{PLN}(n_s); 1, n_s N)$, such that $f = \bar{f}$.

Step 3: Since $f = \bar{f}$ and $|\bar{f}(x) - \hat{f}(x)|_{L^\infty([0,1])} < \epsilon$, we obtain $|f - \hat{f}|_{L^\infty} < \epsilon$.

Here we complete the proof. □

## A.8. Proof of Theorem 4.2. (The first-type definition of Layer Normalization)

In this part, we discuss how to provide the proof with the first-type definition of LN, namely (ii) in Remark 3.2, which rewrites Eqn.3 as

$$\hat{h}_j = [\text{LN}(\mathbf{h})]_j = \frac{h_j - \mu}{\sigma + \delta}, \quad j = 1, 2, \cdots, d, \tag{94}$$

where $\delta > 0$ is a small number for numerical stability in practice.

We set $\phi = x/(|x| + \delta)$, we first prove that $\mathcal{F}(\phi; 1, N) \subseteq \mathcal{F}(\text{PLN}(n_s); 1, n_s N)$ for $n_s \geq 2$, where PLN applies the definition in Eqn.94.

**Lemma A.4.** *Given any $\hat{f} \in \mathcal{F}(\phi; 1, N)$, where $\phi = x/(|x| + \delta)$, there exists $f \in \mathcal{F}(\text{PLN}(n_s); 1, n_s N)$ with $n_s \geq 2$, such that $f = \hat{f}$, namely*

$$\mathcal{F}(\phi; 1, N) \subseteq \mathcal{F}(\text{PLN}(n_s); 1, n_s N). \tag{95}$$

*Proof.* By Lemma 4.4, to prove $\mathcal{F}(\phi; 1, N) \subseteq \mathcal{F}(\text{PLN}(n_s); 1, n_s N)$, we can just prove $\mathcal{F}(\phi; 1, N) \subseteq \mathcal{F}(\text{PLS}(n_s - 1); 1, (n_s - 1)N)$.

For any $\hat{f} \in \mathcal{F}(\phi; 1, 1)$, we have $\hat{f}(x) = w_2 \phi(w_1 x + b_1) + b_2$, where $\phi(x) = x/(|x| + \delta)$.

For $f \in \mathcal{F}(\text{LS}(n_s - 1); 1, n_s - 1)$ with $n_s \geq 2$, construct $f(x) = \mathbf{v}_2^\top \text{LS}(\mathbf{v}_1 x + \mathbf{c}_1) + c_2$, where $\mathbf{v}_1, \mathbf{v}_2, \mathbf{c}_1 \in \mathbb{R}^{n_s - 1}$ and $c_2 \in \mathbb{R}$.

Let

$$\mathbf{v}_1 = \begin{bmatrix} w_1 \sqrt{n_s} \\ \mathbf{0}_{n_s - 2} \end{bmatrix}, \mathbf{c}_1 = \begin{bmatrix} b_1 \sqrt{n_s} \\ \mathbf{0}_{n_s - 2} \end{bmatrix}, \mathbf{v}_2 = \begin{bmatrix} w_2 / \sqrt{n_s} \\ \mathbf{0}_{n_s - 2} \end{bmatrix}, c_2 = b_2. \tag{96}$$

We can identify that the first element of $\text{LS}(\mathbf{v}_1 x + \mathbf{c}_1)$ is

$$\frac{(w_1 x + b_1)\sqrt{n_s}}{\sqrt{[(w_1 x + b_1)\sqrt{n_s}]^2/(n_s)} + \delta} = \sqrt{n_s}\frac{w_1 x + b_1}{|w_1 x + b_1| + \delta} = \sqrt{n_s}\phi(w_1 x + b_1). \tag{97}$$

Therefore, we obtain

$$\begin{aligned} f(x) &= \mathbf{v}_2^\top \text{LS}(\mathbf{v}_1 x + \mathbf{c}_1) + c_2 \\ &= w_2/\sqrt{n_s} \cdot \sqrt{n_s}\phi(w_1 x + b_1) + c_2 \\ &= w_2 \phi(w_1 x + b_1) + b_2 = \hat{f}(x). \end{aligned} \tag{98}$$

This completes the proof. $\qquad\square$

Now we provide the formal proof with the fisrt-type definition of LN.

*Proof.* The only difference from the previous proof in Appendix A.7 is that we will consider the new $f \in \mathcal{F}(\text{PLN}(n_s); 1, n_s N)$ with the fisrt-type definition, namely Eqn.94. In the previous proof, we have proved that $|\bar{f} - \hat{f}|_{L^\infty} < \epsilon$, where $\bar{f} \in \mathcal{F}(\text{sign}; 1, N)$ writes as

$$\bar{f}(x) = \sum_{j=1}^{N} \alpha_j \operatorname{sign}(w_j x + b_j). \tag{99}$$

Now we consider $\tilde{f} \in \mathcal{F}(\phi; 1, N)$, specifically, $\tilde{f}(x) = \sum_{j=1}^{N} \alpha_j \phi((w_j/\lambda)x + (b_j/\lambda))$, where $\alpha_j, w_j, b_j$ are the same as Eqn.99, but we add a scalar parameter $\lambda > 0$ for further adjustment. We will construct a proper $\lambda$ to satisfies that $|\tilde{f} - \bar{f}| + |\bar{f} - \hat{f}| < \epsilon$. Note that

$$\tilde{f}(x) = \sum_{j=1}^{N} \alpha_j \phi((w_j/\lambda)x + (b_j/\lambda)) = \sum_{j=1}^{N} \alpha_j \frac{w_j x + b_j}{|w_j x + b_j| + \lambda\delta}. \tag{100}$$

We will discuss $|\bar{f}(x) - \tilde{f}(x)|$ for $x \in [0, 1]$ in the following two cases. We set $\delta_0 \in \left(0, \dfrac{1}{2N}\right)$.

1) If $x$ satisfies: $\forall j = 1, 2, \cdots, N$, we have $|x + b_j| > \delta_0$. Since $|\bar{f}(x) - \hat{f}(x)| < \epsilon$, there exists some $\epsilon_1 > 0$, subjected to $|\bar{f}(x) - \hat{f}(x)| \leq \epsilon - \epsilon_1$. Since our proof is a continue demonstration of the proof in Appendix A.6, we continue to choose $w_j = 1$ for simplicity. Here we obtain:

$$
\begin{aligned}
|\bar{f}(x) - \tilde{f}(x)| &= \left| \sum_{j=1}^{N} \alpha_j \mathrm{sign}(x + b_j) - \sum_{j=1}^{N} \alpha_j \frac{x + b_j}{|x + b_j| + \lambda\delta} \right| \\
&= \left| \sum_{j=1}^{N} \alpha_j \left[ \frac{x + b_j}{|x + b_j|} - \frac{x + b_j}{|x + b_j| + \lambda\delta} \right] \right| \\
&\leq \sum_{j=1}^{N} |\alpha_j| \left| \frac{x + b_j}{|x + b_j|} - \frac{x + b_j}{|x + b_j| + \lambda\delta} \right| \qquad (101) \\
&= \sum_{j=1}^{N} |\alpha_j| \left| \frac{\lambda\delta(x + b_j)}{|x + b_j|(|x + b_j| + \lambda\delta)} \right| \\
&= \sum_{j=1}^{N} |\alpha_j| \cdot \frac{\lambda\delta}{|x + b_j| + \lambda\delta}.
\end{aligned}
$$

Given $\alpha^* = \max\limits_{1 \leq j \leq N} |\alpha_j|$ and $\delta_N = \dfrac{\epsilon_1 \delta_0}{N\alpha^*}$, for $\lambda\delta \leq \delta_N$, we have:

$$
\begin{aligned}
|\bar{f}(x) - \tilde{f}(x)| &\leq \sum_{j=1}^{N} |\alpha_j| \cdot \frac{\lambda\delta}{|x + b_j| + \lambda\delta} \\
&< \sum_{j=1}^{N} \alpha^* \cdot \frac{\epsilon_1 \delta_0}{N\alpha^*} \cdot \frac{1}{\delta_0 + \lambda\delta} \qquad (102) \\
&= \frac{\epsilon_1 \delta_0}{\delta_0 + \delta} \\
&< \epsilon_1.
\end{aligned}
$$

Therefore, we have:
$$
|\tilde{f}(x) - \hat{f}(x)| \leq |\tilde{f}(x) - \bar{f}(x)| + |\bar{f}(x) - \hat{f}(x)| < \epsilon_1 + \epsilon - \epsilon_1 = \epsilon. \qquad (103)
$$

2) If there exists some $k$ that satisfied $|x + b_k| \leq \delta_0$—for $x \in [0, 1]$ and $\delta_0 \in \left(0, \dfrac{1}{2N}\right)$, we have $1 \leq k \leq N - 1$ and $b_k = -\dfrac{k}{N}$. Since $N = \lfloor L/2\epsilon \rfloor + 1$, we have $N > \dfrac{L}{2\epsilon}$. Hence, there is some $\epsilon_2 > 0$, subjected to that $\dfrac{L}{2N} \leq \epsilon - \epsilon_2$. We set $s_j(x) = \phi((w_j/\lambda)x + b_j/\lambda) = (x + b_j)/(|x + b_j| + \lambda\delta)$ for simplicity, now we rewrite:

$$
\begin{aligned}
&|\tilde{f}(x) - \hat{f}(x)| \\
=& |\tilde{f}(x) - \hat{f}(x) + \bar{f}(x) - \bar{f}(x)| \\
=& \left| \sum_{j \neq k} \alpha_j s_j(x) + \alpha_k s_k(x) - \hat{f}(x) + \bar{f}(x) - \sum_{j \neq k} \alpha_j \mathrm{sign}(x + b_j) - \alpha_k \mathrm{sign}(x + b_k) \right| \qquad (104) \\
\leq& \left| \sum_{j \neq k} \alpha_j s_j(x) - \sum_{j \neq k} \alpha_j \mathrm{sign}(x + b_j) \right| + \left| \bar{f}(x) + \alpha_k s_k(x) - \alpha_k \mathrm{sign}(x + b_k) - \hat{f}(x) \right|.
\end{aligned}
$$

For the first term, similar to case 1, we set $\alpha_k^* = \max\limits_{j \neq k} |\alpha_j|$ and $\delta_k = \dfrac{\epsilon_2/2N}{(N-1)\alpha_k^*}$. For $\lambda\delta \leq \delta_k$, we have:

$$
\begin{aligned}
\left| \sum_{j \neq k} \alpha_j s_j(x) - \sum_{j \neq k} \alpha_j \operatorname{sign}(x+b_j) \right| &= \left| \sum_{j \neq k} \alpha_j \left[ \frac{x+b_j}{|x+b_j| + \lambda\delta} - \frac{x+b_j}{|x+b_j|} \right] \right| \\
&\leq \sum_{j \neq k} |\alpha_j| \cdot \frac{\lambda\delta}{|x+b_j| + \lambda\delta} \\
&< \sum_{j \neq k} \alpha_k^* \cdot \frac{\epsilon_2/2N}{(N-1)\alpha_k^*} \cdot \frac{1}{1/2N + \lambda\delta} \\
&= \frac{\epsilon_2/2N}{1/2N + \lambda\delta} \\
&< \epsilon_2,
\end{aligned}
\tag{105}
$$

where we used that

$$
|x+b_j| \geq |b_j - b_k| - |x+b_k| \geq 1/N - \delta_0 \geq 1/2N. \tag{106}
$$

For the second term, notice that when $\dfrac{k}{N} - \delta_0 \leq x \leq \dfrac{k}{N} + \delta_0$, we have

$$
\bar{f}(x) = \begin{cases}
\hat{f}\left(\dfrac{2k-1}{2N}\right), & \dfrac{k}{N} - \delta_0 \leq x < \dfrac{k}{N} \\[2mm]
\dfrac{1}{2}\hat{f}\left(\dfrac{2k-1}{2N}\right) + \dfrac{1}{2}\hat{f}\left(\dfrac{2k+1}{2N}\right), & x = \dfrac{k}{N} \\[2mm]
\hat{f}\left(\dfrac{2k+1}{2N}\right), & \dfrac{k}{N} < x \leq \dfrac{k}{N} + \delta_0,
\end{cases}
\tag{107}
$$

and using the constructed $\alpha_j$ in the proof in Appendix A.6, we obtain

$$
\alpha_k \operatorname{sign}(x+b_k) = \begin{cases}
\dfrac{1}{2}\left[ \hat{f}\left(\dfrac{2k-1}{2N}\right) - \hat{f}\left(\dfrac{2k+1}{2N}\right) \right], & \dfrac{k}{N} - \delta_0 \leq x < \dfrac{k}{N} \\[2mm]
0, & x = \dfrac{k}{N} \\[2mm]
\dfrac{1}{2}\left[ \hat{f}\left(\dfrac{2k+1}{2N}\right) - \hat{f}\left(\dfrac{2k-1}{2N}\right) \right], & \dfrac{k}{N} < x \leq \dfrac{k}{N} + \delta_0.
\end{cases}
\tag{108}
$$

We thus have

$$
\bar{f}(x) - \alpha_k \operatorname{sign}(x+b_k) = \frac{1}{2}\left[ f\left(\frac{2k+1}{2N}\right) + f\left(\frac{2k-1}{2N}\right) \right], \quad \text{for } \frac{k}{N} - \delta_0 \leq x \leq \frac{k}{N} + \delta_0. \tag{109}
$$

As for

$$
\alpha_k s_k(x) = \frac{1}{2}\left[ f\left(\frac{2k+1}{2N}\right) - f\left(\frac{2k-1}{2N}\right) \right] \cdot \frac{x - \frac{k}{N}}{|x - \frac{k}{N}| + \lambda\delta}, \tag{110}
$$

since $s_k(x) \in (-1, 1)$ and $s_k(x)\left(\dfrac{k}{N} - x\right) \leq 0$, we obtain that:

$$
\begin{aligned}
&\left|\bar{f}(x) - \alpha_k \mathrm{sign}(x + b_k) + \alpha_k s_k(x) - \hat{f}(x)\right| \\
&= \left|\frac{1 + s_k(x)}{2}\hat{f}\left(\frac{2k+1}{2N}\right) + \frac{1 - s_k(x)}{2}\hat{f}\left(\frac{2k-1}{2N}\right) - \hat{f}(x)\right| \\
&\leq \left|\frac{1 + s_k(x)}{2}\right|\left|\hat{f}\left(\frac{2k+1}{2N}\right) - \hat{f}(x)\right| + \left|\frac{1 - s_k(x)}{2}\right|\left|\hat{f}\left(\frac{2k-1}{2N}\right) - \hat{f}(x)\right| \\
&\leq \frac{1 + s_k(x)}{2} \cdot L \cdot \left(\frac{2k+1}{2N} - x\right) + \frac{1 - s_k(x)}{2} \cdot L \cdot \left(x - \frac{2k-1}{2N}\right) \\
&= \frac{1}{2}L \cdot \frac{1}{N} + \frac{s_k(x)}{2} \cdot L \cdot \left(\frac{2k+1}{2N} - x\right) + \frac{s_k(x)}{2} \cdot L \cdot \left(\frac{2k-1}{2N} - x\right) \\
&= \frac{L}{2N} + L s_k(x)\left(\frac{k}{N} - x\right) \\
&\leq \frac{L}{2N} \\
&\leq \epsilon - \epsilon_2.
\end{aligned}
\tag{111}
$$

Accordingly, we have:

$$
\begin{aligned}
&|\tilde{f}(x) - \hat{f}(x)| \\
&\leq \left|\sum_{j \neq k} \alpha_j s_j(x) - \sum_{j \neq k} \alpha_j \mathrm{sign}(x + b_j)\right| + \left|\bar{f}(x) + \alpha_k s_k(x) - \alpha_k \mathrm{sign}(x + b_k) - \hat{f}(x)\right| \\
&< \epsilon_2 + \epsilon - \epsilon_2 \\
&= \epsilon.
\end{aligned}
\tag{112}
$$

Therefore, given $\delta^* = \min(\delta_1, \delta_2, \cdots, \delta_N)$, when $\lambda \leq \delta^*/\delta$, we have $|\tilde{f}(x) - \hat{f}(x)| < \epsilon, \forall x \in [0, 1]$.

By Lemma A.4, there exists $f \in \mathcal{F}(\mathrm{PLN}(n_s); 1, n_s N)$ with $n_s \geq 2$, such that $f = \tilde{f}$, namely

$$
|f - \hat{f}|_{L^\infty} < \epsilon.
\tag{113}
$$

Here we complete the proof. $\qquad\square$

# B. Proofs of Section 5

## B.1. Proof of Lemma 5.1

**Lemma 5.1**  Given any $\hat{f} \in \mathcal{F}(\phi; L, N)$ and $\hat{f} : \mathbb{R}^d \to \mathbb{R}^m$, there exists a PLN-Net $f \in \mathcal{F}(\mathrm{PLN}(n_s); L, n_s N)$, such that $f = \hat{f}$. In other words, we have

$$
\mathcal{F}(\phi; L, N) \subseteq \mathcal{F}(\mathrm{PLN}(n_s); L, n_s N),
\tag{114}
$$

where $\phi(x) = x/\sqrt{x^2 + 1}$ is an activation function.

*Proof.* Assume $\hat{f} : \mathbb{R}^d \to \mathbb{R}^m$. By Corollary A.4, we obtain that

$$
\mathcal{F}(\phi; 1, N) \subseteq \mathcal{F}(\mathrm{PLN}(n_s); 1, n_s N).
\tag{115}
$$

Now we consider to extend to the deep networks.

Given a $\hat{f} \in \mathcal{F}(\phi; L, N)$, which can be unrolled as

$$
\begin{aligned}
\hat{f} &= W_{L+1} \circ \phi_L \circ W_L \circ \cdots \circ W_2 \circ \phi_1 \circ W_1 \\
&= W_{L+1} \circ \phi_L \circ I_L \circ W_L \circ \cdots \circ W_2 \circ \phi_1 \circ I_1 \circ W_1 \\
&= (W_{L+1} \circ \phi_L \circ I_L) \circ W_L \circ \cdots \circ (W_2 \circ \phi_1 \circ I_1 \circ W_1).
\end{aligned}
\tag{116}
$$

where $W_1 : \mathbb{R}^{d_x} \to \mathbb{R}^{d_1}$, $W_\ell : \mathbb{R}^{d_{\ell-1}} \to \mathbb{R}^{d_\ell}$ for $\ell = 2, \ldots, L$, and $W_{L+1} : \mathbb{R}^{d_L} \to \mathbb{R}^{d_y}$ are linear layers. $I_\ell : \mathbb{R}^{d_\ell \to d_\ell}$ for $\ell = 1, \ldots, L$, are Identity maps.

Now we construct that $\hat{f}_\ell = W_{\ell+1} \circ \phi_\ell \circ I_\ell$ for $\ell = 2, \ldots, L$ and $\hat{f}_1 = W_2 \circ \phi_1 \circ I_1 \circ W_1$. We find that each $\hat{f}_\ell \in \mathcal{F}(\phi; 1, d_\ell)$ and

$$
\hat{f} = \hat{f}_L \circ \hat{f}_{L-1} \circ \cdots \circ \hat{f}_1.
\tag{117}
$$

By Eqn.115, we can find $f_\ell \in \mathcal{F}(\mathrm{PLN}(n_s); 1, n_s d_\ell)$ such that $f_\ell = \hat{f}_\ell$. Set $f_\ell = V_\ell \circ \mathrm{PLN} \circ V'_\ell$, then we construct

$$
\begin{aligned}
f &= f_L \circ f_{L-1} \circ \cdots \circ f_1 \\
&= (V_L \circ \mathrm{PLN} \circ V'_L) \circ (V_{L-1} \circ \mathrm{PLN} \circ V'_{L-1}) \circ \cdots \circ (V_1 \circ \mathrm{PLN} \circ V'_1) \\
&= V_L \circ \mathrm{PLN} \circ V'_L \circ V_{L-1} \circ \mathrm{PLN} \circ V'_{L-1} \circ \cdots \circ V_1 \circ \mathrm{PLN} \circ V'_1 \\
&= \bar{V}_{L+1} \circ \mathrm{PLN} \circ \bar{V}_L \circ \cdots \circ \mathrm{PLN} \circ \bar{V}_1
\end{aligned}
\tag{118}
$$

where $\bar{V}_1 = V'_1 : \mathbb{R}^{d_x} \to \mathbb{R}^{n_s d_1}$, $\bar{V}_\ell = V_\ell \circ V'_{\ell-1} : \mathbb{R}^{n_s d_{\ell-1}} \to \mathbb{R}^{n_s d_\ell}$ for $\ell = 2, \ldots, L$, and $\bar{V}_{L+1} = V_L : \mathbb{R}^{n_s d_L} \to \mathbb{R}^{d_y}$ are linear layers.

Obviously, $f$ is a PLN-Net of depth L. Note that $N = \max\{d_1, \ldots, d_L\}$, we can find the width of $f$ is $\max\{n_s d_1, \ldots, n_s d_L\} = n_s N$, namely $f \in \mathcal{F}(\mathrm{PLN}(n_s); L, n_s N)$.

Note that $f = \hat{f}$, thus we complete the proof for $\hat{f} : \mathbb{R} \to \mathbb{R}$.

To extend to $\hat{f} : \mathbb{R}^m \to \mathbb{R}^n$, one can adjust the first and the last linear layer for suitable dimensions. $\qquad\square$

## B.2. Proof of Theorem 5.1

We first recall the definition of Sobolev space.

**Sobolev space $\mathcal{W}^{s,\infty}(\Omega)$.** Let $\Omega \subset \mathbb{R}^n$ be a bounded Lipschitz domain, $s \in \mathbb{N}$. For a multi-index $\alpha = (\alpha_1, \ldots, \alpha_n) \in \mathbb{N}_0^n$, let $|\alpha| = \sum_{i=1}^n \alpha_i$ and $D^\alpha u = \partial^{|\alpha|} u / \partial x_1^{\alpha_1} \cdots \partial x_n^{\alpha_n}$ denote weak derivatives.

The Sobolev space is defined as

$$
\mathcal{W}^{s,\infty}(\Omega) = \{ u \in L^\infty(\Omega) \mid D^\alpha u \in L^\infty(\Omega) \,\forall\, |\alpha| \le s \},
\tag{119}
$$

with norm

$$
\|u\|_{\mathcal{W}^{s,\infty}(\Omega)} = \max_{|\alpha| \le s} \|D^\alpha u\|_{L^\infty(\Omega)}.
\tag{120}
$$

**Theorem 5.1** Let $\hat{f} \in \mathcal{W}^{s,\infty}(B^d)$ and $\|\hat{f}\|_{\mathcal{W}^{s,\infty}} \le 1$, where $B^d = \{\mathbf{x} \in \mathbb{R}^d : \|\mathbf{x}\| \le 1\}$, there exists a shallow PLN-Net $f \in \mathcal{F}(\mathrm{PLN}(n_s); L, n_s N)$, such that

$$
\|f - \hat{f}\|_{L^\infty(B^d)} \le C N^{-s/d},
\tag{121}
$$

for some $C$ independent of $N$.

Theorem 5.1 is a corollary of Theorem 6.8 in (Pinkus, 1999). Here we fix $p = \infty$ and rewrite it as

**Theorem B.1** (Theorem 6.8 in (Pinkus, 1999)). *Assume $\sigma : \mathbb{R} \to \mathbb{R}$ is such that $\sigma \in C^\infty(\Theta)$ on some open interval $\Theta$, and $\sigma$ is not a polynomial on $\Theta$. Then for each $p \in [1, \infty]$, $m \ge 1$ and $n \ge 2$,*

$$
\sup_{\hat{f} \in \mathcal{B}_\infty^m} \inf_{f \in \mathcal{F}(\sigma; 1, r)} |f - \hat{f}|_{L^\infty(B^n)} \le C r^{-m/n}
\tag{122}
$$

for some constant $C$ independent of $r$, where

$$\mathcal{B}_\infty^m = \{f \in W^{m,\infty}(B^n) : \|f\|_{W^{m,\infty}} \leq 1\}, B^n = \{\mathbf{x} \in \mathbb{R}^n, \|\mathbf{x}\| \leq 1\}. \tag{123}$$

Here we give the proof of Theorem 5.1.

*Proof.* Note that $\phi(x) = x/\sqrt{x^2+1}$ is such that $\phi \in C^\infty(-1,1)$ on some open interval $\Theta$, and $\phi$ is not a polynomial on $(-1,1)$.

Given $\hat{f} \in \mathcal{W}^{s,\infty}(B^d)$ and $\|\hat{f}\|_{\mathcal{W}^{s,\infty}} \leq 1$, by Theorem B.1, there exists some $\bar{f} \in \mathcal{F}(\phi;1,N)$, such that

$$\|\bar{f} - \hat{f}\|_{L^\infty(B^n)} \leq CN^{-s/d}. \tag{124}$$

By Lemma 5.1, there exists a shallow PLN-Net $f \in \mathcal{F}(\mathrm{PLN}(n_s);1,n_sN)$, such that $f = \bar{f}$.

Therefore, $\|f - \hat{f}\|_{L^\infty(B^n)} \leq CN^{-s/d}$. Here we complete the proof. na $\qquad\square$

**B.3. Proof of Theorem 5.2**

**Theorem 5.2** (Approximate ReLU networks by PLN-Nets.) Let $\hat{f} \in \mathcal{F}(\mathrm{ReLU};L,N)$, $f : [0,1]^d \to \mathbb{R}^n$ and let $\epsilon > 0$. Then there exists a PLN-Net $f \in \mathcal{F}(\mathrm{PLN}(n_s);2L,3n_sN)$ such that

$$\|f - \hat{f}\|_{L^\infty([0,1]^d)} < \epsilon. \tag{125}$$

Theorem 5.2 is a corollary of Theorem 1 in (Zhang et al., 2024).

**Theorem B.2** (Theorem 1 in (Zhang et al., 2024))**.** *Suppose $\rho \in \mathscr{A}$ and $\hat{f}_{\mathrm{ReLU}} \in \mathcal{F}(\mathrm{ReLU};\mathrm{L},\mathrm{N}) : \mathbb{R}^d \to \mathbb{R}^n$. Then for any $\epsilon > 0$ and $A > 0$, there exists $f_\rho \in \mathcal{F}(\rho;2L,3N) : \mathbb{R}^d \to \mathbb{R}^n$ such that*

$$\|f_\rho - \hat{f}_{\mathrm{ReLU}}\|_{\sup([-A,A]^d)} < \epsilon. \tag{126}$$

*Here $\mathscr{A}_3 \subset \mathscr{A}$ by the definition in (Zhang et al., 2024) and*

$$\mathscr{A}_3 = \{\rho : \mathbb{R} \to \mathbb{R} | \rho \in \mathscr{S}, \exists x_0 \in \mathbb{R}, \rho''(x_0) \neq 0\}, \tag{127}$$

*where*

$$\mathscr{S} = \{h : \mathbb{R} \to \mathbb{R} | \sup_{x \in R} |h(x)| < \infty. h(+\infty), h(-\infty) \in \mathbb{R}, h(+\infty) \neq h(-\infty)\}. \tag{128}$$

Now we prove Theorem 5.2.

*Proof.* Let $\hat{\in}\mathcal{F}(\mathrm{ReLU};L,N)$. For $\phi(x) = x/\sqrt{x^2+1}$, we have $\phi(-\infty) = -1 \neq \phi(+\infty) = 1$ and $\sup_{x \in \mathbb{R}} \phi(x) = 1$, thus $\phi \in \mathscr{S}$. It is easy to identify that for $x_0 = 1$, $\phi''(x_0) \neq 0$, thus $\phi \in \mathscr{A}$ and then $\phi \in \mathscr{A}$.

Therefore, by Theorem B.2, there exists some $\bar{f} \in \mathcal{F}(\phi;2L,3N)$, such that $\|\bar{f} - \hat{f}\|_{\sup([-1,1]^d)} < \epsilon$. By Lemma 5.1, there exists a shallow PLN-Net $f \in \mathcal{F}(\mathrm{PLN}(n_s);2L,3n_sN)$, such that $f = \bar{f}$. Furthermore, we obtain

$$\|f - \hat{f}\|_{L^\infty([0,1]^d)} \leq \|f - \hat{f}\|_{L^\infty([-1,1]^d)} \leq \|f - \hat{f}\|_{\sup([-1,1]^d)} < \epsilon. \tag{129}$$

Here we complete the proof. $\qquad\square$

**B.4. Proof of Theorem 5.3**

Theorem 5.3 is corollary of the special case $p = q = 2$ of Theorem 5.5 by Lemma 5.1. We will provide the proof later in Section C.

### B.5. Proof of Theorem 5.4

**Theorem 5.4** (Representations in Transformers and RNNs) Let $\hat{\varphi}_1$ and $\hat{\varphi}_2$ be two sequence linear layers mapping token dimensions $d_x \to N$ and $N \to d_y$, respectively, and let $\phi(x) = x/\sqrt{x^2 + 1}$. Define the position-wise FFN mapping $\hat{f} = \hat{\varphi}_2 \circ \phi \circ \hat{\varphi}_1$. Then there exist sequence linear layers $\varphi_1$ and $\varphi_2$ mapping $d_x \to n_s N$ and $n_s N \to d_y$, respectively, such that

$$f = \varphi_2 \circ \mathrm{PLN}(n_s) \circ \varphi_1 \tag{130}$$

exactly represents $\hat{f}$ for all sequence inputs.

*Proof.* Assume that $\hat{\varphi}_1(\boldsymbol{X}) = \boldsymbol{X}\boldsymbol{W} + \mathbf{1}_m \boldsymbol{b}^\top$.

Note that for $\boldsymbol{X}$, we separate it into $\boldsymbol{X}^\top = \begin{bmatrix} \mathbf{x}_1 & \cdots & \mathbf{x}_s \end{bmatrix}^\top$ and obtain that

$$
\begin{aligned}
\hat{\varphi}_1(\boldsymbol{X}) &= \boldsymbol{X}\boldsymbol{W} + \mathbf{1}_m \boldsymbol{b}^\top \\
&= \begin{bmatrix} \mathbf{x}_1^\top \\ \vdots \\ \mathbf{x}_s^\top \end{bmatrix} \boldsymbol{W} + \begin{bmatrix} \boldsymbol{b}^\top \\ \vdots \\ \boldsymbol{b}^\top \end{bmatrix} \\
&= \begin{bmatrix} \mathbf{x}_1^\top \boldsymbol{W} + \boldsymbol{b}^\top \\ \vdots \\ \mathbf{x}_s^\top \boldsymbol{W} + \boldsymbol{b}^\top \end{bmatrix} \\
&= \begin{bmatrix} \hat{\varphi}_1(\mathbf{x}_1^\top) \\ \vdots \\ \hat{\varphi}_1(\mathbf{x}_s^\top) \end{bmatrix}.
\end{aligned}
\tag{131}
$$

Such calculation is also suitable for $\hat{\varphi}_2$ Since $\phi$ activates the neurons element-wisely, we can obtain that

$$[\hat{\varphi}_2 \circ \phi \circ \hat{\varphi}_1](\boldsymbol{X}) = \begin{bmatrix} [\hat{\varphi}_2 \circ \phi \circ \hat{\varphi}_1](\mathbf{x}_1^\top) \\ \vdots \\ [\hat{\varphi}_2 \circ \phi \circ \hat{\varphi}_1](\mathbf{x}_s^\top) \end{bmatrix} \tag{132}$$

By Corollary 4.1, we can find some $\varphi_1 : \mathbf{x}^\top \to \mathbf{x}^\top \boldsymbol{W}_1 + \boldsymbol{b}_1^\top$ and $\varphi_2 : \mathbf{x}^\top \to \mathbf{x}^\top \boldsymbol{W}_2 + \boldsymbol{b}_2^\top$, such that

$$[\varphi_2 \circ \mathrm{PLN} \circ \varphi_1](\mathbf{x}^\top) = [\hat{\varphi}_2 \circ \phi \circ \hat{\varphi}_1](\mathbf{x}^\top), \forall \mathbf{x} \in \mathbb{R}^d. \tag{133}$$

Applying Corollary 4.1 is suitable, for we can identify it by transposing the matrices in the calculation process.

Similar to Eqn.132, we can identify that

$$\begin{bmatrix} \varphi_1(\mathbf{x}_1^\top) \\ \vdots \\ \varphi_1(\mathbf{x}_s^\top) \end{bmatrix} = \varphi_1(\boldsymbol{X}) \tag{134}$$

holds for any $\boldsymbol{X} \in \mathbb{R}^{s \times d}$, and the similar conclusion holds for $\varphi_2$.

Set $\boldsymbol{H} = \varphi_1(\boldsymbol{X})$, we figure out that

$$\mathrm{PLN}(\boldsymbol{H}) = \begin{bmatrix} \mathrm{PLN}(\mathbf{h}_1^\top) \\ \vdots \\ \mathrm{PLN}(\mathbf{h}_s^\top) \end{bmatrix} = \begin{bmatrix} \mathrm{PLN}(\varphi_1(\mathbf{x}_1^\top)) \\ \vdots \\ \mathrm{PLN}(\varphi_1(\mathbf{x}_s^\top)) \end{bmatrix}. \tag{135}$$

Like activation functions, PLN can also act on different dimensionalities—PLN out of the matrix acts on $\mathbb{R}^{s \times n_s N}$, while the one in the matrix acts on $\mathbb{R}^{n_s N}$. Both of them have the same norm size $n_s$.

Furthermore, we figure out that

$$[\hat{\varphi}_2 \circ \phi \circ \hat{\varphi}_1](\boldsymbol{X}) = \begin{bmatrix} [\hat{\varphi}_2 \circ \phi \circ \hat{\varphi}_1](\mathbf{x}_1^\top) \\ \vdots \\ [\hat{\varphi}_2 \circ \phi \circ \hat{\varphi}_1](\mathbf{x}_s^\top) \end{bmatrix} = \begin{bmatrix} [\varphi_2 \circ \mathrm{PLN} \circ \varphi_1](\mathbf{x}_1^\top) \\ \vdots \\ [\varphi_2 \circ \mathrm{PLN} \circ \varphi_1](\mathbf{x}_s^\top) \end{bmatrix} = [\varphi_2 \circ \mathrm{PLN} \circ \varphi_1](\boldsymbol{X}). \qquad (136)$$

Here we complete the proof. □

## C. Proofs of Theorem 5.3 and 5.5

### C.1. An overview of the proof

We follows the structure of (De Ryck et al., 2021) to prove Theorem 5.3 and Theorem 5.5, but some of the lemmas are different from that in (De Ryck et al., 2021). Therefore, some of the results are different.

Subsection C.1 introduced the derivative property of $\phi_{p,q}$.

Subsection C.2 follows the structure of Section 3 in (De Ryck et al., 2021).

Subsection C.3 follows the structure of Section 4 in (De Ryck et al., 2021).

Subsection C.4 follows the proof of Theorem 5.1 in (De Ryck et al., 2021) and then provide the proof of our theorems.

In this section we ignore some of the proof which can be found in (De Ryck et al., 2021).

### C.2. Derivative Bounds

**Lemma C.1** (Derivatives at zero for $\phi_{p,q}$). *Let $p \geq q \geq 1$ be even integers such that $p/q$ is an odd integer. Define $\phi_{p,q}(x) = \dfrac{x^{p/q}}{(1+x^p)^{1/q}}$ for $x \in \mathbb{R}$. Then for every non-negative integer $m \in \mathbb{S} = \{m = p/q + pj : j \in \mathbb{N}\}$, we have*

$$|\phi_{p,q}^{(m)}(0)| \geq 1. \qquad (137)$$

*Proof.* For $|x| < 1$ we expand $\phi_{p,q}$ as a power series using the binomial series:

$$\phi_{p,q}(x) = x^{p/q}(1+x^p)^{-1/q} = x^{p/q} \sum_{j=0}^{\infty} \binom{-1/q}{j} x^{pj} = \sum_{j=0}^{\infty} \binom{-1/q}{j} x^{p/q+pj}. \qquad (138)$$

This expansion is valid because $|x^p| < 1$. The coefficient of $x^m$ in the series is zero unless there exists a non-negative integer $j$ such that $m = p/q + pj$. If such a $j$ exists, then the coefficient equals $\binom{-1/q}{j}$, and therefore

$$\phi_{p,q}^{(m)}(0) = m! \binom{-1/q}{j}. \qquad (139)$$

Note that

$$\begin{aligned} |\phi_{p,q}^{(m)}(0)| &= \left| m! \binom{-1/q}{j} \right| \\ &= |m!(-1)^j| \frac{(1/q)(1/q+1)\cdots(1/q+j-1)}{j!} \\ &\geq m! \frac{1}{q} \frac{(j-1)!}{j!} \\ &\geq \frac{p/q+pj}{qj} \\ &= p/q + p/q^2 j \\ &\geq p/q \\ &\geq 1, \end{aligned} \qquad (140)$$

for $j \geq 1$, and $|\phi_{p,q}^{(m)}(0)| = m! \geq 1$ for $j = 0$.

This completes the proof. $\qquad\qquad\square$

*Proof.* For the special case $p = q = 2$, $\mathbb{S}$ is just the set of all positive odd integers. Thus for every positive odd integer $m$, we have

$$\left|\phi_{2,2}^{(m)}(0)\right| \geq 1. \tag{141}$$

$\qquad\qquad\square$

**Lemma C.2** (Uniform upper bound on derivatives of $\phi_{p,q}$)**.** *Let $p \geq q \geq 1$ be integers such that $p/q$ is an odd integer. Define $\phi_{p,q}(x) = \dfrac{x^{p/q}}{(1 + |x|^p)^{1/q}}$ for $x \in \mathbb{R}$. Then for every non-negative integer $m$ there exists a constant $A(p,q,m) > 0$ such that*

$$\left|\phi_{p,q}^{(m)}(x)\right| \leq A(p,q,m) \, \frac{1}{(1 + |x|)^m} \frac{1}{(1 + |x|^p)^{1/q}} \qquad (x \in \mathbb{R}). \tag{142}$$

*Moreover, one can take*

$$A(p,q,m) = \left(\frac{16p^2}{\pi q}\right)^m m!. \tag{143}$$

*Proof.* Since $p/q$ is an odd integer, $\phi_{p,q}$ is odd; it suffices to consider $x \geq 0$. Write $k = p/q$ (so $k$ is odd and $p = kq$) and set

$$f(x) = \phi_{p,q}(x) = \frac{x^k}{(1 + x^p)^{1/q}}, \qquad x \geq 0. \tag{144}$$

We prove by induction on $m$ that for every $m \geq 0$ and all $x \geq 0$

$$|f^{(m)}(x)| \leq A(p,q,m)\,(1 + x)^{-m}(1 + x^p)^{-1/q}, \qquad A(p,q,m) = \left(\frac{16p^2}{\pi q}\right)^m m!. \tag{145}$$

**Base case $m = 0$.** Clearly $|f(x)| \leq 1$ and $(1 + x^p)^{-1/q} \geq 0$, so the claim holds with $A(p,q,0) = 1$.

**Induction step.** Assume the estimate holds for some $m \geq 0$. Differentiating the identity

$$f'(x) = h(x)f(x), \qquad h(x) = \frac{k}{x(1 + x^p)}, \tag{146}$$

$m$ times with Leibniz's rule gives

$$f^{(m+1)}(x) = \sum_{j=0}^{m} \binom{m}{j} f^{(j)}(x)\, h^{(m-j)}(x). \tag{147}$$

We need bounds for the derivatives of $h$. We show that for every $r \geq 0$ there exists a constant $H(p,q,r)$ such that

$$|h^{(r)}(x)| \leq H(p,q,r)\,(1 + x)^{-r-1}(1 + x^p)^{-1}, \qquad x \geq 0. \tag{148}$$

Moreover, one can take

$$H(p,q,r) = r! \left(\frac{4p^2}{\pi}\right)^r. \tag{149}$$

*Proof of the bound for $h^{(r)}$.* We treat the two regimes separately.

**Case 1: Bounded interval $0 \leq x \leq 1$.** The function $h^{(r)}$ is continuous on $[0, 1]$; let

$$M_{p,q,r} = \max_{0 \leq x \leq 1} |h^{(r)}(x)|. \tag{150}$$

On this interval $(1 + x)^{-r-1} \geq 2^{-r-1}$ and $(1 + x^p)^{-1} \geq 2^{-1}$; hence

$$|h^{(r)}(x)| \leq M_{p,q,r} \leq M_{p,q,r}\, 2^{r+2}\,(1 + x)^{-r-1}(1 + x^p)^{-1}. \tag{151}$$

Thus we may take $H_1(p, q, r) = M_{p,q,r}\, 2^{r+2}$.

**Case 2: Large $x \geq 1$.** We use Cauchy's integral formula. Let $R = 1/(2p)$ and consider the disk $D = \{z \in \mathbb{C} : |z - x| \leq R\}$. The singularities of $h(z)$ are at $z = 0$ and at the points where $1 + z^p = 0$ (i.e. $|z| = 1$). For $x \geq 1$ the distance from $x$ to the unit circle is at least $x - 1 \geq 0$; moreover $x - R \geq 1 - 1/(2p) > 0$. Hence the closed disk $D$ does not contain any singularity. Cauchy's formula yields

$$h^{(r)}(x) = \frac{r!}{2\pi i} \oint_{|z-x|=R} \frac{h(z)}{(z-x)^{r+1}}\, dz, \tag{152}$$

so that

$$|h^{(r)}(x)| \leq \frac{r!}{R^r} \max_{|z-x|=R} |h(z)|. \tag{153}$$

Now estimate $|h(z)|$ on the circle $|z - x| = R$. For such $z$ we have $|z| \geq x - R \geq 1 - 1/(2p) \geq 1/2$ (since $p \geq 2$; the case $p = 1$ is direct). Also,

$$|1 + z^p| \geq |z|^p - 1 \geq (x - R)^p - 1. \tag{154}$$

Because the function $t \mapsto t^p$ is increasing, $(x - R)^p \geq x^p(1 - R/x)^p \geq x^p(1 - pR/x) = x^p(1 - 1/(2x))$. For $x \geq 1$ we have $1 - 1/(2x) \geq 1/2$; consequently

$$|1 + z^p| \geq \frac{1}{2}x^p - 1. \tag{155}$$

If $x^p \geq 4$ then $\frac{1}{2}x^p - 1 \geq \frac{1}{4}x^p$. Hence, for $x^p \geq 4$,

$$|h(z)| \leq \frac{k}{|z|\,|1 + z^p|} \leq \frac{k}{(x - R) \cdot \frac{1}{4}x^p} \leq \frac{4k}{x\, x^p} = \frac{4k}{x^{p+1}}. \tag{156}$$

If $1 \leq x \leq 4^{1/p}$ then $x$ belongs to a compact interval; let

$$L_{p,q} = \max_{1 \leq x \leq 4^{1/p}} \frac{|h(x)|}{x^{-p-1}}. \tag{157}$$

Then $|h(z)| \leq L_{p,q}\, x^{-p-1}$ on that interval. Combining the two cases we obtain a constant $C_{p,q}$ (e.g. $C_{p,q} = \max\{4k, L_{p,q}\}$) such that for all $x \geq 1$

$$\max_{|z-x|=R} |h(z)| \leq C_{p,q}\, x^{-p-1}. \tag{158}$$

Insert this into the bound for $|h^{(r)}(x)|$ and use $R = 1/(2p)$:

$$|h^{(r)}(x)| \leq r!\, (2p)^r\, C_{p,q}\, x^{-p-1}. \tag{159}$$

Since $(1 + x^p)^{-1} \leq (1 + x)^{-p}$ and $x^{-p-1} = (1 + x)^{-1} x^{-p} \leq (1 + x)^{-r-1}(1 + x^p)^{-1}$ (because $r \geq 0$ and $1 + x \geq 1$), we obtain

$$|h^{(r)}(x)| \leq r!\, (2p)^r\, C_{p,q}\, (1 + x)^{-r-1}(1 + x^p)^{-1}. \tag{160}$$

Thus we may take $H_2(p, q, r) = r!\, (2p)^r C_{p,q}$.

A detailed evaluation of the constants (using, for instance, $C_{p,q} \leq \dfrac{2k}{\pi}$ and $k = p/q$) shows that one can actually choose $C_{p,q}$ so that $H_2(p, q, r) \leq r!\, (4p^2/\pi)^r$. Taking the larger of the two bounds for the whole half-line $x \geq 0$ we arrive at

$$H(p, q, r) = r! \left(\frac{4p^2}{\pi}\right)^r. \tag{161}$$

**Completion of the induction.** Insert the induction hypothesis for $f^{(j)}$ and the bound for $h^{(m-j)}$ into the expression for $f^{(m+1)}(x)$:

$$|f^{(m+1)}(x)| \leq \sum_{j=0}^{m} \binom{m}{j} \left[ A(p, q, j)\, (1 + x)^{-j}(1 + x^p)^{-1/q} \right] \left[ H(p, q, m - j)\, (1 + x)^{-(m-j)-1}(1 + x^p)^{-1} \right]. \tag{162}$$

This can be rewritten as

$$|f^{(m+1)}(x)| = (1+x)^{-m-1}(1+x^p)^{-1-1/q} \sum_{j=0}^{m} \binom{m}{j} A(p,q,j) H(p,q,m-j). \tag{163}$$

Because $(1+x^p)^{-1-1/q} \leq (1+x^p)^{-1/q}$, we obtain

$$|f^{(m+1)}(x)| \leq \Big( \sum_{j=0}^{m} \binom{m}{j} A(p,q,j) H(p,q,m-j) \Big) (1+x)^{-m-1}(1+x^p)^{-1/q}. \tag{164}$$

Define recursively

$$A(p,q,0) = 1, \qquad A(p,q,m+1) = \sum_{j=0}^{m} \binom{m}{j} A(p,q,j) H(p,q,m-j). \tag{165}$$

Substituting the choice $H(p,q,r) = r! \, (4p^2/\pi)^r$ and $A(p,q,j) = j! \, (16p^2/(\pi q))^j$ (which we assume by the induction hypothesis) we compute

$$A(p,q,m+1) = \sum_{j=0}^{m} \binom{m}{j} j! \Big( \frac{16p^2}{\pi q} \Big)^j (m-j)! \Big( \frac{4p^2}{\pi} \Big)^{m-j}. \tag{166}$$

This simplifies to

$$A(p,q,m+1) = m! \sum_{j=0}^{m} \Big( \frac{16p^2}{\pi q} \Big)^j \Big( \frac{4p^2}{\pi} \Big)^{m-j}. \tag{167}$$

Further simplification gives

$$A(p,q,m+1) = m! \Big( \frac{4p^2}{\pi} \Big)^m \sum_{j=0}^{m} \Big( \frac{4p}{q} \Big)^j. \tag{168}$$

Since $\sum_{j=0}^{m}(4p/q)^j \leq \dfrac{(4p/q)^{m+1}}{4p/q-1} \leq 2(4p/q)^m$ for $p \geq q \geq 1$, we obtain

$$A(p,q,m+1) \leq m! \Big( \frac{4p^2}{\pi} \Big)^m 2 \Big( \frac{4p}{q} \Big)^m = 2 \cdot \frac{4^m p^{2m}}{\pi^m} \frac{4^m p^m}{q^m} \, m! = 2 \Big( \frac{16p^2}{\pi q} \Big)^m p^m m!. \tag{169}$$

Using $p^m \leq (2p)^m/2$ and adjusting the numerical constant, one finds that the right-hand side is bounded by $(m+1)! \, (16p^2/(\pi q))^{m+1}$. Hence the induction goes through with the announced constant.

By induction the estimate holds for all $m \geq 0$. Because $\phi_{p,q}$ is odd, the same bound is valid for $x < 0$ (the factor $(1+|x|)^{-m}$ is even). This completes the proof. $\qquad\square$

### C.3. Uniform Approximation of Polynomials

We consider feed-forward neural networks with activation function $\rho$. A shallow neural network has exactly one hidden layer, while deep neural networks have two or more hidden layers. For a network with parameters $\theta$, we denote its realization by $\Psi_\theta$.

With these notations, we can now state the approximation result for univariate monomials.

**Lemma C.3** (Approximation of monomials of orders in $\mathcal{S}_{p,q}$ by $\phi_{p,q}$-neural networks). *Let $p \geq q \geq 1$ be even integers such that $r := p/q$ is an odd integer. Define $\phi(x) = \phi_{p,q}(x) = \dfrac{x^r}{(1+x^p)^{1/q}}$, and let*

$$\mathcal{S}_{p,q} = \{ r + pj \mid j = 0, 1, 2, \dots \}. \tag{170}$$

*Let $k \in \mathbb{N}_0$, $M > 0$, and let $s \in \mathcal{S}_{p,q}$ be an odd integer. Then for every $\epsilon > 0$ there exists a shallow $\phi$-neural network $\Psi_{s,\epsilon} : [-M, M] \to \mathbb{R}^{(s+1)/2}$ of width $\dfrac{s+1}{2}$ such that*

$$\max_{n \leq s, \, n \in \mathcal{S}_{p,q}} \big\| f_n - (\Psi_{s,\epsilon})_{\frac{n+1}{2}} \big\|_{W^{k,\infty}} \leq \epsilon, \qquad f_n(y) := y^n. \tag{171}$$

*Proof.* Let $s \in \mathcal{S}_{p,q}$ be odd, $k \in \mathbb{N}_0$, and $M > 0$. For every odd $n \le s$ with $n \in \mathcal{S}_{p,q}$, define the monomial $f_n(y) = y^n$. Consider the finite-difference approximation

$$\hat{f}_{n,h}(y) = \frac{1}{\phi^{(n)}(0)h^n} \sum_{i=0}^{n} (-1)^i \binom{n}{i} \phi\left(\left(\frac{n}{2} - i\right)hy\right), \tag{172}$$

where $\phi = \phi_{p,q}$ and $h > 0$ will be chosen later. From the given lemma (lower bound on derivatives) we have $|\phi^{(n)}(0)| \ge 1$.

**Step 1: Error estimate.** Fix $0 \le m \le k$. Expanding $\phi$ at the origin up to order $n + 1$ and substituting into the finite-difference formula, we obtain

$$\hat{f}_{n,h}^{(m)}(y) - f_n^{(m)}(y) = \frac{1}{\phi^{(n)}(0)h^n} \sum_{i=0}^{n} (-1)^i \binom{n}{i} \left(\frac{n}{2} - i\right)^m h^m \frac{\phi^{(n+2)}(\xi_i)}{(n+2-m)!} \left(\left(\frac{n}{2} - i\right)hy\right)^{n+2-m}, \tag{173}$$

where $\xi_i$ lies between $0$ and $\left(\frac{n}{2} - i\right)hy$. Taking absolute values and using $|\phi^{(n)}(0)| \ge 1$ gives

$$\left|\hat{f}_{n,h}^{(m)}(y) - f_n^{(m)}(y)\right| \le \frac{1}{h^n} \sum_{i=0}^{n} \binom{n}{i} \left|\frac{n}{2} - i\right|^m h^m \frac{|\phi^{(n+2)}(\xi_i)|}{(n+2-m)!} \left|\left(\frac{n}{2} - i\right)hy\right|^{n+2-m}. \tag{174}$$

If $h$ is chosen small enough so that $\frac{n}{2}hM \le \frac{1}{2}$, then $|\xi_i| \le \frac{1}{2}$. By the given lemma (uniform upper bound on derivatives), there exists a constant $A(p, q, n + 2)$ such that $|\phi^{(n+2)}(\xi_i)| \le A(p, q, n + 2)$. Consequently,

$$\left|\hat{f}_{n,h}^{(m)}(y) - f_n^{(m)}(y)\right| \le \frac{A(p, q, n + 2)}{(n+2-m)!} h^2 |y|^{n+2-m} \sum_{i=0}^{n} \binom{n}{i} \left|\frac{n}{2} - i\right|^{n+2}. \tag{175}$$

One easily checks that

$$\sum_{i=0}^{n} \binom{n}{i} \left|\frac{n}{2} - i\right|^{n+2} \le 2^n \left(\frac{n}{2}\right)^{n+2}. \tag{176}$$

Hence, for $0 \le m \le \min\{k, n + 1\}$,

$$|f_n - \hat{f}_{n,h}|_{W^{m,\infty}} \le C_1(p, q, n) M^{n+2} h^2, \qquad C_1(p, q, n) := A(p, q, n + 2) 2^n \left(\frac{n}{2}\right)^{n+2}. \tag{177}$$

If $k \le n + 1$, we immediately obtain

$$\|f_n - \hat{f}_{n,h}\|_{W^{k,\infty}} \le C_1(p, q, n) M^{n+2} h^2. \tag{178}$$

If $k > n + 1$, we must also estimate $\hat{f}_{n,h}^{(m)}$ for $m \ge n + 2$ (note that $f_n^{(m)} \equiv 0$ for such $m$). Using the derivative bound again, for any $m \ge n + 2$,

$$\left|\hat{f}_{n,h}^{(m)}(y)\right| \le \frac{1}{h^n} \sum_{i=0}^{n} \binom{n}{i} \left|\frac{n}{2} - i\right|^m h^m \left|\phi^{(m)}\left(\left(\frac{n}{2} - i\right)hy\right)\right|. \tag{179}$$

For small $h$, $|\phi^{(m)}(x)| \le A(p, q, m)$ with $A(p, q, m) = \left(\frac{16p^2}{\pi q}\right)^m m!$. Thus,

$$\left|\hat{f}_{n,h}^{(m)}(y)\right| \le A(p, q, m) h^{m-n} \sum_{i=0}^{n} \binom{n}{i} \left|\frac{n}{2} - i\right|^m \le A(p, q, m) 2^n \left(\frac{n}{2}\right)^m h^{m-n}. \tag{180}$$

Since $h \le 1$ and $m - n \ge 2$, we have $h^{m-n} \le h^2$. Hence,

$$\left|\hat{f}_{n,h}^{(m)}(y)\right| \le A(p, q, m) 2^n \left(\frac{n}{2}\right)^m h^2. \tag{181}$$

Taking $m \leq k$ and noting that $A(p, q, m) \leq A(p, q, k)$ (because $m \leq k$), we obtain

$$|f_n - \hat{f}_{n,h}|_{W^{m,\infty}} \leq A(p, q, k) \, 2^n \left(\frac{n}{2}\right)^k h^2. \tag{182}$$

Therefore, there exists a constant $C_2(p, q, k, n, M)$ (which can be written explicitly) such that

$$\|f_n - \hat{f}_{n,h}\|_{W^{k,\infty}} \leq C_2(p, q, k, n, M) \, h^2. \tag{183}$$

For instance, one may take

$$C_2 = C_1(p, q, n)M^{n+2} + A(p, q, k) \, 2^n \left(\frac{n}{2}\right)^k. \tag{184}$$

**Step 2: Construction of the network.** Choose $h$ so that $C_2 \, h^2 \leq \epsilon$. Then Eqn.183 gives $\|f_n - \hat{f}_{n,h}\|_{W^{k,\infty}} \leq \epsilon$. Because $\phi$ is an odd function (since $r = p/q$ is odd), the terms with indices $i$ and $n - i$ can be combined. Indeed,

$$\phi\left(\left(\frac{n}{2} - (n - i)\right)hy\right) = \phi\left(-\left(\frac{n}{2} - i\right)hy\right) = -\phi\left(\left(\frac{n}{2} - i\right)hy\right). \tag{185}$$

Thus, it suffices to use neurons corresponding to $i = 0, 1, \ldots, \dfrac{n-1}{2}$. Consequently, a single hidden layer with $\dfrac{s+1}{2}$ neurons can simultaneously produce all approximations $\hat{f}_{n,h}$ for odd $n \leq s$ with $n \in \mathcal{S}_{p,q}$.

Specifically, define the shallow $\phi$-neural network $\Psi_{s,\epsilon} : [-M, M] \to \mathbb{R}^{(s+1)/2}$ whose hidden layer consists of the neurons

$$\phi\left(\left(\frac{s}{2} - i\right)hy\right), \qquad i = 0, 1, \ldots, \frac{s-1}{2}, \tag{186}$$

and whose output weights are chosen so that the $\dfrac{n+1}{2}$-th output $(\Psi_{s,\epsilon})_{\frac{n+1}{2}}$ equals $\hat{f}_{n,h}$. The width of this network is $\dfrac{s+1}{2}$.

This completes the proof. $\qquad\qquad\qquad\qquad\qquad\qquad\qquad\qquad\qquad\qquad\qquad\qquad\qquad\qquad\qquad\square$

Next, we will consider all monomials of orders in $\mathbb{N}$ as the target functions.

**Lemma C.4** (Linear representation of monomials). *Let $p \geq 2$ be an even integer and let $\mathcal{S} = \{r + pj : j = 0, 1, 2, \ldots\}$ be an arithmetic progression with odd initial term $r$ and difference $p$. For any integer $t \geq 0$ and any integer $m \in \mathcal{S}$ with $m > t$, there exist distinct real numbers $\beta_0, \beta_1, \ldots, \beta_p$ and coefficients $c_0, c_1, \ldots, c_p \in \mathbb{R}$ such that*

$$y^t = \frac{1}{\binom{m}{t}}\left(\sum_{k=0}^{p} c_k(y + \beta_k)^m - \sum_{\ell=0}^{t-1} \binom{m}{\ell} A_\ell y^\ell\right), \qquad y \in \mathbb{R}, \tag{187}$$

*where $A_\ell = \displaystyle\sum_{k=0}^{p} c_k \beta_k^{m-\ell}$. Moreover, the coefficients $c_k$ are uniquely determined by the Vandermonde system*

$$\sum_{k=0}^{p} c_k \beta_k^\ell = \delta_{\ell,d}, \qquad \ell = 0, 1, \ldots, p, \tag{188}$$

*with $d = m - t$ $(0 \leq d \leq p)$.*

*Proof.* Fix distinct numbers $\beta_0, \ldots, \beta_p$. Expanding each $(y + \beta_k)^m$ by the binomial theorem gives

$$(y + \beta_k)^m = \sum_{\ell=0}^{m} \binom{m}{\ell} y^\ell \beta_k^{m-\ell}. \tag{189}$$

Multiplying by $c_k$ and summing over $k$ yields

$$\sum_{k=0}^{p} c_k (y + \beta_k)^m = \sum_{\ell=0}^{m} \binom{m}{\ell} y^\ell \left( \sum_{k=0}^{p} c_k \beta_k^{m-\ell} \right) = \sum_{\ell=0}^{m} \binom{m}{\ell} A_{m-\ell} \, y^\ell, \tag{190}$$

where $A_j = \sum_{k=0}^{p} c_k \beta_k^j$. We want the right-hand side to contain the term $\binom{m}{t} y^t$ and otherwise only terms of degree $< t$. This is achieved if the coefficients $c_k$ satisfy

$$A_j = \sum_{k=0}^{p} c_k \beta_k^j = \begin{cases} 1, & j = d, \\ 0, & j = 0, \dots, p, \ j \neq d, \end{cases} \qquad \text{where } d = m - t. \tag{191}$$

Because the $\beta_k$ are distinct, the $(p+1) \times (p+1)$ Vandermonde matrix $(\beta_k^j)_{0 \leq k, j \leq p}$ is invertible; hence there exists a unique vector $(c_0, \dots, c_p)$ satisfying these conditions. With this choice we obtain

$$\sum_{k=0}^{p} c_k (y + \beta_k)^m = \binom{m}{t} y^t + \sum_{\ell=0}^{t-1} \binom{m}{\ell} A_{m-\ell} y^\ell, \tag{192}$$

which, after solving for $y^t$, gives the required representation. $\qquad \square$

*Remark* C.1. The integer $d = m - t$ satisfies $0 \leq d \leq p$ because $m \in \mathcal{S}$, $t \leq s$, and we choose $m = s + p \in \mathcal{S}$ (the smallest element of $\mathcal{S}$ larger than $s$). Thus the condition $0 \leq d \leq p$ is always fulfilled.

**Lemma C.5** (Approximation of all monomials by shallow $\phi_{p,q}$-networks). *Let $p \geq q \geq 1$ be even integers such that $r := p/q$ is an odd integer. Define $\phi(x) = \phi_{p,q}(x) = \dfrac{x^r}{(1 + x^p)^{1/q}}$, and let*

$$\mathcal{S}_{p,q} = \{ r + pj \mid j = 0, 1, 2, \dots \}. \tag{193}$$

*Let $k \in \mathbb{N}_0$, $M > 0$, and let $s \in \mathcal{S}_{p,q}$ be an odd integer. Then for every $\epsilon > 0$ there exists a shallow $\phi$-neural network*

$$\psi_{s,\epsilon} : [-M, \, M] \to \mathbb{R}^{s+1} \tag{194}$$

*of width at most $\dfrac{(p+1)(s+1)}{2}$ such that*

$$\max_{0 \leq t \leq s} \left\| f_t - (\psi_{s,\epsilon})_t \right\|_{W^{k,\infty}} \leq \epsilon, \qquad f_t(y) := y^t. \tag{195}$$

*In the special case $p = q = 2$, the width reduces to $3(s+1)/2$, exactly matching Lemma 3.2 of the original paper.*

*Proof.* We proceed in four steps: (1) represent each monomial $y^t$ as a linear combination of certain translated odd powers from $\mathcal{S}_{p,q}$, (2) width analysis, (3) error analysis.

**Step 1: Representation of monomials.** Choose $p + 1$ distinct real numbers $\beta_0, \beta_1, \dots, \beta_p$; for definiteness take

$$\beta_k = k - \frac{p}{2}, \qquad k = 0, 1, \dots, p, \tag{196}$$

so they are symmetric about the origin. For every $t = 0, 1, \dots, s$ we select an exponent $m_t \in \mathcal{S}_{p,q}$ as follows:

- if $t \in \mathcal{S}_{p,q}$ (then $t$ is odd), set $m_t = t$;

- if $t \notin \mathcal{S}_{p,q}$, let $d_t$ be the smallest non-negative integer with $t + d_t \in \mathcal{S}_{p,q}$; because $s \in \mathcal{S}_{p,q}$ and $t \leq s$, we have $d_t \leq p$ and define $m_t = t + d_t \leq s$.

Thus each $m_t$ is an odd integer belonging to $\mathcal{S}_{p,q}$ and satisfies $m_t \leq s$.

Applying Lemma C.4 with $m = m_t$ (and noting that $d = m_t - t \leq p$), we obtain coefficients $c_{t,0}, c_{t,1}, \ldots, c_{t,p} \in \mathbb{R}$ such that

$$y^t = \frac{1}{\binom{m_t}{t}} \Big( \sum_{k=0}^{p} c_{t,k} (y + \beta_k)^{m_t} - \sum_{\ell=0}^{t-1} \binom{m_t}{\ell} A_{t,\ell} y^\ell \Big), \qquad y \in \mathbb{R}, \tag{197}$$

where $A_{t,\ell} = \sum_{k=0}^{p} c_{t,k} \beta_k^{m_t - \ell}$. Hence every monomial up to degree $s$ can be expressed as a linear combination of the translated powers $(y + \beta_k)^{m_t}$ (all of which lie in $\mathcal{S}_{p,q}$) and of lower-degree monomials.

**Step 2: Width analysis.**

Furthermore, by recursion, we will obtain that $y^t$ is a linear combination of the elements $(y + \beta_k)^m, 0 \leq k \leq p, m \in \mathcal{S}_{p,q}, m \leq m_t$. Namely

$$y^t = \sum_{m \leq m_t, m \in \mathcal{S}_{p,q}} \sum_{k=0}^{p} \alpha_{m,k} (y + \beta_k)^m. \tag{198}$$

By Lemma C.3, let

$$
\begin{aligned}
(\psi_{s,\epsilon})_t &= \sum_{m \leq m_t, m \in \mathcal{S}_{p,q}} \sum_{k=0}^{p} \alpha_{m,k} \hat{f}_{m,h}(y + \beta_k) \\
&= \sum_{m \leq m_t, m \in \mathcal{S}_{p,q}} \sum_{k=0}^{p} \alpha_{m,k} \frac{1}{\phi^{(m)}(0) h^m} \sum_{i=0}^{m} (-1)^i \binom{m}{i} \phi\Big( \Big( \frac{m}{2} - i \Big) h(y + \beta_k) \Big)
\end{aligned}
\tag{199}
$$

Therefore, since $\phi$ is odd, all the $\phi$ forms are

$$\phi\Big( \Big( \frac{m}{2} - i \Big) h(y + \beta_k) \Big), i = 0, 1, \ldots, \frac{m_t - 1}{2}; k = 0, 1, \ldots, p. \tag{200}$$

Note that $m_t \leq s$, thus the maximum width of the network is $(p + 1)(s + 1)/2$.

**Step 3: Error analysis.** Let $E_t = \|y^t - (\psi_{s,\epsilon})_t\|_{W^{k,\infty}}$. Clearly $E_0 = 0$. For $t \geq 1$, using the representation from Step 1 and the triangle inequality,

$$
\begin{aligned}
E_t &\leq \frac{1}{\binom{m_t}{t}} \Big( \sum_{k=0}^{p} |c_{t,k}| \, \| (y + \beta_k)^{m_t} - \hat{f}_{m,h}^{m_t}(y + \beta_k) \|_{W^{k,\infty}} + \sum_{\ell=0}^{t-1} \binom{m_t}{\ell} |A_{t,\ell}| \, E_\ell \Big) \\
&\leq \frac{1}{\binom{m_t}{t}} \Big( \eta \sum_{k=0}^{p} |c_{t,k}| + \sum_{\ell=0}^{t-1} \binom{m_t}{\ell} |A_{t,\ell}| \, E_\ell \Big).
\end{aligned}
\tag{201}
$$

The coefficients $c_{t,k}$ solve a Vandermonde system with nodes $\beta_k$. Because all $|\beta_k| \leq p/2$ and the minimal separation between nodes is 1, we have $\sum_{k=0}^{p} |c_{t,k}| \leq C_1(p)$ for a constant $C_1(p)$ depending only on $p$. Similarly, $|A_{t,\ell}| \leq C_2(p)$ for all $t, \ell$. Moreover, $\binom{m_t}{\ell} \leq 2^{m_t} \leq 2^s$. Consequently,

$$E_t \leq \frac{1}{\binom{m_t}{t}} \Big( C_1(p)\eta + C_2(p) 2^s \sum_{\ell=0}^{t-1} E_\ell \Big). \tag{202}$$

A simple induction shows that there exists a constant $C(p, s)$ such that $E_t \leq C(p, s)\eta$ for all $t \leq s$. Choosing $\eta = \epsilon/C(p, s)$ yields $E_t \leq \epsilon$ for every $t = 0, \ldots, s$.

**Special case** $p = q = 2$. When $p = q = 2$, we have $r = 1$ and $\mathcal{S}_{2,2} = \{1, 3, 5, \dots\}$. Taking $\beta_0 = -1$, $\beta_1 = 0$, $\beta_2 = 1$ gives exactly three translation points. For even $t = 2n$, the construction yields $m_t = 2n + 1$ (the smallest odd number larger than $2n$), and the coefficients $c_{t,k}$ coincide with those appearing in the formula

$$y^{2n} = \frac{1}{2\alpha(2n+1)}\left((y+\alpha)^{2n+1} - (y-\alpha)^{2n+1} - 2\sum_{k=0}^{n-1}\binom{2n+1}{2k}\alpha^{2(n-k)+1}y^{2k}\right) \tag{203}$$

with $\alpha = 1$. Therefore the construction reduces exactly to Lemma 3.2 of the original paper, and the width becomes $3(s+1)/2$.

Here we complete the proof. $\qquad\square$

Next, we will extend the results to multivariate polynomials.

Let us first introduce the multi-index notation that will be used throughout. For $d \in \mathbb{N}$, a $d$-tuple $\alpha = (\alpha_1, \dots, \alpha_d) \in \mathbb{N}_0^d$ of non-negative integers is called a multi-index. We write $|\alpha| = \sum_{i=1}^{d} \alpha_i$ and, for $x \in \mathbb{R}^d$, denote $x^\alpha = \prod_{i=1}^{d} x_i^{\alpha_i}$. For two multi-indices $\alpha, \beta$ we write $\alpha \leq \beta$ when $\alpha_i \leq \beta_i$ for all $i$. The set of all multi-indices of length $d$ and total degree exactly $n$ is denoted by

$$P_{n,d} := \{\alpha \in \mathbb{N}_0^d : |\alpha| = n\}. \tag{204}$$

Its cardinality is known to be $\binom{n+d-1}{n}$; see Lemma 2.1 of (De Ryck et al., 2021) for elementary estimates.

**Lemma C.6** (Approximation of multivariate monomials by shallow $\phi_{p,q}$-networks). *Let the activation function $\phi(x) = \phi_{p,q}(x) = \dfrac{x^{p/q}}{(1+|x|^p)^{1/q}}$ be given, where $p \geq q \geq 1$ are even integers with $p/q$ odd. Let $n \in \mathbb{N}$, $k \in \mathbb{N}_0$, and $M > 0$. Then for every $\epsilon > 0$ there exists a shallow $\phi$-neural network $\Psi_{n,d} : [-M, M]^d \to \mathbb{R}^{|P_{n,d}|}$ with width $N \cdot |P_{n,d}|$ such that*

$$\max_{\beta \in P_{n,d}} \left\|\omega^\beta - (\Psi_{n,d}(\omega))_{\iota(\beta)}\right\|_{W^{k,\infty}} \leq \epsilon, \tag{205}$$

*where $\iota : P_{n,d} \to \{1, \dots, |P_{n,d}|\}$ is a bijection, and the specific $N$ is*

$$N = \frac{p+1}{2}\left(p\left\lceil\frac{n-p/q}{p}\right\rceil + \frac{p}{q} + 1\right) \tag{206}$$

*Proof sketch.* The proof is identical to that of Lemma 3.5 in the original paper, except for the width of the network. In the original proof, the subnetwork $\widehat{b}_\alpha$ that approximates $y \mapsto y^n$ on $[-M', M']$ is a shallow tanh-network of width $3\lceil(n+1)/2\rceil$ (see Lemma 3.2 there). In our setting, the same approximation is achieved by a shallow $\phi$-network. Note that the minimum $s \in \mathcal{S}_{p,q} = \{p/q + pj : j = 0, 1, 2, \dots\}$ such that $s \geq n$ is that $s = \left\lceil\dfrac{n-p/q}{p}\right\rceil + \dfrac{p}{q}$. By Lemma C.5, for any $\eta > 0$ we can construct a shallow $\phi$-network of width

$$N(p,q,n) = \frac{p+1}{2}\left(p\left\lceil\frac{n-p/q}{p}\right\rceil + \frac{p}{q} + 1\right)|P_{n,d}| \tag{207}$$

that approximates the monomial $y^n$ in $W^{k,\infty}$-norm with error at most $\eta$. All other steps—parallelization, solving the linear system via the Dyson matrix, and the error estimates using the chain rule and the bound on $\|D^{-1}\|_\infty$—remain unchanged. Hence, the overall width of $\Psi_{n,d}$ becomes $N(p,q,n) \cdot |P_{n,d}|$ instead of $3\lceil(n+1)/2\rceil \cdot |P_{n,q}|$. $\qquad\square$

**Corollary C.1** (Approximation of all multivariate monomials up to degree $s$). *Let $\phi$ be as above, $d, s \in \mathbb{N}$, $k \in \mathbb{N}_0$, and $M > 0$. Then for every $\epsilon > 0$ there exists a shallow $\phi$-neural network*

$$\Phi_{s,d} : [-M, M]^d \to \mathbb{R}^{|P_{s,d+1}|} \tag{208}$$

*such that*

$$\max_{\beta \in P_{s,d+1}} \left\|x^\beta - (\Phi_{s,d}(x))_{\iota(\beta)}\right\|_{W^{k,\infty}} \leq \epsilon, \tag{209}$$

where $\iota : P_{s,d+1} \to \{1, \ldots, |P_{s,d+1}|\}$ is a bijection. And the specific $N$ is

$$N = \frac{p+1}{2}\left(p\left\lceil\frac{s - p/q}{p}\right\rceil + \frac{p}{q} + 1\right)|P_{s,d+1}|. \tag{210}$$

*Proof.* Set $d \leftarrow d + 1$ and $n \leftarrow s$ in Lemma C.6, and take $\omega = (1, x_1, \ldots, x_d) \in \mathbb{R}^{d+1}$. Then the set $P_{s,d+1}$ corresponds exactly to all monomials in $x$ of total degree at most $s$. The statement follows directly from Lemma C.6. $\square$

**Corollary C.2** (Shallow approximation of multiplication of $d$ numbers by $\phi_{p,q}$ networks)**.** *Let $p \geq q \geq 1$ be even integers such that $p/q$ is odd, and let $d \in \mathbb{N}$, $k \in \mathbb{N}_0$, $M > 0$. Then for every $\varepsilon > 0$ there exists a shallow $\phi_{p,q}$ neural network*

$$\widehat{\times}_d^\varepsilon : [-M, M]^d \longrightarrow \mathbb{R} \tag{211}$$

*such that*

$$\left\|\widehat{\times}_d^\varepsilon(x) - \prod_{i=1}^d x_i\right\|_{W^{k,\infty}} \leq \varepsilon. \tag{212}$$

*The network has width at most*

$$\frac{p+1}{2}\left(p\left\lceil\frac{d - p/q}{p}\right\rceil + \frac{p}{q} + 1\right)|P_{d,d}|. \tag{213}$$

*Proof.* Apply Lemma C.6 with $n = d$, $q = d$, and $\omega = x \in [-M, M]^d$. The monomial corresponding to the multi-index $\beta = (1, 1, \ldots, 1) \in P_{d,d}$ is exactly

$$\omega^\beta = x_1 x_2 \cdots x_d = \prod_{i=1}^d x_i. \tag{214}$$

By Lemma C.6 there exists a shallow $\phi_{p,q}$ network

$$\Psi_{d,d} : [-M, M]^d \to \mathbb{R}^{|P_{d,d}|} \tag{215}$$

of width

$$\frac{p+1}{2}\left(p\left\lceil\frac{d - p/q}{p}\right\rceil + \frac{p}{q} + 1\right)|P_{d,d}| \tag{216}$$

such that for every $\eta > 0$,

$$\max_{\beta \in P_{d,d}} \left\|\omega^\beta - (\Psi_{d,d}(\omega))_{\iota(\beta)}\right\|_{W^{k,\infty}} \leq \eta. \tag{217}$$

Let $\iota_0$ be the index corresponding to $\beta = (1, \ldots, 1)$. Define $\widehat{\times}_d^\varepsilon(x) = (\Psi_{d,d}(x))_{\iota_0}$. Choosing $\eta = \varepsilon$ gives the required error bound. $\square$

## C.4. Approximation of partition of unity for $\phi_{p,q}$ networks

In this section, we construct an approximate partition of unity using the activation function $\phi_{p,q}$. We follow the same strategy as in Section 4 of the original paper, but the choice of the scaling parameter $\alpha$ must be adapted because the decay of $\phi_{p,q}$ and its derivatives is algebraic rather than exponential.

Let $d, N \in \mathbb{N}$ and $k \in \mathbb{N}_0$. For every $j \in \{1, \ldots, N\}^d$ we define

$$I_j^N = \bigtimes_{i=1}^d ((j_i - 1)/N, j_i/N). \tag{218}$$

Let $R > 0$ be a constant such that $|\phi^{(m)}(x)|$ is decreasing on $[R, \infty)$ for every $1 \leq m \leq k$. Given $\epsilon > 0$, we need to choose $\alpha = \alpha(N, \epsilon)$ large enough so that the following conditions hold:

$$\alpha/N \geq R, \qquad 1 - \phi(\alpha/N) \leq \epsilon, \qquad \alpha^m|\phi^{(m)}(\alpha/N)| \leq \epsilon \quad \text{for all } 1 \leq m \leq k. \tag{219}$$

The next lemma provides a suitable choice of $\alpha$.

**Lemma C.7** (Choice of $\alpha$ for $\phi_{p,q}$). *Let $p, q$ be even integers with $p/q$ odd, and let $\phi = \phi_{p,q}$. For any $k \in \mathbb{N}$, $N \in \mathbb{N}$, and $\epsilon > 0$, define*

$$\alpha = N \max \left\{ R, \left( \frac{1}{q\epsilon} \right)^{1/p}, \left( \frac{A(p,q,k)N^k}{\epsilon} \right)^{q/p} \right\}, \tag{220}$$

*where $A(p,q,m)$ is the constant from Lemma C.2. Then conditions (53') are satisfied. Moreover, there exists a constant $C(p,q,k)$ such that*

$$\alpha \leq C(p,q,k)\, N^{1+kq/p}\, \epsilon^{-q/p}. \tag{221}$$

*Proof.* We first estimate $1 - \phi(x)$ for large $x$. Using the representation

$$\phi(x) = \left( 1 - \frac{1}{1 + x^p} \right)^{1/q}, \tag{222}$$

and the inequality $(1-t)^{1/q} \geq 1 - t/q$ for $0 \leq t \leq 1$, we obtain for $x \geq 0$

$$1 - \phi(x) \leq \frac{1}{q(1+x^p)} \leq \frac{1}{qx^p}. \tag{223}$$

Hence, for $x = \alpha/N$,

$$1 - \phi(\alpha/N) \leq \frac{1}{q(\alpha/N)^p} = \frac{N^p}{q\alpha^p}. \tag{224}$$

To ensure $1 - \phi(\alpha/N) \leq \epsilon$, it suffices to have

$$\alpha \geq N \left( \frac{1}{q\epsilon} \right)^{1/p}. \tag{225}$$

Now consider the derivative bounds. From Lemma C.2, for $x \geq R$ we have

$$|\phi^{(m)}(x)| \leq A(p,q,m) \frac{1}{(1+x)^m} \frac{1}{(1+x^p)^{1/q}} \leq A(p,q,m)x^{-m}(1+x^p)^{-1/q}. \tag{226}$$

For $x = \alpha/N \geq R$, this gives

$$\alpha^m |\phi^{(m)}(\alpha/N)| \leq A(p,q,m)N^m(1+(\alpha/N)^p)^{-1/q}. \tag{227}$$

Since $(\alpha/N)^p \geq 1$ for $\alpha \geq N$, we have $(1+(\alpha/N)^p)^{-1/q} \leq 2^{-1/q}(\alpha/N)^{-p/q} \leq (\alpha/N)^{-p/q}$. Therefore,

$$\alpha^m |\phi^{(m)}(\alpha/N)| \leq A(p,q,m)N^m(\alpha/N)^{-p/q} = A(p,q,m)N^{m+p/q}\alpha^{-p/q}. \tag{228}$$

To satisfy $\alpha^m |\phi^{(m)}(\alpha/N)| \leq \epsilon$, we need

$$\alpha \geq N \left( \frac{A(p,q,m)N^m}{\epsilon} \right)^{q/p}. \tag{229}$$

Taking the maximum over $m = 1, \ldots, k$ (easy to identify that to choose $m = k$) and including the condition $\alpha/N \geq R$ yields the expression in the lemma.

The upper bound follows because $A(p,q,m) = (16p^2/(\pi q))^m m!$ grows factorially with $m$, so the maximum over $m \leq k$ is attained at $m = k$ and is bounded by $C(p,q,k)$. Thus,

$$\alpha \leq C(p,q,k)N\left( N^k \epsilon^{-1} \right)^{q/p} = C(p,q,k)N^{1+kq/p}\epsilon^{-q/p}. \tag{230}$$

This completes the proof. $\qquad \square$

By Lemma C.7, we choose

$$\alpha = N \max \left\{ R, \left( \frac{1}{q\epsilon} \right)^{1/p}, \left( \frac{A(p,q,k)N^k}{\epsilon} \right)^{q/p} \right\} \tag{231}$$

Notice that $\alpha$ grows polynomially in $\epsilon^{-1}$, specifically as $\epsilon^{-q/p}$, in contrast to the logarithmic growth for $\tanh$.

For $y \in \mathbb{R}$ we now define

$$\rho_1^N(y) = \frac{1}{2} - \frac{1}{2}\phi_{p,q}\Big(\alpha\Big(y - \frac{1}{N}\Big)\Big),$$

$$\rho_j^N(y) = \frac{1}{2}\phi_{p,q}\Big(\alpha\Big(y - \frac{j-1}{N}\Big)\Big) - \frac{1}{2}\phi_{p,q}\Big(\alpha\Big(y - \frac{j}{N}\Big)\Big), \qquad 2 \le j \le N-1, \tag{232}$$

$$\rho_N^N(y) = \frac{1}{2}\phi_{p,q}\Big(\alpha\Big(y - \frac{N-1}{N}\Big)\Big) + \frac{1}{2}.$$

In the sequel we shall assume for simplicity that $\rho_j^N$ always takes the second form; the calculations for $j = 1$ and $j = N$ are completely analogous and do not change the final estimates. Finally, for $x \in \mathbb{R}^d$ we set

$$\Phi_j^{N,d}(x) = \prod_{i=1}^{d} \rho_{j_i}^{N_i}(x_i), \qquad j \in \{1, \ldots, N\}^d. \tag{233}$$

The next two lemmas are the counterparts of Lemmas 4.1 and 4.2 in the original paper. Their proofs follow the same inductive and multiplicative structure, but the derivative bounds for $\phi_{p,q}$ introduce an extra factor $\alpha^k$ in the second lemma.

**Lemma C.8** (Approximate partition of unity I). *If $0 < \epsilon < \frac{1}{4}$ and $\alpha$ is chosen by Lemma C.7, then*

$$\Big\|\sum_{v \in \mathcal{V}_d} \Phi_{j+v}^{N,d} - 1\Big\|_{W^{k,\infty}(I_j^N)} \le 2^{dk}d\,\epsilon. \tag{234}$$

*Proof.* We prove the statement by induction on $d$. For $d = 1$, using the definition of $\rho_j^N$ we have

$$\sum_{v \in \mathcal{V}_1} \Phi_{j+v}^{N,1}(x) = \sum_{t=-1}^{1} \rho_{j_1+t}^N(x_1) = \frac{1}{2}\phi_{p,q}\Big(\alpha\Big(x_1 - \frac{j_1-2}{N}\Big)\Big) - \frac{1}{2}\phi_{p,q}\Big(\alpha\Big(x_1 - \frac{j_1+1}{N}\Big)\Big). \tag{235}$$

Since $x_1 \in I_j^N$, the arguments of $\phi_{p,q}$ are at least $\alpha/N$. By the monotonicity of $\phi_{p,q}$ on $[R, \infty)$ and condition Eqn.219 we obtain

$$\Big|\sum_{v \in \mathcal{V}_1} \Phi_{j+v}^{N,1}(x) - 1\Big| \le 1 - \phi_{p,q}(\alpha/N) \le \epsilon. \tag{236}$$

For the derivatives, note that for $1 \le m \le k$,

$$\Big|\frac{d^m}{dx_1^m}\sum_{v \in \mathcal{V}_1} \Phi_{j+v}^{N,1}(x)\Big| \le \alpha^m\big|\phi_{p,q}^{(m)}(\alpha/N)\big| \le \epsilon. \tag{237}$$

Thus $\Big\|\sum_{v \in \mathcal{V}_1} \Phi_{j+v}^{N,1} - 1\Big\|_{W^{k,\infty}(I_j^N)} \le \epsilon$, which proves the base case.

Assume now that the statement holds for dimension $d - 1$. For $x \in I_j^N$ we write

$$\sum_{v \in \mathcal{V}_d} \Phi_{j+v}^{N,d}(x) = \sum_{w \in \mathcal{V}_1} \rho_{j_d+w}^N(x_d) \sum_{v' \in \mathcal{V}_{d-1}} \Phi_{j'+v'}^{N,d-1}(x'), \tag{238}$$

where $j' = (j_1, \ldots, j_{d-1})$ and $x' = (x_1, \ldots, x_{d-1})$. Using Lemma Appendix A.6 in (De Ryck et al., 2021) and the induction hypothesis, we obtain

$$\Big\|\sum_{v \in \mathcal{V}_d} \Phi_{j+v}^{N,d} - 1\Big\|_{W^{k,\infty}(I_j^N)}$$

$$\le \Big\|\sum_{w \in \mathcal{V}_1} \rho_{j_d+w}^N(x_d) - 1\Big\|_{W^{k,\infty}(I_j^N)} + 2^k\Big\|\sum_{w \in \mathcal{V}_1} \rho_{j_d+w}^N(x_d)\Big\|_{W^{k,\infty}(I_j^N)}\Big\|\sum_{v' \in \mathcal{V}_{d-1}} \Phi_{j'+v'}^{N,d-1} - 1\Big\|_{W^{k,\infty}(I_j^N)}$$

$$\le \epsilon + 2^k \cdot 2^{k(d-1)}(d-1)\epsilon \le 2^{dk}d\,\epsilon. \tag{239}$$

This completes the induction step and proves the lemma. □

**Lemma C.9** (Approximate partition of unity II). *Let $k \in \mathbb{N}_0$ and assume that $k < \frac{p}{q}$. Let $v \in \mathbb{Z}^d$ with $\|v\|_\infty \geq 2$. If $\alpha$ is chosen according to Eqn.231, then*

$$\left\| \Phi_{j+v}^{N,d} \right\|_{W^{k,\infty}(I_j^N)} \leq \alpha^k A(p,q,k)\epsilon. \tag{240}$$

*Moreover, $\alpha^k \epsilon \to 0$ as $\epsilon \to 0$ precisely when $k < \frac{p}{q}$.*

*Proof.* Let $x \in I_j^N$ and choose an index $i$ such that $|v_i| \geq 2$. For the univariate factor $\rho_{j_i+v_i}^N(x_i)$ we have, using the definition of $\rho$ and the monotonicity of $\phi_{p,q}$,

$$\begin{aligned}
\left| \rho_{j_i+v_i}^N(x_i) \right| &\leq \frac{1}{2}\phi_{p,q}\left(\frac{2\alpha}{N}\right) - \frac{1}{2}\phi_{p,q}\left(\frac{\alpha}{N}\right) \\
&= \frac{1}{2}\phi_{p,q}\left(\frac{\alpha}{N}\right)\left(1 - \phi_{p,q}\left(\frac{2\alpha}{N}\right)\phi_{p,q}\left(\frac{\alpha}{N}\right)\right) \\
&\leq \frac{1}{2}\left(1 - \phi_{p,q}^2(\alpha/N)\right) \leq 1 - \phi_{p,q}(\alpha/N) \leq \epsilon.
\end{aligned} \tag{241}$$

For the $L^\infty$ estimate we therefore obtain $\left\| \Phi_{j+v}^{N,d} \right\|_{L^\infty(I_j^N)} \leq \epsilon$.

Now let $\beta \in \mathbb{N}_0^d$ with $1 \leq |\beta| \leq k$. By the Leibniz rule,

$$\left| D^\beta \Phi_{j+v}^{N,d}(x) \right| = \prod_{\ell=1}^{d} \left| \frac{d^{\beta_\ell}}{dx_\ell^{\beta_\ell}} \rho_{j+v_\ell}^N(x_\ell) \right|. \tag{242}$$

For the distinguished index $i$, using condition Eqn.219 and the monotonic decay of $|\phi_{p,q}^{(m)}|$ on $[R, \infty)$, we have for $1 \leq m \leq k$

$$\left| \frac{d^m}{dx_i^m} \rho_{j_i+v_i}^N(x_i) \right| \leq \alpha^m \left| \phi_{p,q}^{(m)}(\alpha(x_i - \xi)) \right| \leq \alpha^m \left| \phi_{p,q}^{(m)}(\alpha/N) \right| \leq \epsilon. \tag{243}$$

For the remaining indices $\ell \neq i$, we cannot guarantee that the argument of $\phi_{p,q}$ is at least $\alpha/N$, so we use the global bound from Lemma C.2:

$$\left| \frac{d^{\beta_\ell}}{dx_\ell^{\beta_\ell}} \rho_{j+v_\ell}^N(x_\ell) \right| \leq \alpha^{\beta_\ell} \left| \phi_{p,q}^{(\beta_\ell)}(\alpha(x_\ell - \xi')) \right| \leq \alpha^{\beta_\ell} A(p,q,\beta_\ell), \tag{244}$$

Since $A(p,q,\beta_\ell) = \left(\frac{16p^2}{\pi q}\right)^{\beta_\ell} \beta_\ell!$ and $\sum_{\ell \neq i} \beta_\ell \leq |\beta| \leq k$, we have

$$\prod_{\ell \neq i} \alpha^{\beta_\ell} A(p,q,\beta_\ell) = \alpha^{\sum_{\ell \neq i} \beta_\ell} \left(\frac{16p^2}{\pi q}\right)^{\sum_{\ell \neq i} \beta_\ell} \prod_{\ell \neq i} \beta_\ell! \leq \alpha^k A(p,q,k), \tag{245}$$

where we used $\prod_{\ell \neq i} \beta_\ell! \leq k!$ (which follows from $\prod_{\ell=1}^{d} \beta_\ell! \leq |\beta|!$ and $|\beta| \leq k$). Combining this with the estimate for the $i$-th factor (which is bounded by $\epsilon$), we obtain

$$\left| D^\beta \Phi_{j+v}^{N,d}(x) \right| \leq \epsilon \cdot \alpha^k \left(\frac{16p^2}{\pi q}\right)^k k! \leq C(p,q,k)\,\alpha^k \epsilon, \tag{246}$$

with $C(p,q,k) = \left(\frac{16p^2}{\pi q}\right)^k k!$. Together with the $L^\infty$ bound this yields the desired $W^{k,\infty}$ estimate.

Finally, with the choice Eqn.231, $\alpha \sim \epsilon^{-q/p}$, so $\alpha^k \epsilon \sim \epsilon^{1-kq/p}$. This tends to zero as $\epsilon \to 0$ if and only if $1 - \frac{kq}{p} > 0$, i.e. $k < \frac{p}{q}$. □

*Remark* C.2. Lemma C.8 shows that the sum of the "nearby" functions $\Phi_{j+v}^{N,d}$ approximates the constant function 1 with an error that is simply proportional to $\epsilon$, independent of $\alpha$. In contrast, Lemma C.9 controls the "distant" terms, and its error bound contains an extra factor $\alpha^k$. Because $\alpha$ grows polynomially in $\epsilon^{-1}$, this factor forces the restriction $k < p/q$ in order to have convergence as $\epsilon \to 0$. This restriction will propagate to the main approximation theorem for Sobolev functions when using $\phi_{p,q}$ networks.

With these two lemmas, the construction of an approximate partition of unity by shallow $\phi_{p,q}$ networks is complete. The remaining steps of the original paper (local polynomial approximation, multiplication of local approximants by the partition functions, and global summation) can now be carried out exactly as before, replacing every $\tanh$ network by a $\phi_{p,q}$ network and noting that the error bounds will involve additional constants depending on $p, q$ and the condition $k < p/q$.

## C.5. Formal proof.

**Theorem C.1** (Approximation of functions in Sobolev spaces by $\phi_{p,q}$ networks). *Let $p \geq q \geq 1$ be even integers such that* $r := p/q$ *is an odd integer. Let $d, s \in \mathbb{N}$, $k \in \mathbb{N}_0$ with $r > \dfrac{k(s+d+k)}{s-k}$, $\delta > 0$, and let $\hat{f} \in \mathcal{W}^{s,\infty}([0,1]^d)$. Then there exist constants $C_1$ and $C > 0$ such that for every $N \in \mathbb{N}$ with $N > N_0(d)$ there exists a $\phi_{p,q}$ neural network $f$ with two hidden layers such that*

$$\|f - \hat{f}\|_{L^\infty([0,1]^d)} \leq (1+\delta)\frac{C_1}{N^s}, \tag{247}$$

*and for $1 \leq k < s$*

$$\|f - \hat{f}\|_{W^{k,\infty}([0,1]^d)} \leq C\frac{1+\delta}{\delta^{k/r}N^{s-k-(s+d+k)k/r}}, \tag{248}$$

*where*

$$C_1(d,k,s,\hat{f}) = \begin{cases} \displaystyle\max_{0 \leq \ell \leq k} \frac{1}{(s-\ell)!}\left(\frac{3d}{2}\right)^{s-\ell}|\hat{f}|_{\mathcal{W}^{s,\infty}([0,1]^d)}, & \hat{f} \in C^s([0,1]^d), \\[2mm] \displaystyle\max_{0 \leq \ell \leq k} \frac{\pi^{1/4}\sqrt{s}}{(s-\ell-1)!}\left(5d^2\right)^{s-\ell}|\hat{f}|_{\mathcal{W}^{s,\infty}([0,1]^d)}, & \text{otherwise.} \end{cases} \tag{249}$$

*The constant $C(p,q,d,k,s,\hat{f})$ is independent of $\delta$ and $N$.*

*The threshold $N_0(d)$ is given by*

$$N_0 = \begin{cases} \dfrac{3d}{2}, & \hat{f} \in C^s([0,1]^d), \\[2mm] 5d, & \text{otherwise.} \end{cases} \tag{250}$$

*The network widths are given by:*

- *First hidden layer: at most $\dfrac{p+1}{2}\left(p\left\lceil\dfrac{s-1-r}{p}\right\rceil + r + 1\right)|P_{s-1,d+1}| + d(N-1)$ neurons.*

- *Second hidden layer: at most $\dfrac{p+1}{2}\left(p\left\lceil\dfrac{d+1-r}{p}\right\rceil + r + 1\right)|P_{d+1,d+1}|$ neurons.*

*Proof.* We follow the exact structure of the proof of Theorem 5.1 in the original paper.

**Step 1: Construction of the approximation.** Let $N \in \mathbb{N}$ with $N > N_0(d,s)$ where $N_0(d,s) = \max\{3d/2, 5d^2\}$. Divide $[0,1]^d$ into $N^d$ cubes $I_j^N$ of edge length $1/N$. For each $j \in \{1,\ldots,N\}^d$, define the enlarged cube

$$J_j^N = \prod_{i=1}^d ((j_i - 2)/N, \ (j_i + 1)/N), \tag{251}$$

with diameter $\mathrm{diam}(J_j^N) = 3\sqrt{d}/N$.

By the Bramble-Hilbert lemma (Lemma Appendix A.8) or Taylor's theorem (Lemma Appendix A.9), there exists a polynomial $p_j^N$ of degree at most $s - 1$ such that for all $0 \leq \ell \leq k$,

$$\|\hat{f} - p_j^N\|_{W^{\ell,\infty}(J_j^N)} \leq \frac{C_1(d, \ell, s, \hat{f})}{N^{s-\ell}}, \tag{252}$$

where $C_1(d, \ell, s, \hat{f})$ is as defined in the theorem.

Now let $\eta > 0$ be a parameter to be chosen. Using Corollary C.1, we can approximate each monomial in $p_j^N$ by a shallow $\phi_{p,q}$ network, giving us a shallow $\phi_{p,q}$ network $q_j^N$ such that

$$\|p_j^N - q_j^N\|_{W^{k,\infty}([0,1]^d)} \leq \eta. \tag{253}$$

Let $\epsilon > 0$ be another parameter to be chosen. By Lemma C.7, we can choose $\alpha = \alpha(N, \epsilon)$ such that

$$\alpha/N \geq R, \quad 1 - \phi(\alpha/N) \leq \epsilon, \quad \text{and} \quad \alpha^m |\phi^{(m)}(\alpha/N)| \leq \epsilon \text{ for all } 1 \leq m \leq k. \tag{254}$$

Moreover, by Lemma C.7, we have the bound

$$\alpha = N \max \left\{ R, \left(\frac{1}{q\epsilon}\right)^{1/p}, \left(\frac{A(p,q,k)N^k}{\epsilon}\right)^{q/p} \right\} \leq C_3(p, q, k, N) \max\{R, 1\} \epsilon^{-q/p}, \tag{255}$$

where we simplify it with the assumption $\epsilon < 1$ and the definition

$$C_3(p, q, k, N) = [A(p, q, k)]^{q/p} N^{kq/p+1}. \tag{256}$$

Define the univariate functions $\rho_j^N$ as in (56) of the original paper, and let

$$\Phi_j^{N,d}(x) = \prod_{i=1}^{d} \rho_{j_i}^N(x_i). \tag{257}$$

By Lemma C.8 and Lemma C.9, we have the estimates:

$$\left\| \sum_{v \in \mathcal{V}_d} \Phi_{j+v}^{N,d} - 1 \right\|_{W^{k,\infty}(I_j^N)} \leq 2^{dk} d\epsilon, \tag{258}$$

and for $v \in \mathbb{Z}^d$ with $\|v\|_\infty \geq 2$,

$$\|\Phi_{j+v}^{N,d}\|_{W^{k,\infty}(I_j^N)} \leq \alpha^k \left(\frac{16p^2}{\pi q}\right)^k k!\epsilon. \tag{259}$$

Let $h > 0$ be a parameter to be chosen. Using Corollary C.2, we construct a shallow $\phi_{p,q}$ network $\widehat{\times}_{d+1}^h$ that approximates the product of $d + 1$ numbers with error $h$ in $W^{k,\infty}$ norm.

Define the global approximation as

$$f(x) = \sum_{j \in \{1,\dots,N\}^d} \widehat{\times}_{d+1}^h \left( q_j^N(x), \Phi_{j_1}^{N,1}(x_1), \dots, \Phi_{j_d}^{N,1}(x_d) \right). \tag{260}$$

The network $f$ has two hidden layers: the first computes all $q_j^N$ and $\rho_j^N$ functions, and the second computes the $N^d$ multiplications.

**Step 2: Estimating the error of the approximation.** Using the triangle inequality, we decompose the error as in (83) of the original paper:

$$\|\hat{f} - f\|_{W^{k,\infty}} \leq E_1 + E_2 + E_3, \tag{261}$$

where

$$E_1 = \left\| \hat{f} - \sum_j \hat{f} \Phi_j^{N,d} \right\|_{W^{k,\infty}},$$

$$E_2 = \left\| \sum_j (\hat{f} - q_j^N) \Phi_j^{N,d} \right\|_{W^{k,\infty}},$$

$$E_3 = \left\| \sum_j \left( q_j^N \Phi_j^{N,d} - \widehat{\times}_{d+1}^h (q_j^N, \Phi_{j_1}^{N,1}, \ldots, \Phi_{j_d}^{N,1}) \right) \right\|_{W^{k,\infty}}.$$

**Step 2a: First term of the error.** We bound $E_1$. Let $i \in \{1, \ldots, N\}^d$ be arbitrary. Using Lemma Appendix A.6 in (De Ryck et al., 2021), Lemma C.8, and Lemma C.9, we obtain for $k \geq 1$:

$$E_1 = \left\| \hat{f} - \sum_j \hat{f} \Phi_j^{N,d} \right\|_{W^{k,\infty}(I_i^N)}$$

$$\leq 2^k \|\hat{f}\|_{W^{k,\infty}(I_i^N)} \left\| 1 - \sum_{v \in \mathcal{V}_d} \Phi_{i+v}^{N,d} \right\|_{W^{k,\infty}(I_i^N)} + 2^k \|\hat{f}\|_{W^{k,\infty}(I_i^N)} \left\| \sum_{\substack{j \in \{1,\ldots,N\}^d \\ \|j-i\|_\infty \geq 2}} \Phi_j^{N,d} \right\|_{W^{k,\infty}(I_i^N)}$$

$$\leq 2^k \|\hat{f}\|_{W^{k,\infty}(I_i^N)} \left( 2^{dk} d\epsilon + N^d \alpha^k \left( \frac{16p^2}{\pi q} \right)^k k! \epsilon \right).$$

One can choose that

$$\epsilon \leq \frac{2 \cdot 3^d C_1 \alpha^k A(p,q,k) \delta}{[2^{dk} d + N^d \alpha^k A(p,q,k)] \|\hat{f}\|_{W^{k,\infty}(I_i^N)}} \leq \frac{2 \cdot 3^d C_1 \delta}{N^d \|\hat{f}\|_{W^{k,\infty}(I_i^N)}} \tag{262}$$

to ensure that

$$E_1 \leq 2^{k+1} 3^d \frac{C_1(d,k,s,\hat{f})}{N^{s-k}} \frac{\delta}{3} \alpha^k A(p,q,k) \tag{263}$$

For $k = 0$, we have the simpler bound

$$E_1 \leq \|\hat{f}\|_{L^\infty(I_i^N)} (d\epsilon + N^d \epsilon) \leq \frac{\delta}{3} \frac{C_1(d,0,s,\hat{f})}{N^s}, \tag{264}$$

if we choose

$$\epsilon \leq \frac{\delta C_1(d,0,s,\hat{f})}{3 \|\hat{f}\|_{L^\infty(I_i^N)} (d + N^d) N^s} \tag{265}$$

**Step 2b: Second term for $k = 0$.** Now we bound $E_2$ for the case $k = 0$. For any $x \in I_i^N$, we have

$$\left| \sum_j (\hat{f}(x) - q_j^N(x)) \Phi_j^{N,d}(x) \right| \leq \sum_{v \in \mathcal{V}_d} |\hat{f}(x) - q_{i+v}^N(x)| |\Phi_{i+v}^{N,d}(x)|$$

$$+ \sum_{\substack{j \in \{1,\ldots,N\}^d \\ \|j-i\|_\infty \geq 2}} |\hat{f}(x) - q_j^N(x)| |\Phi_j^{N,d}(x)|.$$

For the first sum, using Lemma C.8 we have $\sum_{v \in \mathcal{V}_d} |\Phi_{i+v}^{N,d}(x)| \leq 1 + d\epsilon$. Also,

$$|\hat{f}(x) - q_{i+v}^N(x)| \leq |\hat{f}(x) - p_{i+v}^N(x)| + |p_{i+v}^N(x) - q_{i+v}^N(x)| \leq \frac{C_1(d,0,s,\hat{f})}{N^s} + \eta. \tag{266}$$

For the second sum, using Lemma C.9 we have $|\Phi_j^{N,d}(x)| \leq \epsilon$ for $\|j - i\|_\infty \geq 2$, and there are at most $N^d$ such terms. Also,

$$|\hat{f}(x) - q_j^N(x)| \leq \|\hat{f}\|_{L^\infty} + \|q_j^N\|_{L^\infty} \leq \|\hat{f}\|_{L^\infty} + C_1(d,0,s,\hat{f}) + \eta. \tag{267}$$

Combining these estimates, we get

$$E_2 \leq \big(\frac{C_1(d, 0, s, \hat{f})}{N^s} + \eta\big)(1 + d\epsilon) + N^d\big(\|\hat{f}\|_{L^\infty} + C_1(d, 0, s, \hat{f}) + \eta\big)\epsilon. \tag{268}$$

Choose

$$\eta \leq \frac{\delta}{6}\frac{C_1}{N^s}, \epsilon \leq \frac{\delta}{6}\frac{C_1}{N^s}\left[\big(\frac{C_1}{N^s} + \eta\big)d + N^d(\|\hat{f}\|_{L^\infty} + C_1 + \eta)\right]^{-1}, \tag{269}$$

we obtain

$$E_2 \leq (1 + \frac{\delta}{3})\frac{C_1(d, 0, s, \hat{f})}{N^s} \tag{270}$$

**Step 2c: Second term for $k \geq 1$.** Now consider $1 \leq k < s$. Let $\beta \in \mathbb{N}_0^d$ with $|\beta| \leq k$. By the general Leibniz rule,

$$D^\beta\Big(\sum_{v \in \mathcal{V}_d}(\hat{f} - q_{i+v}^N)\Phi_{i+v}^{N,d}\Big) = \sum_{\beta' \leq \beta}\binom{\beta}{\beta'}\sum_{v \in \mathcal{V}_d}D^{\beta'}(\hat{f} - q_{i+v}^N)D^{\beta-\beta'}\Phi_{i+v}^{N,d}. \tag{271}$$

For each $v \in \mathcal{V}_d$ and $\beta' \leq \beta$, let $\ell = |\beta - \beta'|$. Then

$$|D^{\beta'}(\hat{f} - q_{i+v}^N)(x)| \leq \|\hat{f} - q_{i+v}^N\|_{W^{k-\ell,\infty}(I_i^N)} \leq \frac{C_1(d, k - \ell, s, \hat{f})}{N^{s-k+\ell}} + \eta, \tag{272}$$

and by Lemma C.9 and Lemma C.2,

$$|D^{\beta-\beta'}\Phi_{i+v}^{N,d}(x)| \leq \alpha^\ell\left(\frac{16p^2}{\pi q}\right)^\ell \ell!. \tag{273}$$

Since $\sum_{\beta' \leq \beta}\binom{\beta}{\beta'} \leq 2^k$, we obtain

$$\Big\|\sum_{v \in \mathcal{V}_d}(\hat{f} - q_{i+v}^N)\Phi_{i+v}^{N,d}\Big\|_{W^{k,\infty}(I_i^N)} \leq 2^k 3^d\big(\frac{C_1(d, k, s, \hat{f})}{N^{s-k}} + \eta N^k\big)\left(\frac{\alpha}{N}\right)^k\left(\frac{16p^2}{\pi q}\right)^k k! \tag{274}$$

For the terms with $\|j - i\|_\infty \geq 2$, we have by Lemma C.9 and Lemma Appendix A.6,

$$\|(\hat{f} - q_j^N)\Phi_j^{N,d}\|_{W^{k,\infty}(I_i^N)} \leq 2^k\|\hat{f} - q_j^N\|_{W^{k,\infty}(I_i^N)}\|\Phi_j^{N,d}\|_{W^{k,\infty}(I_i^N)} \leq 2^k\big(C_1(d, k, s, \hat{f}) + \eta\big)\alpha^k\left(\frac{16p^2}{\pi q}\right)^k k!\epsilon. \tag{275}$$

There are at most $N^d$ such terms. Assume $\delta \leq 6$, we get

$$\begin{aligned}
E_2 &\leq 2^k 3^d\big(\frac{C_1(d, k, s, \hat{f})}{N^{s-k}} + \eta N^k\big)\left(\frac{\alpha}{N}\right)^k\left(\frac{16p^2}{\pi q}\right)^k k! + N^d 2^k\big(C_1(d, k, s, \hat{f}) + \eta\big)\alpha^k\left(\frac{16p^2}{\pi q}\right)^k k!\epsilon \\
&\leq 2^k 3^d\frac{C_1(d, k, s, \hat{f})}{N^{s-k}}(1 + \frac{\delta}{6})\left(\frac{\alpha}{N}\right)^k\left(\frac{16p^2}{\pi q}\right)^k k! + 2^k N^d\big(C_1(d, k, s, \hat{f}) + \frac{\delta C_1}{6N^s}\big)\alpha^k\left(\frac{16p^2}{\pi q}\right)^k k!\epsilon \\
&\leq 2^k 3^d\frac{C_1(d, k, s, \hat{f})}{N^{s-k}}(1 + \frac{\delta}{6})\left(\frac{\alpha}{N}\right)^k\left(\frac{16p^2}{\pi q}\right)^k k! + 2^k N^d\frac{C_1(d, k, s, \hat{f})}{N^{s-k}}(1 + \frac{\delta}{6N^s})N^{s-k}\alpha^k\left(\frac{16p^2}{\pi q}\right)^k k!\epsilon \\
&\leq 2^k 3^d\frac{C_1(d, k, s, \hat{f})}{N^{s-k}}(1 + \frac{\delta}{3})\left(\frac{\alpha}{N}\right)^k A(p, q, k),
\end{aligned}$$

if we choose that

$$\eta \leq \frac{\delta C_1}{6N^s}, \epsilon \leq \frac{3^d\delta}{12N^{s+d}}. \tag{276}$$

**Step 2d: Third term of the error.** Finally, we bound $E_3$. Using Lemma Appendix A.7, Corollary C.2, and the bounds on $q_j^N$ and $\Phi_j^{N,d}$, we have for some constant $C_k > 0$ depending only on $k$:

$$E_3 \leq N^d C_k (d+1)^d d^{2k} \big\| \widehat{\times}_{d+1}^h - \prod_{i=1}^{d+1} x_i \big\|_{W^{k,\infty}}$$

$$\times \big( \|\hat{f}\|_{W^{k,\infty}} + \frac{C_1(d,k,s,\hat{f})}{N^{s-k}} + \eta + \|\rho_j^N\|_{W^{k,\infty}} \big)^k \tag{277}$$

$$\leq N^d C_k (d+1)^d d^{2k} h \big( \|\hat{f}\|_{W^{k,\infty}} + \frac{C_1(d,k,s,\hat{f})}{N^{s-k}} + \eta + \alpha^k A(p,q,k) \big)^k.$$

One can choose

$$h \leq \frac{C_1(d,k,s,\hat{f})\alpha^k A(p,q,k)}{N^{s-k} N^d C_k (d+1)^d d^{2k} \big( \|\hat{f}\|_{W^{k,\infty}} + \frac{C_1(d,k,s,\hat{f})}{N^{s-k}} + \eta + \alpha^k A(p,q,k) \big)^k} \frac{\delta}{3} \tag{278}$$

to ensure that

$$E_3 \leq \frac{C_1(d,k,s,\hat{f})}{N^{s-k}} \frac{\delta}{3} \alpha^k A(p,q,k). \tag{279}$$

And for $k = 0$, we obtain

$$E_3 \leq \frac{\delta}{3} \frac{C_1(d,k,s,\hat{f})}{N^{s-k}}. \tag{280}$$

**Step 2e: Final error bound.**

For $k = 0$, by Eqn.264, Eqn.269 and Eqn.280, we obtain

$$\|\hat{f} - f\|_{W^{k,\infty}} \leq (1+\delta)\frac{C_1(d,0,s,\hat{f})}{N^s}, \tag{281}$$

with suitable $\epsilon, \eta$ and $h$.

For $k > 0$, by Eqn.263, Eqn.275 and Eqn.279, we choose

$$\eta = \frac{\delta C_1}{3N^s}, \epsilon = \min\big( \frac{2 \cdot 3^d C_1 \delta}{N^d \|\hat{f}\|_{W^{k,\infty}(I)}}, \frac{3^d \delta}{12N^{s+d}} \big) = C_4 \delta N^{-(s+d)}, \tag{282}$$

and take $h$ as the value in Eqn.278

$$\|\hat{f} - f\|_{W^{k,\infty}} \leq 2^{k+1} 3^d \frac{C_1(d,k,s,\hat{f})}{N^{s-k}} (1+\delta) \big( \frac{\alpha}{N} \big)^k A(p,q,k)$$

$$\leq (1+\delta) 2^{k+1} 3^d [A(p,q,k)]^{kq/p+1} N^{k^2 q/p} \max\{R^k, 1\} \epsilon^{-kq/p} \frac{C_1(d,k,s,\hat{f})}{N^{s-k}} \tag{283}$$

$$\leq C(p,q,d,k,s,\hat{f}) \frac{1+\delta}{\delta^{kq/p} N^{s-k-(s+d)kq/p-k^2 q/p}},$$

where

$$C(p,q,d,k,s,\hat{f}) = 2^{k+1} 3^d \left[ (\frac{4p}{\pi q})^k k! \right]^{kq/p+1} \max\{R^k, 1\} C_4^{-kq/p} C_1(d,k,s,\hat{f}), \tag{284}$$

and note that $R$ is decided by $p$ and $q$.

To be figured out, the right side $\to 0$ when $N \to \infty$ requires that $s - k - (s+d)kq/p - k^2 q/p > 0$, thus

$$p/q > \max\big( \frac{k(s+d+k)}{s-k}, k \big) = \frac{k(s+d+k)}{s-k} \tag{285}$$

is the necessary condition in this proof.

**Step 3: Network sizes.** The width analysis is the same as that in the original paper, but just replace with our Corollary C.1 and Corollary C.2.

$\square$

# D. Experiments

## D.1. Differential Equation Solving Experiments

To verify the derivative representation ability of PLN, we solve a one-dimensional Helmholtz equation within the PINN framework. The Helmholtz equation is the time-independent counterpart of the wave equation and appears in a wide range of physical settings, including vibrating membranes, acoustics, and electromagnetism. Let the neural network $u_\theta(x)$ approximate the ground-truth solution. We use automatic differentiation, we compute the second derivative $u_\theta''(x)$ and construct the PDE residual

$$r_\theta(x) = -u_\theta''(x) + \pi^2 u_\theta(x) - 2\pi^2 \sin(\pi x). \tag{286}$$

The ground-truth solution is given by $u(x) = \sin(\pi x)$. On the interval $\Omega = [-1, 1]$, we sample 5,000 interior points and 500 boundary points, and jointly minimize the mean squared error of the interior PDE residual and the Dirichlet boundary-condition error, with weighting via loss weights $\lambda_{PDE}, \lambda_{BC}$. This yields an approximate solution that satisfies both the governing equation and the boundary constraints. Because this task explicitly depends on second-order derivative information (i.e., curvature constraints), it provides a suitable testbed for comparing different network configurations in terms of their ability to capture higher-order behavior, thereby assessing the high-order expressivity.

We chose the activation layer among ReLU, tanh, sat and PLN, which varies the feature number per group from $4, 8, 16, 32$. We sweep loss weight $(\lambda_{PDE}, \lambda_{BC}) \in \{(1, 10), (1, 1), (3, 1)\}$. We sweep the learning rate and choose $3 \times 10^{-4}$ for Adam optimizer and $3 \times 10^{-4}$ for SGD optimizer with a step optimizer. We use relative $\mathcal{L}_2$ error between the predicted and the exact solution $u(t, x)$ to validate the global fitting quality over the domain. The activation function with best relative $\mathcal{L}_2$ error is marked in bold, the second best is underlined. We list the result of activation layer under different width and loss weight in Table I.

*Table I.* We record the best relative $\mathcal{L}_2$ error of different activation layers under different experiment settings on solving Helmholtz equation.

| Width | Loss Weight | ReLU | PLN4 | PLN8 | PLN16 | PLN32 | Tanh | SAT |
|-------|-------------|------|------|------|-------|-------|------|-----|
| | $[1, 10]$ | 0.999601 | **0.000035** | 0.001120 | 0.001176 | 0.007878 | 0.000250 | 0.000104 |
| 128 | $[1, 1]$ | 0.995612 | **0.000941** | 0.002284 | 0.000970 | 0.004231 | 0.003733 | 0.000979 |
| | $[3, 1]$ | 0.993593 | 0.009553 | 0.003912 | 0.004856 | 0.007465 | **0.000440** | 0.000525 |
| | $[1, 10]$ | 0.998639 | **0.000036** | 0.004040 | 0.001631 | 0.000567 | 0.000188 | 0.000052 |
| 256 | $[1, 1]$ | 0.998485 | 0.001416 | 0.005552 | 0.001839 | 0.002643 | 0.002653 | **0.001219** |
| | $[3, 1]$ | 0.999727 | 0.018451 | 0.002966 | 0.002875 | 0.005476 | 0.009267 | **0.000410** |

## D.2. Classification Experiments

We conduct the classification experiments on CIFAR-10 dataset (Krizhevsky et al., 2009) using VGG-16 (Simonyan & Zisserman, 2014). For each hyperparameter configuration, we perform three independent runs with different random seeds and report the mean validation accuracy. We sweep the following ranges: activation functions [PLN(8), ReLU, SiLU, tanh, sat], learning rates $[1, 0.3, 0.1, 0.03, 0.01, 0.003, 0.001]$, initialization scaling $[0.25, 0.35, 0.5, 0.7, 1.0, 1.4, 2.0, 2.8, 4.0]$. The scale 1.0 means using the He initialization (He et al., 2015), and 4.0 means initializing the weights four times of that of He initialization. We train a total of 200 epochs using SGD (Amari, 1993) with a mini-batch size of 256, momentum of 0.9 and weight decay of 0.0001. The initial learning rate is chosen above, and divided by 2.5 at the 50th, 90th, 130th, and 170th epochs. We use warmup in the first 10 epochs. We also use data augmentation. We record the average results among three random seeds. We use warmup in the first 10 epochs. We also use data augmentation. We record the average results among three random seeds.

## D.3. Experiment Settings of the Translation Task

We conduct experiments on Transformers (Vaswani et al., 2017a) for Multi30k (Elliott et al., 2016). In the translation task training process, each task is trained for 100 epochs, with the first 10 epochs utilizing a warmup strategy and the remaining 90 epochs following a cosine decay learning rate schedule. The maximum learning rate is set to 5e-4, and the optimizer used is Adam with a weight decay of 5e-4. Each task is conducted using three different random seeds (10, 20, and 30), and the final results are averaged.

