# OpenReview forum: "Parallel Layer Normalization for Universal Approximation"
_ICML.cc/2026/Conference — Submitted to ICML 2026_

### Official Review · Reviewer_LJUV · 2026-02-21

**Soundness:** 2
**Presentation:** 2
**Significance:** 1
**Originality:** 2
**Overall Recommendation:** 3
**Confidence:** 4

**Summary:**

This paper analyzes the approximation capability of layer normalization and parallel
layer normalizations. It first proves that there exists a function that can not be approximated by shallow LN networks. On the other hand, shallow PLNs have the universal approximation property in both the shallow and deep narrow cases. The proof strategy is straightforward. First, approximate a sign or $x/\sqrt{x^2+1}$ function using a finite number of neurons.

**Compliance With Llm Reviewing Policy:**

Affirmed.

**Final Justification:**

This paper attempted to prove the Unified Application of Layer Normalization (UAP), but while it demonstrated that it cannot be achieved in shallow layers, it failed to prove it in deep layers. The authors did not provide a sufficient answer regarding this. Therefore, I believe that weak reject is appropriate.

**Key Questions For Authors:**

Could the authors address the issues mentioned?

**Limitations:**

Yes

**Strengths And Weaknesses:**

Strength.



The claim of the paper is solid, and I could not find any flaws in the paper although I didn’t check all the details.



Weakness


1. Overall, the contribution of the paper is marginal, and the proof is a direct consequence of the existing result with a slight modification.

2. Although the paper discusses the universal approximation property of the deep narrative MLP, the minimum width provided in the paper is rough, about three times the input.

3. There is no analysis of whether LN has the universal approximation property in the deep narrow MLP setting.

4. The choice of the PLN network is artificial. I don’t think Remark 3.3 is sufficient for its justification.

5. The explanations in the captions of figures and tables are not self-contained and are insufficient.

---

> ### Author Rebuttal · Authors · 2026-03-31
>
> ## Reply to the Reviewer LJUV
>
> ### Response to Weakness 1:
>
> We have discussed the background in detail in the Introduction. To the best of our knowledge, no prior work has studied universal approximation theorems (UAT) **with normalization**, and our paper conducts the first investigation on this topic. To avoid any misunderstanding, we clarify our contributions as follows.
>
> We study the UAT results of LN/PLN‑Nets to fill the gap in understanding the nonlinearity of LayerNorm. Our theoretical results may further reveal the potential of PLN‑Nets, either as standalone models or as substructures in larger architectures. While these results may not be surprising in the UAT field, they are important for the theoretical understanding of the nonlinearity introduced by LN.
>
> Regarding theoretical contributions, all results in Section 4 are original. They discuss **UAT with LayerNorm** for the first time. Moreover, only Theorems 5.1 and 5.2 can be directly deduced from prior work, based on our Lemma 5.1. We devote no more than two pages in the Appendix to their proofs, so that readers do not need to revisit known proofs. For Theorems 5.3 and 5.5, we follow the proof structure of [1], but the differences between tanh and $\phi(x)=x/\sqrt{x^2+1}$ required substantial additional work (Pages 27–44) to bridge the gap. These pages contain only necessary lemmas and exclude those that can be directly derived from existing work. Therefore, we emphasize that these pages are **not a direct reproduction** but required significant effort. In Remark 5.5, we also discuss the source of the differences and why additional lemmas were necessary to complete the proof for $\phi(x)=x/\sqrt{x^2+1}$.
>
> Overall, almost all results in this manuscript are new to the field, rather than straightforward consequences of existing results with minor modifications. We reiterate that **no previous work has studied UAT with normalization**; our work is the first to do so.
>
> ---
>
> ### Response to Weakness 2:
>
> We would like to point out that the minimum width is not the main focus of this manuscript. Our aim is to establish the existence of UAT for LN/PLN‑Nets. We are the first to investigate UAT for networks with normalization. While the minimum width is an interesting question, we believe it should be addressed in future work.
>
> ---
>
> ### Response to Weakness 3:
>
> We have indeed considered the UAT of deep LN‑Nets, but we did not include them in the manuscript because the results are preliminary. We believe the representation capacity of deep LN-Nets are possibly weak, but not find the evidence yet. If the reviewer is interested, we provide the following preliminary findings:
>
> - We can prove that: Let $\Omega=\{x_i\in\mathbb{R}^d:1\le i\le m\}$ and let $\hat{f}:\Omega\to\{0,1\}$ be a binary‑valued function. Then there exists an LN‑Net $f\in\mathcal{F}(\mathrm{LN}(n_s);m,n_s)$ with width $n_s\ge3$ such that $\|f-\hat{f}\|_{L^\infty(\Omega)}=0$.
>
> - Although deep LN‑Nets have universal classification capacity on finite datasets, whether they admit universal approximation over continuous input domains remains unknown. By Lemma 5.1, LN‑Nets can represent certain function classes of the form $\mathcal{F}(\phi;L,1)$ (i.e., width 1 and depth $L$). However, [2] shows that most activation functions require width at least 2 for universal approximation. This gap makes the approximation power of deep LN‑Nets on dense domains an open and challenging theoretical problem, which we leave for future work.
>
> ---
>
> ### Response to Weakness 4:
>
> The choice of PLN‑Nets is not artificial. This architecture has been studied in [3] and has been shown to outperform GroupNorm. PLN‑Nets are also substructures of some recent network architectures like Mamba-2 and RetNet (noted as [1] and [2] in our reply to the Reviewer pnmS).
>
> We suspect that a misunderstanding may have arisen from our writing. In Remark 3.3 (lines 141–157), the reviewer may have overlooked the second paragraph. We also kindly invite the reviewer to refer to **“Clarification on the comparison of GN and PLN”** in our reply to Reviewer  WmME for a more detailed comparison. If these points are still insufficient, we draw the reviewer’s attention to Theorem 5.4 (UAT of position‑wise FFNs) and its proof in Appendix B.5, which indicate the necessity of our definition of PLN.
>
> ---
>
> Should there be any remaining questions, please feel free to raise them.
>
> ---
>
> ### References
>
> [1] De Ryck, T., Lanthaler, S., and Mishra, S. On the approx imation of functions by tanh neural networks. Neural Networks, 143:732–750, 2021.
>
> [2] De Ryck T, Lanthaler S, Mishra S. On the approximation of functions by tanh neural networks[J]. Neural Networks, 2021, 143: 732-750.
>
> [3] Ni Y, Guo Y, Jia J, et al. On the nonlinearity of layer normalization[J]. arXiv preprint arXiv:2406.01255, 2024.

---

> > ### Author Rebuttal · Reviewer_LJUV · 2026-04-02
> >
> > My comment regarding the UAP of deep layer normalization was not adequately addressed. However, since other issues have been partially resolved, I will raise the score to 3.

---

> > > ### Author Response · Authors · 2026-04-07
> > >
> > > We thank the reviewer for raising the score to 3. Regarding the UAP of deep (sequential) LN-Nets: we respectfully note that **this is not the main focus of our paper**, which studies **Parallel LN (PLN)** as indicated by the title. We have stated in our manuscript (lines 114–119) that we adopt the parallel structure specifically to realize UAT, rather than relying on increased depth. The question of UAT has already been answered in our manuscript using PLN. A discussion of the UAT of deep LN‑Nets is not necessary for this manuscript, as its title focuses on "Parallel Layer Normalization" rather than "Layer Normalization" alone. More importantly, **to the best of our knowledge, this question itself is open — whether deep sequential LN-Nets are universal approximators at all is not known, let alone how to prove it**. We have also discussed the potential difficulty of studying the UAT of deep LN-Nets in the first round. A proper treatment of this open problem is beyond the scope of the current work. We have noted it as an interesting future direction.

---

### Official Review · Reviewer_NgLS · 2026-03-04

**Soundness:** 2
**Presentation:** 2
**Significance:** 2
**Originality:** 3
**Overall Recommendation:** 3
**Confidence:** 4

**Summary:**

This paper investigates the universal approximation properties of neural network architectures that utilize Layer Normalization (LN) and Parallel Layer Normalization (PLN). The focus is on understanding how these normalization layers influence the networks' expressive power. A key contribution of this study is the distinction between the limited approximation capabilities of shallow LN-Nets and the enhanced universal approximation properties of shallow PLN-Nets. Additionally, the paper presents a comprehensive theoretical framework that includes univariate and Sobolev-space approximation results, deep PLN constructions, equivalence results for RMSNorm/LS-style variants, representations of position-wise feed-forward networks, and initial results on derivative approximations under generalized $(p,q)$-normalization.

**Compliance With Llm Reviewing Policy:**

Affirmed.

**Key Questions For Authors:**

How sensitive are the theory and empirical outcomes to norm size ($n_s$), grouping strategy, and learnable affine parameters in Layer Normalization (LN) and Probabilistic Layer Normalization (PLN)? How do these outcomes change with the standard $\epsilon$-stabilized LayerNorm definition that includes learnable gain and bias? Which theorem statements are fundamentally new contributions, and which are mostly corollaries derived from existing approximation theorems?

**Limitations:**

The paper needs a comprehensive review to improve the clarity of its presentation.

**Strengths And Weaknesses:**

Paper Strengths:
The paper tackles an important gap in the literature: many analyses of universal approximation overlook normalization layers, despite their prevalence in modern neural architectures. Therefore, exploring approximation properties when normalization is included is a valid and valuable direction for research. The paper draws a clear distinction between two settings: shallow LN-Nets are shown to possess limited expressive power, while shallow PLN-Nets successfully achieve universal approximation.

Weaknesses:
1. PLN is conceptually related to LN-G and GroupNorm, but the paper provides limited formal comparisons of expressivity or optimization behavior against alternatives like GroupNorm or RMSNorm, aside from the LS-equivalence discussion.
2. The theory excludes learnable affine parameters in Layer Normalization (LN) for simplicity and adopts a custom convention for the case when $\sigma = 0$, setting $\mathrm{LN}(h)=0$ unless an $\epsilon$-stabilized alternative is used. While these choices are reasonable for analysis, they significantly impact practical implementation. The manuscript notes these choices but does not sufficiently explore their real-world implications.
3. In the experiments, PLN is compared with activation functions like ReLU, SiLU, Tanh, and SAT, as well as serving as a substitute for Layer Normalization in transformer architectures. However, these comparisons often do not isolate the effects of PLN from other architectural or hyperparameter variations.
4. How should practitioners interpret PLN relative to standard transformer feed-forward blocks that already combine activation, normalization, and residual structure? Do you have evidence beyond the current small-scale Multi30k and VGG experiments that PLN is
beneficial in modern architectures?
5. The paper has writing issues, including examples such as “feed-froward” instead of “feed-forward,” awkward phrasing in some theorem discussions, and inconsistent capitalization/style choice.

---

> ### Author Rebuttal · Authors · 2026-03-31
>
> ## Reply to the Reviewer NgLS
>
> We thank the reviewer for the valuable suggestions and insightful questions.
>
> ---
>
> ### Response to Weakness 1 (comparison of PLN, GN and RMSNorm)
>
> We have shown that RMSNorm is equivalent to LS (lines 247–250) in the origin manuscript. For the comparison between PLN and GN, we briefly discuss their differences in Remark 3.3 (lines 147–157), and we kindly invite the reviewer to refer to **“Clarification on the comparison of GN and PLN”** in our reply to Reviewer WmME for a more detailed discussion, including why we adopt PLN instead of GN.
>
> ---
>
> ### Response to the comparison of PLN settings
>
> **Norm sizes and affine parameters.**
> Theoretically, too large a norm size reduces representation capacity. For MLPs, when the norm size reaches the full width, PLN degenerates to LN, and a shallow LN‑Net has limited capacity (Theorem 4.1). Conversely, too small a norm size can hinder training. For example, with $\epsilon=0$ and $n_s=2$, PLN outputs are in $\{-1,0,1\}$ and gradients w.r.t. input vanish almost everywhere, effectively freezing parameters under gradient descent methods.
>
> Affine parameters do not affect theoretical representation capacity (they can be absorbed into the following linear layer) but may influence optimization. In our experiments, we use affine parameters by default. We conducted an ablation on VGG (learning rate 0.01, init factor 1.0, other settings unchanged). The table below reports average test accuracy (%) over three seeds. We find that removing affine parameters slightly reduces accuracy.
>
> | Norm Size | 2     | 4     | 8     | 16    | 32    | 64    |
> | --------- | ----- | ----- | ----- | ----- | ----- | ----- |
> | Affine    | 10.00 | 85.8  | 88.14 | 88.64 | 88.71 | 88.48 |
> | No Affine | 10.00 | 84.57 | 87.97 | 88.42 | 88.60 | 88.29 |
>
> For CNNs, large norm sizes do not significantly harm performance because PLN does not degenerate to LN; however, prior work has identified weak representation of LN and GN (see our reply to Reviewer WmME). For MLPs, our PINN experiments (Appendix D.1) show that larger norm sizes tend to increase errors, and toy experiments in the reply to the Reviewer WmME also show that LN-Nets possess weak representation capacities.
>
> **Small $\epsilon$ in the denominator.**
> We discussed this in Remark 4.1 (lines 237–244) and Appendix A.8. In our experiments, we used $\epsilon=1e-5$ for numerical stability; setting $\epsilon=0$ leads to instability.
>
> We will add these setting details in the revised manuscript.
>
> ---
>
> ### Response to our contributions
>
> All theoretical results in Section 4 are original, with detailed proofs in the appendix. We discussed the relationship of LN/PLN-Nets with conventional $\phi$-Nets in detail for the first time. In Section 5, only Theorems 5.1 and 5.2 are directly deducible from previous work, but they necessarily rely on our original Lemma 5.1. We devote no more than two pages to their proofs. For Theorems 5.3 and 5.5, we follow the proof structure of [1], but the differences between tanh and $\phi(x)=x/\sqrt{x^2+1}$ require substantial additional lemmas (Appendix C), as explained in Remark 5.5. We believe this reorganization is necessary.
>
> More importantly, our main contribution is to supplement the understanding of the nonlinearity of LN, rather than simply constructing new universal approximators. Although LN/LS is widely used in current models, especially LLMs, no one had previously considered the approximation capacity of LNs. Some existing works [2] [3] simplify LN as a linear layer, which may leave gaps; we believe our work provides a more precise refinement and offers new insights.
>
> ---
>
> ### Response to Weakness 3
>
> When comparing PLN with other nonlinear modules, the only difference is the module itself; all other settings are identical. We compare PLN with both LN and activation layers in different experiments because PLN can serve either role.
>
> ---
>
> ### Response to Weakness 4
>
> Our PLN with fixed norm sizes has been (equivalently) advocated in certain new architectures, e.g., Mamba‑2 and RetNet, where standard self‑attention is replaced by generalized linear attention. They call the applied layers "GroupNorm", which are equivalent to PLN in implementation with appropriate hyperparameters, rather than the standard GroupNorm in CNNs.
>
> ---
>
> If further questions remain, we are happy to provide additional discussion.
>
> ---
>
> ### References
>
> [1] De Ryck T, Lanthaler S, Mishra S. On the approximation of functions by tanh neural networks[J]. Neural Networks, 2021, 143: 732-750.
>
> [2] Yun C, Bhojanapalli S, Rawat A S, et al. Are transformers universal approximators of sequence-to-sequence functions?[J]. arXiv preprint arXiv:1912.10077, 2019.
>
> [3] Dunefsky J, Chlenski P, Nanda N. Transcoders find interpretable llm feature circuits[J]. Advances in Neural Information Processing Systems, 2024, 37: 24375-24410.

---

> > ### Author Rebuttal · Reviewer_NgLS · 2026-04-01
> >
> > Thank you to the authors for their detailed and thoughtful rebuttal. However, I encourage the authors to revise and resubmit.
> >
> > - The proof in Appendix C establishes that standard PLN (p=q=2) cannot approximate derivatives in the Sobolev sense. Why then does Section 6 claim PLN(4) demonstrates "derivative representation ability" in PINNs? Is the PINN advantage due to optimization/conditioning rather than approximation theory? If PLN's PINN advantage is optimization-based rather than approximation-based, the entire framing of Section 6 and its connection to Theorem 5.5 needs to be rewritten. What is the actual theoretical justification for the PINN experiment? Would PLN with p=6, q=2 (r=3) — which satisfies the theorem conditions for k=1 — perform better on PINN tasks? Have the authors tested this?
> >
> > -  Is there a practical regime of (d, s, k, p, q) where the derivative approximation rate of Theorem 5.5 gives useful convergence (i.e., the exponent is meaningfully positive) while remaining computationally feasible?
> > In Appendix C.5, Equation 283, the exponent s-k-(s+d+k)kq/p must be positive for convergence. This gives p/q > k(s+d+k)/(s-k). For PINNs with typical d=2,3, s=4, k=2, what is the minimum p/q required? Is such a (p,q)-normalization practically implementable? The bound in Lemma C.9 (Equation 240) contains α^k·ε, where α grows as N^(1+kq/p)·ε^(-q/p) per Lemma C.7. For the final bound in Step 2e to converge, α^k·ε ∼ ε^(1-kq/p) → 0 requires k < p/q. But the theorem condition is r=p/q > k(s+d+k)/(s-k) which is stronger than k < p/q. Is the extra strength in the condition entirely driven by Lemma C.9, or are there other places in the proof requiring it?
> >
> > - If PLN is allowed to have variable group sizes (some groups of size 2, others larger), could the total width be reduced below nsN while maintaining the same approximation error? Is equal group size necessary for any theorem in the paper?
> > Looking at Equation 32 in Appendix A.1: the construction wastes ns-2 neurons per group as zero-padded entries. Why not use ns=2 universally in Theorem 4.2? The theorem states ns≥2; is ns=2 actually sufficient for the entire proof chain?
> > Is there a matching lower bound showing that any PLN-Net approximating an L-Lipschitz function to error ε requires Ω(L/ε) total neurons? Without this, the approximation rate cannot be compared to standard activation networks.
> >
> > - In Remark 5.2, you state "PLN groups ns neurons within each token." But in Corollary 4.1's construction (Appendix A.6), the grouping is determined by the weight matrix W₁ through the parameter stacking in Equation 79. After applying φ₁ in Theorem 5.4, do the ns neurons in each PLN group still correspond to meaningful "within-token" features, or could the φ₁ transformation mix features across what were originally different token dimensions?
> > The Mamba-2 and RetNet connection mentioned in the rebuttal — can you show this formally? Specifically, what is the norm size ns and grouping strategy used in Mamba-2's GroupNorm, and does Theorem 5.4 apply directly, or does it require modification?

---

> > > ### Author Response · Authors · 2026-04-07
> > >
> > > To begin with, we note that the questions raised in the first round are not raised again in this round. The questions in the second round are new and not directly related to the first round. We believe that **the reviewer's concerns in the first round have been addressed** in our previous rebuttal.
> > >
> > > Now we reply to the questions in the second round.
> > >
> > > ---
> > >
> > > ### Response to Q3
> > >
> > > **Why use $n_s\ge2$?:** $n_s=2$ is sufficient for Theorem 4.2 in theory, as the construction in Lemma 4.1 shows. However, we present the theorem for general $n_s \ge 2$ because larger $n_s$ (e.g., standard LN corresponds to $n_s = d$) are more common in practice.
> > >
> > > **Variable group sizes:** We assume equal group sizes for simplicity (line 139-140), but this is not a strict requirement. Even if group sizes differ, each group can still implement the $n_s=2$ construction (as in Eqn. 32) by ignoring extra neurons.
> > >
> > > **Lower bound:** We find that $\mathcal{F}(\mathrm{PLN}(2);1,2N)$ degenerates to $\mathcal{F}(\text{sign};1,N)$, which requires at least $\Omega(L/\epsilon)$ neurons. As for $n_s>2$, the analysis is more involved; we leave it as future work. **The main difficulty is that each LN piece is a multivariate mapping, which is more complex than traditional element-wise activation functions.**
> > >
> > > ---
> > >
> > > ### Response to Q4
> > >
> > > $\varphi_1$ does **not** mix features across different tokens. In position-wise FFNs, the same affine transformation is applied independently to each token — the sequence (token) dimension behaves analogously to the batch dimension in standard MLPs. Hence, the propagation of different tokens is independent, which is why the conclusions from the MLP setting extend directly. A detailed proof is provided in Appendix B.5.
> > >
> > > Regarding Mamba-2 and RetNet: Their "GroupNorm" also operates **within each token** independently; different tokens do not interact. GroupNorm uses the number of groups $g$ as a hyperparameter, while PLN uses the norm size $n_s$. For a token dimension $d$, GroupNorm with $g$ groups is equivalent to PLN with $n_s = d/g$. Therefore, Theorem 5.4 applies directly to both architectures.
> > >
> > > ---
> > >
> > > ### Response to Q1 and Q2
> > >
> > > To begin with, we highlight that **the results on derivative approximations are preliminary** (mentioned in line 343, left column). **The exploratory experiments are also preliminary** (mentioned in line 376-378, right column). Regarding the motivation of Section 6, we stated that we aim to explore the potential of PLN-Nets (lines 373-378, right column), as such networks are not yet widely used in current applications.
> > >
> > > - **Condition about $p/q$:** Appendix C **did not claim** that standard PLN ($p=q=2$) cannot approximate derivatives in the Sobolev sense. The bound $p/q > k(s+d+k)/(s-k)$ in Theorem 5.5 is **sufficient but not necessary**. This condition is primarily driven by Lemma C.9, as discussed in Remark 5.5, which also marks the main difference from the $\tanh$ case.
> > > - **Practical usage of $(d,s,k,p,q)$:** Because the condition in Theorem 5.5 is sufficient but not necessary, the fact that standard PLN ($p=q=2$) does **not** satisfy it does **not** imply that it cannot approximate derivatives in the Sobolev sense. In practice, we believe $p=q=2$ is a reasonable default choice, balancing expressivity and optimization. Hence, we consider the PINN experiment settings to be reasonable. We will revise the manuscript to clarify the gap between theory and practice in the final version.
> > >
> > > - **Performance of the case $(p,q)=(6,2)$:** We further conducted PINN experiments for the generalized setting $(p,q)=(6,2)$. The results show that (6,2)-PLN(4) (namely using$(p,q)=(6,2),n_s=4$) still underperforms (2,2)-PLN(4), SAT, and tanh in terms of final test accuracy, although it remains better than ReLU: its relative L2 error is 7.28e−3, whereas the best results of (2,2)-PLN(4), SAT, tanh, and ReLU are 3.6e−5, 5.2e−5, 1.88e−4, and 9.98e−1 (as shown in Table 1), respectively. Therefore, on the current PINN task, the generalized setting $(p,q)=(6,2)$, although satisfying the condition in Theorem 5.5, does not lead to better final empirical performance. This indicates that our sufficient condition is not necessary. (2,2)-PLN(4) works well in practice even without theoretical guarantees.

---

### Official Review · Reviewer_pnmS · 2026-03-10

**Soundness:** 4
**Presentation:** 4
**Significance:** 2
**Originality:** 3
**Overall Recommendation:** 3
**Confidence:** 4

**Summary:**

This paper studies approximation properties of neural networks including forms of layer normalization.
The main theoretical result is that shallow neural networks with layer normalization are limited in expressivity, while adding parallel layer normalization (PLN) facilitates universal approximation.
Concretely, the paper first shows that a shallow LN-Net can realize certain nontrivial univariate nonlinearities, including a sign-type unit or smoother variants, while remaining limited overall.
It then uses the observation that a parallel layer normalization is essentially a sum of shallow neural networks with layer normalization to derive a univariate universal approximation theorem.
This result is then further expanded to obtain multivariate Sobolev-type approximation guarantees, deep-network extensions, and variants for RMSNorm and position-wise FFNs in sequence models. The paper also includes exploratory experiments in PINNs, VGGs, and Transformers.

**Compliance With Llm Reviewing Policy:**

Affirmed.

**Final Justification:**

I have explained as a response to the AC why I would like to keep my original score.

**Key Questions For Authors:**

I think formulating reasonable questions in this situation is hard because I do not have any technical issues with the manuscript. In fact, I believe it is a good manuscript. I simply feel that the contribution is not deep enough for a top venue. Nonetheless, my attempts are below:

1. I concede that I may have overlooked some difficulties, so I would invite the authors to clarify what they view as the main conceptual difficulty beyond the ideas that I have repeated already above.

2. Can the authors identify an approximation result where PNL networks outperform classical multilayer perceptrons in a significant way? Such a result, would necessarily not just be a transfer result from classical approximation theory.

**Limitations:**

yes

**Strengths And Weaknesses:**

I believe that a) the topic is very relevant. Layer normalization plays an increasing role in modern architectures and understanding its role is essential. b) I think that the mathematical arguments are correct. c) The core idea is very well explained. The paper identifies a simple but useful observation: a width-2 shallow neural network with layer normalization can already realize a nontrivial univariate nonlinearity. Using this non-linearity essentially as an activation function then yields all sorts of universal approximation theorems. d) The paper has an appropriate scope in the sense that it does not stop at a single univariate universal approximation theorem, but extends the discussion to Sobolev approximation, deep PLN-Nets, RMSNorm, and position-wise FFNs, and it includes a reasonably broad exploratory experimental section.

That being said, I have some reservations about the manuscript. First on significance: It could be that I have seen too many universal approximation theorem-type papers, which means that I find it hard to get excited about these ideas. If the consensus of the other reviewers and the AC is that these type of results are very exciting, then they should focus on the positive statements above. I would not assign too much practical relevance to a universality result for neural network architectures, that were, to put it a bit strongly, essentially designed to achieve this property.
In addition, I am hesitant when it comes to the depth and contribution of this work. depth of the contribution. The central idea is nice, but a large portion of the paper seems to boil down to this one key observation: a small, shallow layer normalization neural network already produces a nonlinear univariate function, and once that function can serve as an activation surrogate, standard universality arguments follow. This makes the core theoretical contribution feel more like a clever representability trick than a substantially new approximation-theoretic development.
Relatedly, Theorem 4.1, while useful and appropriate in the paper, appears mathematically fairly straightforward: once one computes the structure of shallow LN-Net outputs, the “at most one stationary point” property follows by a direct argument. So even the negative side of the theory, though helpful for framing, did not strike me as especially deep.

Overall, I am not fully convinced that the contribution is deep enough for a top venue. The paper contains a neat and well-executed idea, and I do think the results are interesting, but from my perspective, most of the arguments reduce to one elegant construction from which the main consequences then follow rather directly. In that sense, I see the work more as a clean representability observation with several corollaries than as a major theoretical advance.

---

> ### Author Rebuttal · Authors · 2026-03-31
>
> ## Reply to the Reviewer pnmS
>
> We thank the reviewer for the valuable suggestions and insightful questions.
>
> ---
>
> ### Clarification on our contributions
>
> We agree that our findings may not be groundbreaking within the UAT field, but they serve as an important supplement to the theoretical understanding of the nonlinearity of layer normalization. We would like to clarify that our main contribution is **not further exploration in UAT per se, but rather a deeper analysis of layer normalization**. Our experiments reveal some potential of PLN, even though PLN is not as commonly used as LN. Notably, PLN$(n_s)$ has been advocated (equivalently) in certain recent network architectures, such as Mamba‑2 [1] and RetNet [2], which employ generalized linear attention rather than standard self‑attention. Moreover, what they call "GroupNorm" is in fact equivalent to PLN in implementation under appropriate hyperparameter settings, rather than the standard GroupNorm used in CNNs [3]. Our results provide a theoretical advantage of PLN over LN, and we believe they may help investigate the factors that limit the adoption of PLN and further improve architectural design.
>
> ---
>
> ### Response to Weakness 1 (the difficulty of this paper)
>
> Unlike prior UAT work, our paper considers normalization layers for the first time. To the best of our knowledge, **no existing work has studied UAT with normalization**.
>
> One difficulty—or perhaps a novel and important question—is: what does a network with normalization actually represent? Specifically, how can we transfer a network with normalization into a more familiar network architecture? Take Theorem 4.1 (mentioned by the reviewer) as an example. While the theorem itself may appear simple and its proof straightforward, we emphasize that **finding this theorem was not easy**. The ease of proving a theorem does not imply that the theorem is easy to discover. Although LN/LS is widely used in current models, especially LLMs, no one had previously considered this question.
>
> Furthermore, only Theorem 5.1 and Theorem 5.2 can be directly deduced from prior work, based on our Lemma 5.1. We devote no more than two pages in the Appendix to their proofs, so that readers do not need to revisit known results. For Theorem 5.3 and Theorem 5.5, we follow the proof structure of [4], but the differences between tanh and $\phi(x)=x/\sqrt{x^2+1}$ required substantial additional work (Pages 27–44) to bridge the gap. These pages contain only the necessary lemmas and exclude those that can be directly derived from existing work. In Remark 5.5, we also discuss the source of these differences and why additional lemmas were necessary.
>
> The above arguments highlight the mathematical challenges we addressed. However, we believe that the more important contribution lies in the novelty of our theoretical results within the normalization field. While these results may not be impressive from a pure UAT perspective, they are significant for understanding the nonlinearity of LN. Some existing works [5] [6] simplify LN as a linear layer, which may leave gaps; we believe our work provides a more precise refinement and offers new insights.
>
> ---
>
> ### Response to Question 2 (the advantage of PLN)
>
> From the perspective of theoretical UAT analysis, we do not claim that PLN‑Nets offer any inherent advantage. Nevertheless, UAT analysis for LN‑Net/PLN architectures has been lacking, and our work mainly supplements this gap by studying the nonlinearity of LN through the lens of UAT.
>
> When it comes to training dynamics, we believe PLN‑Nets are easier to train in practice, as PLN inherits the favorable optimization properties of conventional normalization layers (e.g., scale invariance of the input). Our preliminary experiments support this claim: we observe that PLN‑Nets are insensitive to hyperparameter configurations, as shown in the VGG experiments in Section 6.
>
> ---
>
> Should there be any remaining questions, please feel free to raise them.
>
> ---
>
> ### References
>
> [1] Dao T, Gu A. Transformers are ssms: Generalized models and efficient algorithms through structured state space duality[J]. arXiv preprint arXiv:2405.21060, 2024.
>
> [2] Sun Y, Dong L, Huang S, et al. Retentive network: A successor to transformer for large language models[J]. arXiv preprint arXiv:2307.08621, 2023.
>
> [3] Wu Y, He K. Group normalization[C]//Proceedings of the European conference on computer vision (ECCV). 2018: 3-19.
>
> [4] De Ryck T, Lanthaler S, Mishra S. On the approximation of functions by tanh neural networks[J]. Neural Networks, 2021, 143: 732-750.
>
> [5] Yun C, Bhojanapalli S, Rawat A S, et al. Are transformers universal approximators of sequence-to-sequence functions?[J]. arXiv preprint arXiv:1912.10077, 2019.
>
> [6] Dunefsky J, Chlenski P, Nanda N. Transcoders find interpretable llm feature circuits[J]. Advances in Neural Information Processing Systems, 2024, 37: 24375-24410.

---

> > ### Author Rebuttal · Reviewer_pnmS · 2026-04-01
> >
> > As I mentioned in my original assessment, there are many positive aspects about this paper. My criticism concerning the type of result that we see here remains, though. The authors claim that these results should be interpreted as "not further exploration in UAT per se, but rather a deeper analysis of layer normalization". I do not see it quite like this. It is simply so that the definition of PNLs guarantees universality by allowing arbitrarily many normalization blocks. In that sense, the observation does not tell us anything deep about layer normalization; it is just another instance of the principle that arbitrarily many superpositions of ridge functions are dense. The exact same result would hold if the normalization step were replaced by any other nonlinear function. Hence, it is not specific to the normalization and does not show anything specific about its role.
> >
> > This criticism is fundamental, and it cannot be overcome by changing the manuscript.

---

> > > ### Author Response · Authors · 2026-04-07
> > >
> > > Thanks to the reviewer for the further reply. We will attempt to address the reviewer's concerns from the following perspectives.
> > >
> > > ---
> > >
> > > ### Further understanding of LN
> > >
> > > Beyond the UAT of PLN networks, our Theorem 4.1 further **quantitatively characterizes** the nonlinearity of LN. We note that previous work [7] only revealed the nonlinearity of LN but did not discuss the representation capacity of shallow LN networks, while most other works **either neglected LN or incorrectly approximated it as a linear transformation** (e.g., [5] and [6] mentioned in our previous reply). We believe our work is meaningful in correcting these inaccurate approximations.
> > >
> > > ---
> > >
> > > ### Further understanding of PLN networks
> > >
> > > It is worth noting that PLNs have already been adopted in existing work (e.g., [1] and [2] mentioned previously). We adopt the MLP form as a controlled setting to facilitate theoretical analysis. To the best of our knowledge, this specific MLP-based PLN architecture has not been used in prior work, which may make it appear artificially constructed. Nevertheless, we believe that analyzing this simplified setting reveals insights that extend to more complex architectures (e.g., sequential models), thereby providing theoretical understanding for existing designs.
> > >
> > > ---
> > >
> > > ### Response to "The exact same result would hold if the normalization step were replaced by any other nonlinear function"
> > >
> > > We agree with the reviewer's argument, and we again acknowledge that PLN networks do not exhibit better approximation rates. **However, we now highlight the NTK perspective (as also discussed in our response to reviewer WmME) to distinguish our PLN networks from arbitrary nonlinearities.** Our NTK analysis (see our response to Reviewer WmME) shows that PLN networks possess a higher effective rank of the NTK spectrum, which implies a better-conditioned NTK spectrum. This property ensures that PLN networks achieve better practical training performance, even if their theoretical approximation rates are inferior. The optimization properties of LN/PLN have been widely studied, and we hope this work fills the gap in theoretical analysis of the representation capacity of LN/PLN. While NTK is not directly related to UAT, we believe that **the ease of training** a universal approximator is equally important. Even if replacing LN with other ridge functions preserves the UAT property, such favorable NTK properties are **not guaranteed**. We have presented the NTK analysis in this rebuttal and will further incorporate it into the final manuscript.
> > >
> > >
> > > ---
> > >
> > > ### Additional References
> > >
> > > [7] Ni Y, Guo Y, Jia J, et al. On the nonlinearity of layer normalization[J]. arXiv preprint arXiv:2406.01255, 2024.

---

### Official Review · Reviewer_WmME · 2026-03-12

**Soundness:** 3
**Presentation:** 2
**Significance:** 3
**Originality:** 3
**Overall Recommendation:** 5
**Confidence:** 3

**Summary:**

The authors propose *parallel layer normalizations*, a scheme for applying normalization operations group-wise across a layer consisting of many such parallel groups, aiming to use the inherent nonlinearity of such operations to endow the network with nonlinear expressivity. Notably, the authors prove that even without conventional element-wise activation functions, a universal approximation theorem exists for even shallow networks interleaving linear layers and PLN layers, while expectedly equivalent networks using standard layer normalizations instead of PLNs do not have such approximation capacity. The authors engage in comprehensive analysis of convergence rates of such networks, and provide some experiments demonstrating practical utility of PLN networks.

**Compliance With Llm Reviewing Policy:**

Affirmed.

**Final Justification:**

I was originally quite positive about the contributions of the paper, and found that the rebuttal provided interesting additional perspective that addressed my main concerns and increased my positivity about the contributions of the paper. I acknowledge the other reviewers' concerns that the UAT result is not specific to layer norm in any real sense, but I do not find this to be a significant issue with the paper; rather, I interpret this theorem as being in support of the paper's core contribution: a new and relatively interesting layer type making use of existing understanding of layer norm.

**Key Questions For Authors:**

- The results presented in Table 1 show a suspiciously large discrepancy in performance between PLN and its competitors. Can the authors explain why PLN outperforms so significantly in this problem setting, despite being a relatively minor architectural change? In particular, I find the underperformance of ReLU networks to be quite surprising, especially if the experiment was designed in part to demonstrate universal approximation capabilities (which are of course shared by the baselines)? Does this difference in performance occur in distribution or in extrapolatory parts of the evaluation domain?
- I would also like to reiterate my above requests for a simple toy experiment demonstrating the behaviour of PLN networks compared to other baselines, e.g. ReLU networks or LN networks. This would help understand the inductive biases of PLN networks, and may explain why they outperform in certain experimental settings.
- This is a significant ask so is not strictly necessary, but a complentary analysis would be to look at the effect of PLN layers replacing activation functions on the NTK (Jacot et al. 2018) of the network. Do the authors have any thoughts on this?

Jacot, Arthur, Franck Gabriel, and Clément Hongler. "Neural tangent kernel: Convergence and generalization in neural networks." Advances in neural information processing systems 31 (2018).

**Limitations:**

Yes

**Strengths And Weaknesses:**

# Strengths #
- Soundness: The provided theory is substantial. Without having thorougly examined them, the proofs of this theory seem plausible and seem to have been cleverly constructed. Some experiments are provided.
- Presentation: Generally speaking, notation is appropriately dense, and figures and tables seem to be well-made. Some effort is made to situate the contributions of the paper in the literature.
- Significance and Originality: While an old area of research, investigating novel ways o incorporating nonlinearity into deep learning models remains useful dude to their prominence. To the best of my knowledge, this is the first work interpreting such norm structures as activation function-like objects and proving UAT results around this.

# Weaknesses
- Soundness: The provided experiments predominantly attempt to show that networks constructed with PLN as their nonlinearity source outperform other models, but do little to demonstrate the approximation capacity of these networks. It is doubtful that this outperformance would be exhibited for every problem setting (for example, I would expect PLN networks will extrapolate in magnitude in a bounded fashion, which will limit performance OOD in applications where an unbounded extrapolation is desirable) so there is some cherry-picking risk here. More informative experiments would directly target the approximation capacity of such networks, as this pertains directly to the main contribution of the paper, the UAT results. For example, it would be nice to see the trained outputs of networks using PLN vs LN networks, standard activation function networks etc on a toy problem with scalar input and output, such that we may better understand the behaviours of such networks on an intuitive level.
- Presentation: My main concern here is in how the contribution is presented. While the difference between PLN and Group Norm is explained in detail, I am not convinced that there is any significant difference in what they are mechanically. The real contribution is in interpreting this structure as a source of nonlinearity, and the corresponding theoretical analyses, and this should be highlighted more. Similarly, I think referring to PLNs as a form of *Layer Normalization* is incorrect, in that no normalization ever occurs at the layer level, and rather this is an application of generic "normalization / standardization", which of course is of much broader scope than the relatively specific Layer Normalization. This is semantic though, and not of great importance to the science of the paper. I also think that comparison to standard LN networks is significantly overdone throughout the paper, both due to the divergence of PLNs from the notion of Layer Normalization, and due to its closer resemblance (both in interpretation and operation) to standard activation functions or exotic nonlinearities. A much more interesting explanatory comparison would be for example to networks using tanh activation functions (as is done in the experiments), especially considering the recently discovered correspondence between LN and tanh (Zhu et al. 2025) which directly interprets LN as an elementwise tanh activation function in wide networks.
- Significance and originality: Despite how the paper is presented, I do not find the *machinery* introduced here to have significance or originality. Rather, it is the supporting theory that provides both of these merits, which does indeed deepen understanding of such networks.

Overall, I am positive about the theoretical contributions of the paper, but do not think in its current state the manuscript accurately portrays the contributions of the paper. If these presentation issues are addressed, I am willing to advocate for acceptance.

Zhu, Jiachen, et al. "Transformers without normalization." Proceedings of the computer vision and pattern recognition conference. 2025.

---

> ### Author Rebuttal · Authors · 2026-03-31
>
> ## Reply to the Reviewer WmME
>
> We thank the reviewer for the valuable suggestions and insightful questions.
>
> ---
>
> ### Clarification on the comparison of GN and PLN
>
> We first discuss GN and PLN in CNNs, where GroupNorm was originally defined. Remark 3.3 (lines 147–157) briefly highlights their differences. To make it more intuitive, take an image $X \in \mathbb{R}^{c * h * w}$ as an example. Both GN and PLN partition the input into groups and normalize within each group. Following the hyperparameter settings, GN can use $g$ groups, each of size $(c/g) * h * w$; PLN can use a fixed norm size $n_s$, each group of size $n_s * 1 * 1$. Thus PLN creates many more groups, potentially producing stronger nonlinearity. Previous work [1] also shows that GN has weaker nonlinearity than PLN in ResNet experiments. In theory, this design is essential for our UAT construction in position‑wise FFNs (Theorem 5.4) and generalizes to $1*1$ convolutions (Remark 5.3).
>
> GN is rarely used in sequential models. To our knowledge, there is no unified definition of GN for sequential data, as 2D inputs lack the concept of "channels". Some recent models (e.g., Mamba‑2 and RetNet, noted as [1] and [2] in our reply to the Reviewer pnmS) apply a normalization operation they call "GroupNorm", but this is closer to PLN than to the standard GN used in CNNs.
>
> Unifying 1D/2D/3D inputs, we clarify that our PLN is defined independently, with a focus on UAT analysis, rather than claiming precedence over these engineering adaptations.
>
> ---
>
> ### Response to the imprecision of our contributions
>
> Based on the above — PLN differs from GN in CNNs, and while PLN‑like designs appear in some sequential models, they are not widely used — we propose to add the following statement to the end of the Introduction to clarify our contributions:
>
> > “Although a structure similar to PLN‑Net has been used in Mamba‑2 and RetNet , such designs are not yet widely adopted in current architectures. Our theoretical results aim to further explore the potential of PLN‑Nets, either as standalone models or as substructures, and to fill the gap in understanding the nonlinearity of LN/LN‑Net through the lens of approximation capacity.”
>
> We believe this addition makes our contributions more focused and clear.
>
> ---
>
> ### Response to Question 1
>
> To clarify, the evaluation set follows a different sampling scheme from the training set but the results remain consistent. We record gradient flow on a representative subset of the original PINN PDE settings. PLN(4) maintains large gradients across layers ($\sim 10^{3}$), indicating stronger cross-layer optimization. In contrast, ReLU often has near-zero gradients in early layers, leaving shallow features undertrained for oscillatory Helmholtz solutions, which may account for its low-loss yet high-error behavior. Besides, we believe another important reason is that ReLU does not inherently capture higher‑order (≥2) information, which may hinder accurate learning of PDEs.
>
> ---
>
> ### Response to Weakness 1 and Question 2 (toy experiments)
>
> We initially explored the **potential** of PLN‑Nets on toy tasks (target $f(\mathbf{x}) = 3\sin(x_1x_2\cdots x_8)$ with 1000 samples) but found that PLN only performed comparably to other activation functions. The best training MSEs among total 450 combinations of settings for each activation are: 9.01e-15 for SAT, 9.99e-15 for ReLU; 1.30e-14 for PLN and Tanh; and 1.65e-14 for SiLU. They are close and do not show the potential so we omitted these results in the manuscript. In contrast, using LN gave best MSE > 3, confirming its weak representation capacity.
>
> ---
>
> ### Response to Question 3
>
> Empirical NTK analysis (64 1D samples mapping $R\to R$) in PINN PDE shows PLN yields a better‑conditioned NTK spectrum, with higher effective rank (PLN(4): 8, ReLU: 4, Tanh: 2, SAT: 1, LN: 1). In another investigation on 500 random samples (mapping $R^8\to R$) using an MLP (width 64, depth 4), the effective ranks are: PLN(4): 198, ReLU: 119, Tanh: 35, SAT: 43, LN: 7.
>
> ---
>
> ### Comparison with (Zhu et al. 2025)
>
> We applied their DYT in our translation experiments and find it unable to train, even though we sweep the learning rates from 1e-3 to 1e-6. We believe the reason is that Post-LN mode is used in our Transformers while Pre-LN (better to relieve gradient vanishing) is applied in (Zhu et al. 2025). Using the Pre-LN structure, the BLEUs are: DYT(29.11), LN(31.95), PLN(35.58), where PLN still performs the best. And we can say that DYT can not completely replace normalization layers for it may be hard to train in some scenarios.
>
> ---
>
> Should there be any remaining questions or the experimental details omitted here due to space constraints, please feel free to raise them and we may discuss in the next round.
>
> ---
>
> #### References
>
> [1] Ni Y, Guo Y, Jia J, et al. On the nonlinearity of layer normalization[J]. arXiv preprint arXiv:2406.01255, 2024.

---

> > ### Author Rebuttal · Reviewer_WmME · 2026-03-31
> >
> > I thank the authors for the additional discussion and results. I particularly appreciate the NTK spectrum effective rank results, which are very interesting and reveal some insights into the deeper behaviour of PLN networks. I am satisfied with the responses to my original questions, and will thus increase my score to 5.

---

> > > ### Author Response · Authors · 2026-04-07
> > >
> > > Thank you very much for your positive feedback and for raising your score to 5. We are delighted that you provide the additional perspectives, especially the NTK spectrum effective rank analysis, insightful and helpful for understanding PLN networks. Your original comments were very constructive and helped us improve the paper significantly. We truly appreciate your time and expertise.

---

### Decision · Program_Chairs · 2026-04-30

**Decision:**

Reject

**Comment:**

*Motivation:* Designing a normalization layer capable of ensuring universal function approximation in a single-layer neural network, without relying on additional nonlinear activations.

*Main Contribution:* The paper establishes that a single linear layer with layer normalization (LN) does not satisfy the universal approximation property, and proposes a novel normalization layer, called PLN, that implicitly implements nonlinear activation, thereby guaranteeing universal approximation without explicit nonlinearity.

*Review Summary:* Since normalization layers are primarily developed to improve the optimization and training stability of neural networks, most reviewers found the motivation somewhat unclear. Specifically, reviewers questioned why a high-capacity normalization layer is desirable if it does not demonstrably accelerate training. More broadly, the argument from expressivity alone was seen as insufficient to justify why PLN should be preferred over LN in practice.

*Rebuttal Summary:* Reviewers confirmed that the authors' response did not adequately address their concerns regarding the motivation behind PLN.

*AC Opinion:* While the theoretical result is interesting, and the design of effective normalization layers for LLMs is undoubtedly important, I believe the current research narrative may be confusing to a broad audience and would benefit from revision. In particular, I encourage the authors to address the following:

- The paper focuses primarily on expressivity and does not discuss the optimization implications of PLN, which is central to how normalization layers are evaluated in practice.
- There is a significant gap between theory and experiment: the theoretical analysis of LN omits nonlinear activations, yet in practice LN is employed primarily to enhance optimization.

I vote for rejection, but look forward to seeing a revised version in a future venue.